# A missense mutation in human *INSC* causes peripheral neuropathy

Jui-Yu Yeh [1], Hua-Chuan Chao[2,3,4], Cheng-Li Hong[1], Yu-Chien Hung[1], Fei-Yang Tzou [1], Cheng-Tsung Hsiao[2,5], Jeng-Lin Li [6,7], Wen-Jie Chen[8,9], Cheng-Ta Chou [10], Yu-Shuen Tsai[11], Yi-Chu Liao [2,5,12], Yu-Chun Lin [13,14], Suewei Lin [8], Shu-Yi Huang [15], Marina Kennerson[16,17,18], Yi-Chung Lee [2,5,12 ✉] & Chih-Chiang Chan [1 ✉]

## Abstract

PAR3/INSC/LGN form an evolutionarily conserved complex required for asymmetric cell division in the developing brain, but its post-developmental function and disease relevance in the peripheral nervous system (PNS) remains unknown. We mapped a new locus for axonal Charcot–Marie-Tooth disease (CMT2) and identified a missense mutation c.209 T > G (p.Met70Arg) in the *INSC* gene. Modeling the *INSC^M70R* variant in *Drosophila*, we showed that it caused proprioceptive defects in adult flies, leading to gait defects resembling those in CMT2 patients. Cellularly, PAR3/INSC/LGN dysfunction caused tubulin aggregation and necrotic neurodegeneration, with microtubule-stabilizing agents rescuing both morphological and functional defects of the *INSC^M70R* mutation in the PNS. Our findings underscore the critical role of the PAR3/INSC/LGN machinery in the adult PNS and highlight a potential therapeutic target for *INSC*-associated CMT2.

**Keywords** Charcot–Marie–Tooth Neuropathy Type 2; *Inscuteable*; Microtubule-Stabilizing Agents; Necrosis; Proprioception
**Subject Categories** Genetics, Gene Therapy & Genetic Disease; Neuroscience

## Introduction

Charcot–Marie–Tooth disease (CMT) is a clinically and genetically heterogeneous group of inherited neuropathies characterized by slowly progressive distal limb weakness and atrophy, sensory loss,

foot deformities, and depressed tendon reflexes (Rossor et al, 2016). The population prevalence of CMT is estimated at 1 in 2500 individuals (Rossor et al, 2016). Clinically, CMT is classified into demyelinating (CMT1) or axonal (CMT2) types based on forearm motor nerve conduction velocities (MNCVs) below or above 38 m/s, respectively (Harding and Thomas, 1980). Although mutations in more than 100 genes have been implicated in CMT, the genetic diagnosis for more than 70% of CMT2 patients remains elusive (Cortese et al, 2020).

Several common gene mutations have been linked to CMT2, including *MFN2* (CMT2A), *RAB7* (CMT2B), *GARS* (CMT2D), and *NEFL* (CMT2E). Previous literature shows that these mutations affect mitochondrial trafficking (CMT2A), microtubule trafficking of lysosomes (CMT2B), acetylated tubulin (CMT2D), and neurofilaments (CMT2E) (Markworth et al, 2021). Notably, these CMT2 subtypes share a common defect of decreased microtubule stabilization caused by disrupting α-tubulin acetylation (Ackerley et al, 2006; Mo et al, 2018; Brownlees et al, 2002), which subsequently leads to microtubule breakdown and axonal defects. Since microtubules regulate axonal transport by forming a dynamic network that enables efficient intraneuronal transport, these findings suggest that microtubule destabilization within axons can be a common feature across genetically diverse forms of CMT2. Understanding the underlying genetic and molecular mechanisms can aid in improving the diagnosis, treatment, and management of CMT and provide crucial insights into the pathogenic mechanisms of CMT2.

During the development of the central nervous system, asymmetric cell division plays a critical role in balancing stem cell self-renewal and differentiation in the brain (Kraut and Campos-Ortega, 1996; Kraut et al, 1996; Kuchinke et al, 1998). The PAR3(PARD3)/INSC/LGN(GPSM2) (hereafter referred as PIL)

[1]Graduate Institute of Physiology, National Taiwan University, Taipei, Taiwan. [2]Department of Neurology, National Yang Ming Chiao Tung University School of Medicine, Taipei, Taiwan. [3]Institute of Clinical Medicine, National Yang Ming Chiao Tung University, Taipei, Taiwan. [4]Division of Neurology, Department of Medicine, Taoyuan General Hospital, Ministry of Health and Welfare, Taoyuan, Taiwan. [5]Department of Neurology, Neurological Institute, Taipei Veterans General Hospital, Taipei, Taiwan. [6]Ph.D. Program in Translational Medicine, National Taiwan University and Academia Sinica, Taipei, Taiwan. [7]Department of Neurology, National Taiwan University Hospital Jinshan Branch, New Taipei City, Taiwan. [8]Institute of Molecular Biology, Academia Sinica, Taipei, Taiwan. [9]Taiwan International Graduate Program in Interdisciplinary Neuroscience, National Cheng Kung University and Academia Sinica, Tapiei, Taiwan. [10]Department of Neurology, Neurological Institute, Taichung Veterans General Hospital, Taichung, Taiwan. [11]Cancer and Immunology Research Center, National Yang Ming Chiao Tung University, Taipei, Taiwan. [12]Brain Research Center, National Yang Ming Chiao Tung University School of Medicine, Taipei, Taiwan. [13]Institute of Molecular Medicine, National Tsing Hua University, HsinChu, Taiwan. [14]Department of Medical Science, National Tsing Hua University, HsinChu, Taiwan. [15]Department of Medical Research, National Taiwan University Hospital, Taipei, Taiwan. [16]Northcott Neuroscience Laboratory, ANZAC Research Institute, Sydney Local Health District, Concord, NSW, Australia. [17]School of Medical Sciences, Faculty of Medicine and Health, University of Sydney, Sydney, NSW, Australia. [18]Molecular Medicine Laboratory, Concord Hospital, Concord, NSW, Australia. ✉E-mail: ycli@vghtpe.gov.tw; chancc1@ntu.edu.tw

complex is crucial for asymmetric cell division, regulating microtubules to establish cell polarity. Mutations of the PIL complex have been shown to cause neurodevelopmental disorders, including neural tube defects resulting from the pathogenic variants of *PAR3* (Chen et al, 2017) and Chudley-McCullough syndrome (CMS) caused by *LGN* variants (Walsh et al, 2010). While *LGN* mutations in CMS patient cells impaired microtubule stability, *Par3*-deficient mice with neural defects exhibited unstable microtubules in neural progenitor cells (Chen et al, 2013). Despite the well-established role of PAR3 and LGN during CNS development, there is no published evidence for the requirement of the PIL complex in the adult peripheral nervous system (PNS).

Unlike *PAR3* and *LGN*, *INSC* has never been linked to any genetic disorder. INSC was first identified in *Drosophila* larval neuroblasts (Kraut and Campos-Ortega, 1996). *Drosophila* LGN is required for INSC to asymmetrically localize during asymmetric cell division (Yu et al, 2000). INSC and LGN participate in the cytoskeleton-membrane association in the apical side of neuroblasts and induce pulling forces on the astral microtubule for the asymmetric division (Yu et al, 2006). After the association between INSC and LGN, different modes of regulation on asymmetric division have been proposed. In one scenario, the dynein-adaptor protein NuMA competes with INSC for LGN binding (Zhu et al, 2011). The LGN-bound NuMA complex then recruits dynein, a microtubule motor protein, to induce pulling forces on the astral microtubule for asymmetric divisions (Fig. 1A) (Wang and Chia, 2005). Alternatively, INSC and LGN can form stable tetramers to regulate asymmetric cell division without involving dynein (Culurgioni et al, 2018). Both findings indicate the importance of INSC-LGN association in regulating microtubules. LGN encodes an evolutionarily conserved tetratricopeptide repeat (TPR) motif that interacts with the LGN-binding domain of INSC (Fig. 1B) (Yu et al, 2000, 2003). Whether the PIL complex, especially INSC, may be involved in CMT2 pathology via its role in microtubule regulation is not known.

The *Drosophila* femoral chordotonal organ (FeCO) is considered functionally homologous to human muscle spindles, the primary proprioceptive sensory organs. Tuthill and Azim, 2018 playing a critical role in detecting mechanical stretches such as muscle tension and joint position (Chen et al, 2021). The chordotonal organ is formed by scolopidia, the basic unit of mechanoreceptor organ comprising over a hundred of mechanosensory neurons (Lipovšek et al, 1999). FeCO neurons are crucial for the precise control of leg movements during behaviors like walking and target reaching. Proprioceptive cell death is known to cause neurological disorders (Ilieva et al, 2008). Patients of Charcot–Marie–Tooth (CMT) disease show proprioception defects, making FeCO a suitable organ for studying peripheral neuropathy because of its microtubule-rich structure and highly conserved function. Taken together, the chordotonal organs in general, and the FeCO neurons in specific, provide an excellent platform for studying proprioceptive biology and unraveling the underlying mechanisms of peripheral neuropathy.

Here, we describe the discovery of the pathogenic variant p.Met70Arg (M70R) in the *INSC* gene to be associated with Charcot–Marie-Tooth disease type 2 (CMT2). We developed a corresponding fly model to study the effect of the PIL complex in the adult PNS. Our study demonstrated the PIL complex regulates microtubule stability in the adult PNS and suggests enhancing microtubule acetylation as a potential therapeutic strategy for addressing CMT disease caused by the *INSC* mutation.

## Results

### Identification of *INSC* p.Met70Arg in a pedigree with autosomal dominant CMT2

We report a three-generation Asian family with autosomal dominant CMT2, in which eight affected and four unaffected individuals were recruited (Fig. 1C). The genetic diagnosis of the proband (III-8) remained unsolved after extensive genetic analysis. All patients presented with slowly progressive distal sensory loss and gait unsteadiness with absent or mild weakness in the limbs with onset ranging from age 7 to 29 years (Table 1; Appendix Fig. S1). The two eldest patients (II-2 and II-3) had distal weakness after 55 years of age and required a walker or cane to assist with ambulation (Movie EV1). The nerve conduction studies revealed axonal sensorimotor polyneuropathy with sensory predominant features (Appendix Table S1). Genome-wide linkage analysis in this family mapped the CMT2 locus to a 17-Mb interval on chromosome 11p15.1-15.4, flanked by the markers rs231359 and rs7118901 (Appendix Fig. S2). A multipoint LOD score of 3.0 established a significant linkage to the genomic interval 11:2,694,606–19,743,250 (hg37). Whole genome sequencing of patients III-3 and III-8 (Appendix Table S2) identified a heterozygous missense variant, c.209 T > G (p.Met70Arg), in the *INSC* gene (RefSeq NM_001031853.5) (Fig. 1D), which localizes to the linkage interval. The variant fully segregated with the disease and was absent in 1517 Taiwanese health controls from the Taiwan biobank database. The p.Met70Arg variant is located within a ~35 amino acid motif of INSC that is critical for binding to LGN and is functionally conserved across species (Postiglione et al, 2011; Culurgioni et al, 2011; Yuzawa et al, 2011). Bioinformatic tools predicted the *INSC* p.Met70Arg variant to be a disease-causing by CADD v1.6 (Rentzsch et al, 2019) (Phred score 23.5) and MutationTaster2 (Schwarz et al, 2014) (probability value: 0.989). Pathogenicity was also supported by the variant being absent in the gnomAD (Karczewski et al, 2020) and BRAVO databases (https://bravo.sph.umich.edu/freeze8/hg38/), which in total contain genetic variants from over 800,000 human genomes.

### Loss-of-function of PIL complex of the adult PNS cause proprioceptive defects in aging fly

The requirement of INSC in neurodevelopment is evolutionary conserved (Postiglione et al, 2011; Culurgioni et al, 2011). To evaluate the underlying mechanisms of *INSC* in the CMT2 patients, we first depleted *Drosophila Insc* (*dInsc*) by RNAi to determine the effect on the locomotion of flies. We employed an inducible pan-neuronal *elav-GS*-Gal4 driver, which is a modified Gal4/UAS system. This enabled transgene expression upon RU486 drug treatment, thereby allowing adult-onset knockdown of the gene and avoiding any developmental influences. In the locomotor assay, *dInsc*-RNAi flies exhibited normal behavior at day 3 post-eclosion, but showed a climbing deficit at week 1, indicating adult-onset degeneration (Fig. 2A; Appendix Fig. S3D). To investigate whether the same machinery utilized to regulate asymmetric cell division in

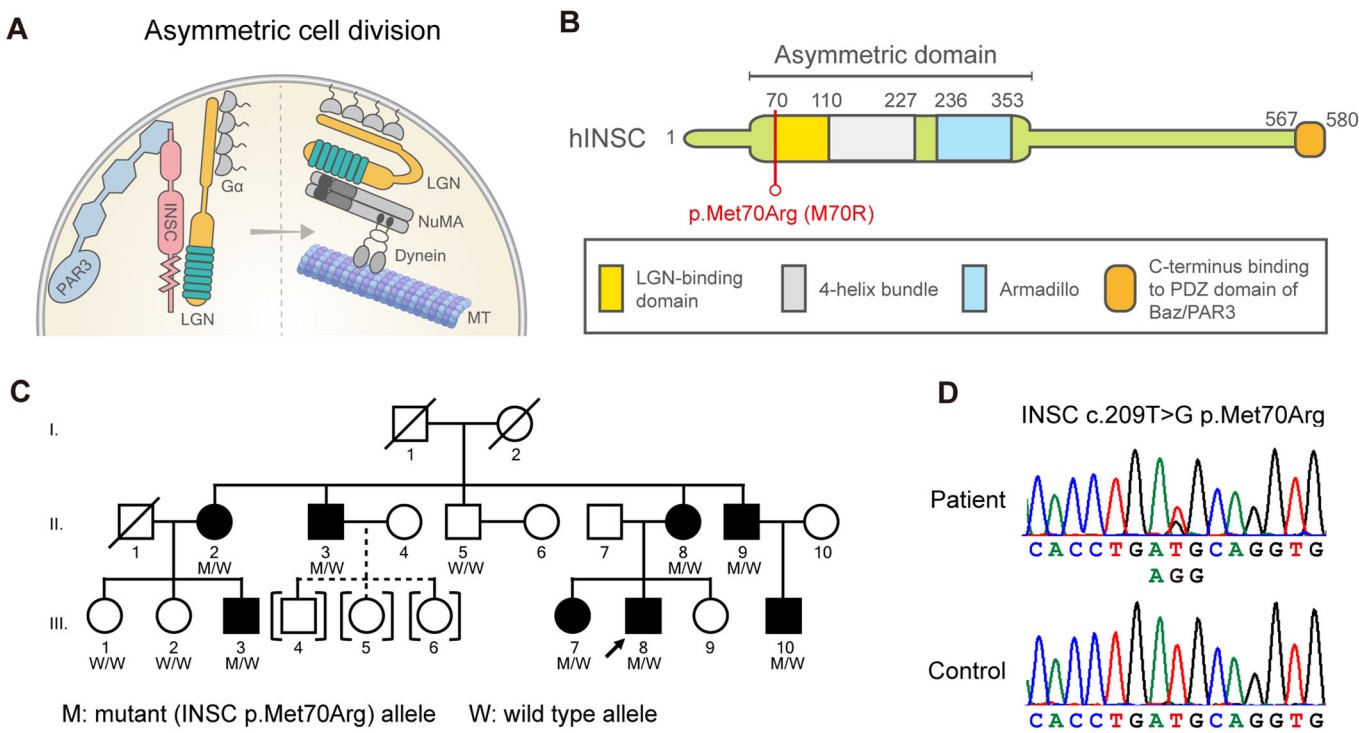

**Figure 1. Identification of a CMT2-associated *INSC* variant.**

(A) Schematic figure of asymmetric cell division in neuroblast. Inscuteable (INSC), an adaptor protein in asymmetric cell division, regulates microtubules by binding with PAR3 and LGN. (B) The *INSC* p.Met70Arg variant resides in the LGN-binding domain of the human INSC protein. (C) The CMT2 pedigree carrying the *INSC* mutation. "M" represents the *INSC* mutant allele, and "W" stands for wild-type allele. Open symbol: unaffected; filled symbol: affected; symbol with a diagonal line: deceased; arrow: proband; squares: males; circles: females; brackets with dashed line: adopted into the pedigree. (D) Sanger sequencing traces confirming the c. 290 T > G (p.Met70Arg, M70R) variant.

neuroblasts during brain development is also required in the adult PNS, we knocked down other PIL complex genes in the adult PNS. The knockdown efficiency of each component is shown in Appendix Figs. S3D and S4A,B. Interestingly, knocking down *bazooka* (the fly homolog of human *PAR3*) or *pins* (the fly homolog of human *LGN*) also resulted in climbing defects (Fig. 2A; Appendix Fig. S3D). Rescue experiments were performed using the *dInsc^InSITE* Gal4 driver, an enhancer trap allele with Gal4 insertion in the *dInsc* gene, resulting in the disruption of *dInsc* expression. Heterozygous *dInsc^InSITE* flies exhibited impaired locomotion that was exacerbated in the *dInsc*-RNAi background, suggesting a dosage-dependent phenotype. The locomotor impairment was rescued by transgenic expression of human *INSC* (*hINSC^WT*) but not the disease-associated *hINSC^M70R* (Fig. 2B), suggesting that the locomotor function of INSC is conserved between human and flies, and the p.M70R mutation represents a loss-of-function mutation, likely rendering haploinsufficiency.

Next, we investigated the impact of the human *INSC* mutation of p.M70R on peripheral neuropathy by establishing a fly model. To generate disease-relevant flies with the p.M70R mutation, the corresponding amino acid residue (K305) in the endogenous *dInsc* locus was edited using seamless CRISPR-Cas9 techniques (Appendix Fig. S5A). Although the amino acid residues are not conserved, they are located within a stretch of structurally and functionally conserved motif critical for interacting with LGN (Yuzawa et al, 2011; Culurgioni et al, 2018). Two alleles, *dInsc^K305R* (p.Lys305Arg)

to mimic the disease variant and *dInsc^K305M* (p.Lys305Met) to represent the human wild type were engineered (Fig. 2C; Appendix Fig. S5B). While *dInsc^K305M* behaved similarly to the wild type in climbing activity, *dInsc^K305R* flies exhibited a dosage-dependent, progressive locomotion defect, supporting the notion that *hINSC^M70R* is a haploinsufficient variant (Fig. 2D).

To investigate the post-developmental expression of *dInsc* in the PNS of flies, the *dInsc^1407*-Gal4 driver was employed, which has been extensively utilized in previous studies (Luo et al, 1994; Betschinger et al, 2006) to recapitulate the *dInsc* gene activity. As previously reported, we confirmed *dInsc* expression in the larval brain and gut during larval development (Kraut and Campos-Ortega, 1996). Furthermore, our results revealed additional labeling in the leg discs but not the wing discs (Appendix Fig. S6A). To study the subcellular localization of Insc, we generated a UAS-*dInsc*-EGFP transgene which was then driven by *Insc^1407*-Gal4. The apical localization of *dInsc*-EGFP in a crescent-shaped pattern in the larval neuroblasts was also confirmed (Appendix Fig. S6B) as previously reported in an immunostaining study (Kraut et al, 1996). Notably, in the adult leg, the *mCD8-GFP* signal was present in proprioceptive structures, including the femoral chordotonal organ (FeCO), stretch receptors, and tibiotarsal chordotonal organs (tiCO) (Fig. 2E–H), in a pattern similar to *mCD8-GFP* signal expressed under a proprioceptor-specific driver *Iav*-Gal4 (Fig. EV1A). Knockdown of the PIL genes with the *Iav*-Gal4 showed similar locomotor impairments, indicating the PIL proteins are

**Table 1. Clinical manifestations of the affected individuals carrying *INSC* p.Met70Arg.**

| Subject | II-2 | II-3 | II-8 | II-9 | III-3 | III-7 | III-8 | III-10 |
|---|---|---|---|---|---|---|---|---|
| Age at exam (y) | 59 | 57 | 52 | 49 | 30 | 25 | 22 | 21 |
| Age at onset (y) | 19 | 20 | 20 | 12 | 29 | 22 | 7 | 7 |
| Sex | Female | Male | Female | Male | Male | Female | Male | Male |
| Onset symptom | Unsteady gait | Unsteady gait | Unsteady gait | Unsteady gait | Unsteady gait | Unsteady gait | Unsteady gait | Unsteady gait |
| Limb weakness UL (MRC) | Distal: 4 | Distal: 4+ | Distal: 4+ | Normal | Normal | Normal | Normal | Normal |
| Limb weakness LL (MRC) | Distal: 3 | Distal: 4 | Normal | Normal | Normal | Normal | Normal | Distal: 4+ |
| Foot deformity | Pes cavus | Pes cavus | Pes cavus | Pes cavus | Pes planus | Pes cavus | Pes planus | Normal |
| Biceps/triceps DTR | Absent | Absent | Absent | Absent | Reduced | Absent | Reduced | Absent |
| Knee/ankle DTRs | Absent | Absent | Absent | Absent | Absent | Absent | Absent | Absent |
| Pinprick sensation | ↓Distal to knee | ↓Distal to knee | ↓Distal to knee | ↓Distal to ankle | Normal | Normal | ↓Distal to knee | ↓Fingers and distal to knee |
| Vibration sensation | ↓Distal to ankle | ↓Toes | ↓Distal to knee | ↓Toes | ↓Toes | ↓Toes | ↓Distal to knee | ↓Distal to wrist and knee |
| Paresthesia | Hands, feet, legs | Hands, feet, legs | Fingers and feet | Absent | Absent | Absent | Absent | Absent |
| Ulnar nerve MNCV (m/s) | 63.1 | 51 | 58.8 | 48 | 59.4 | 63.6 | 50 | 54.7 |
| Ulnar nerve cMAP (mV) | 6.9 | 9.7 | 8.1 | 9.7 | 10.1 | 8 | 11 | 11.7 |
| Ulnar nerve SNAP (uV) | NR | NR | NR | 5 | NR | NR | NR | NR |

*MRC* Medical Research Council scale, *UL* upper limbs, *LL* lower limbs, *DTR* deep tendon reflex, *MNCV* motor nerve conduction velocity, *cMAP* compound motor action potential, *SNAP* sensory nerve action potential, ↓ decreased, *NR* not recordable, *NA* not available.
UL and LL distal weakness assessed by first dorsal interosseous and anterior tibialis.

required in the proprioceptive organs (Fig. 2I). In addition to recapitulating the molecular mechanisms, *Drosophila* models of neurodegenerative diseases have also been shown to share sufficient molecular machineries, allowing the measurement of locomotive characteristics such as gait and tremor (Gonçalves et al, 2022; Sreedharan et al, 2015). An automated leg-tracking system was utilized to analyze and quantify leg trajectory in aged flies (Wu et al, 2019). We characterized gait features, including footprint regularity, stride length, ratio of hind/mid (T3/T2) legs, and leg intersection domain. By comparing the gait signatures between the control and *dInsc* knockdown flies, significant changes were observed in gait patterns. Specifically, the *dInsc*-RNAi flies exhibited poor footprint regularity (Fig. 2J,K), increased stride length in the hind (T3) leg (Fig. 2L), and the ratio of hind/mid (T3/T2) legs (Fig. 2M), as well as uncoordinated leg displacement with an enlarged leg intersection domain (Fig. 2N). These alterations in gait resembled the walking difficulties and movement dysfunction observed in patients with CMT disease (Appendix Fig. S7; Movies EV2 and EV3).

## Neurons of loss-of-function of PIL complex displayed adult-onset necrosis

To visualize degenerating neurons, dual-staining with propidium iodide (PI) as a marker for degeneration and DAPI to label the cell nuclei was performed (Venkatachalam et al, 2008; Klemm et al, 2021). The colocalization of both markers enabled the distinguishment between healthy (DAPI + , PI−), dying (DAPI + , PI + ), or dead (DAPI−, PI + ) cells (Fig. 3A). The loss of *hINSC* in the SH-SY5Y neuroblastoma cells induced necrosis, as two independent clones stably expressing *hINSC*-shRNA exhibited a higher ratio of DAPI-PI colocalization compared to the control clone (Fig. 3B,C). As a control, the hINSC protein and mRNA levels of the stable knockdown cells are shown in Appendix Fig. S3A–C.

In FeCO neurons of PIL knockdown flies, no difference was observed in PI intensity or colocalization of DAPI and PI signals in week 1. However, by week 3, elevated PI staining and increased DAPI-PI colocalization was observed, which was inversely correlated with locomotor decline, reflecting an association between neuronal loss and the progression of neurodegeneration over time (Fig. 3D,E). The homozygous $dInsc^{K305R}$ (R/R) flies showed higher DAPI-PI colocalization and more severe locomotor defects compared to the heterozygous $dInsc^{K305R}$ (K/R) and wild-type *dInsc* (K/K) flies (Figs. 3F and EV2A). In addition, the level of necrosis was associated with the functional decline, while both neuronal loss and function were alleviated by transgenic expression of the human wild-type transgene (Figs. 3G and EV2B).

## p.M70R variant reduced INSC expression in old age and differentially affects interaction with LGN and PAR3

To monitor the *hINSC* level in patients, we prepared cDNA and protein lysate from the peripheral blood mononuclear cells

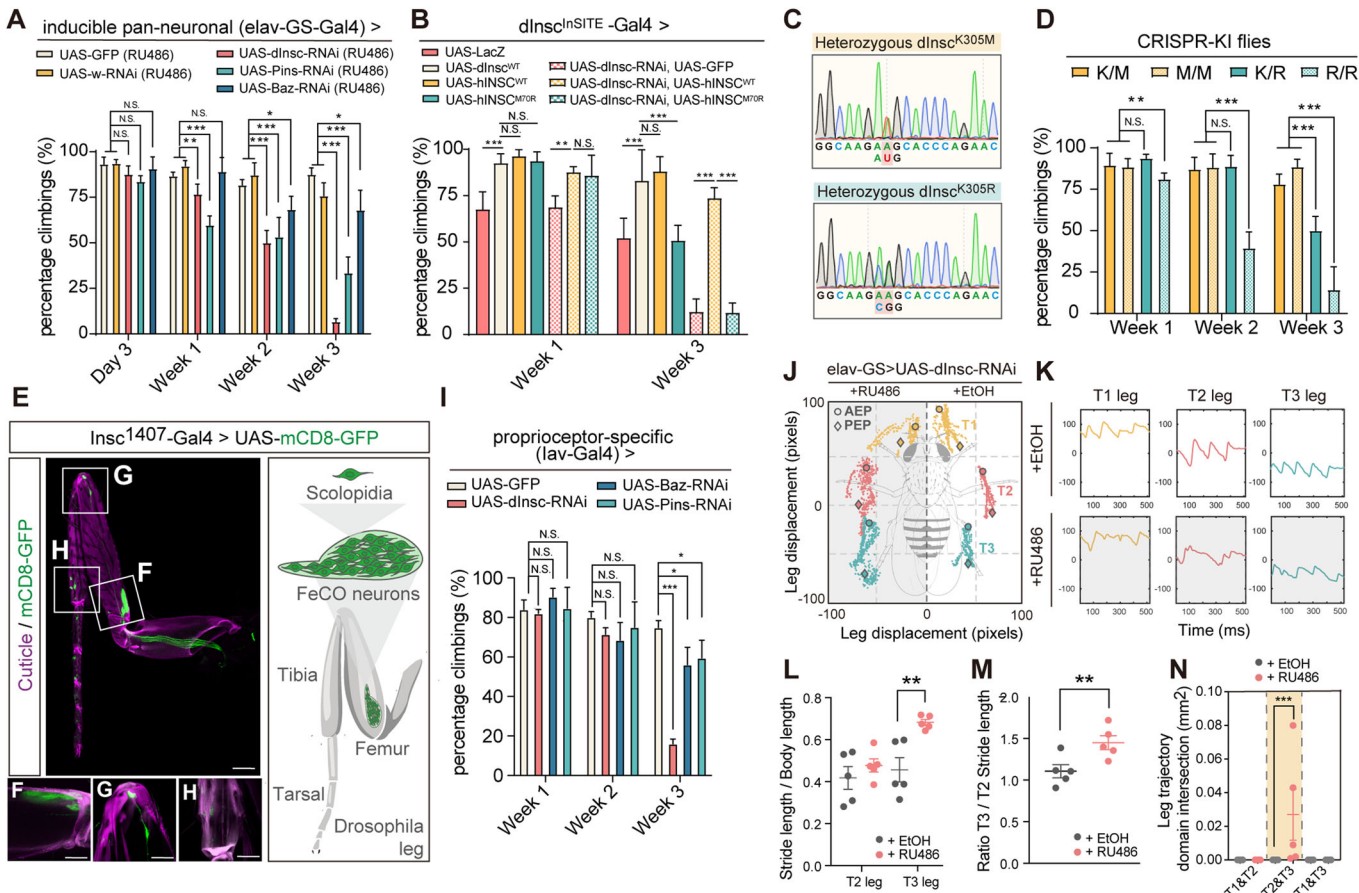

**Figure 2. Adult-onset depletion of PIL complex causes locomotor and proprioceptive defects.**

(A) Quantification of the climbing activity of 3-day-old to 3-week-old adult flies with *dInsc*-RNAi, *Baz*-RNAi and *Pins*-RNAi under the control of inducible pan-neuronal driver (*elav-GS*-Gal4) upon feeding with RU486, comparing with the age-matched controls (*mCD8-GFP*) and (*w*-RNAi); n = 100–249 flies/genotype from five independent fly crosses. (B) Quantification of the climbing activity of 1- and 3-week-old flies with overexpressing *dInsc-WT*, *LacZ*, *hINSC^WT*, *hINSC^M70R*, *dInsc*-RNAi, UAS-*GFP* and the groups co-overexpressing *dInsc*-RNAi with *hINSC^WT* or *hINSC^M70R* under the control of *dInsc^InSITE*-Gal4 driver; n = 57–110 flies/genotype from five independent fly crosses. (C) Sanger sequencing traces confirming the heterozygous *dInsc^K305M* and *dInsc^K305R* CRISPR-KI flies. (D) Quantification of the climbing activity of 1–3-week-old heterozygous +/*dInsc^K305M* (K/M) and +/*dInsc^K305R* (K/R), homozygous *dInsc^K305M* / *dInsc^K305M* (M/M) and *dInsc^K305R*/ *dInsc^K305R* (R/R) CRISPR-KI flies; n = 49–56 flies/genotype from five independent fly crosses. (E) Thoracic segments 1 (T1) leg of an adult fly expressing *mCD8-GFP* (green) under the control of *dInsc^1407*-Gal4. (Right) a schematic of the FeCO neuron in an adult leg. Scale bars: 50 μm. Magenta is the auto-fluorescence of the cuticle. (F–H) Magnified images of the femoral chordotonal organ (F), stretch receptor (G), and tibiotarsal chordotonal organ (H). Scale bars: 20 μm. (I) Quantification of the climbing activity of 1–3-week-old *dInsc*-RNAi, *Baz*-RNAi and *Pins*-RNAi flies under the control of *lav*-Gal4, compared with the age-matched controls (*mCD8-GFP*); n = 44–129 flies/genotype from 5 independent fly crosses. (J) Body-centered leg trajectories plot of 3-week-old *elav-GS*-Gal4>UAS-*dInsc*-RNAi flies feeding with RU486, compared with the age-matched solvent feeding controls. AEP: anterior extreme positions; PEP: posterior extreme positions. (K) The leg displacement plots of T1 (yellow), T2 (pink) and T3 (cyan) legs from the same groups as (J). (L) Quantification of T2 and T3 leg stride length normalized to body length; n = 5 flies/condition from three independent fly crosses. (M) Quantification of the ratio T3/T2 stride length of RU486-fed group, compared with the solvent-fed control; n = 5 flies/condition from three independent fly crosses. (N) Quantification of the leg trajectory domain intersection of RU486-fed group, compared with the solvent-fed control; n = 5 flies/condition from three independent fly crosses. Data information: Error bars indicate mean ± SEM. Statistical analysis was performed using two-tailed Student's *t* test. *P < 0.05, **P < 0.01, ***P < 0.001. Source data are available online for this figure.

(PBMCs) of both affected and unaffected family members. Both *hINSC* mRNA and protein levels were significantly decreased in young affected individuals compared to young healthy individuals. The *hINSC* mRNA and protein abundance further decreased in older affected individuals (Fig. 4A,B). Similarly, we found the mRNA and protein level of *R/R* decrease significantly in aging flies, compared to *K/K* and *M/M* control (Fig. EV2A–C). Furthermore, we observed an age-dependent decline in hINSC^M70R protein levels, particularly in the dendrites of FeCO neurons. This observation suggests that the reduction in protein expression is inherent to the p.M70R mutation (Fig. 4C,D; Appendix Fig. S4C). Besides, the

colocalization of *hINSC^M70R-EGFP* and *Pins-mCherry* decreased in the FeCO neurons of 3-week-old flies compared with *hINSC^WT* control (Fig. 4E,F). Since the p.M70R mutation resides within the LGN-binding motif of hINSC, the effect of the mutation on the association with the PIL complex was investigated. In human SH-SY5Y cells, both co-immunostaining and co-immunoprecipitation (Co-IP) of hINSC with LGN showed decreased hINSC-LGN colocalization in hINSC^M70R-expressing cells (Figs. 4G,H and EV3D). Conversely, co-immunostaining and Co-IP with HA-PAR3 showed higher colocalization between PAR3 and hINSC^M70R (Figs. 4I,J and EV2D). Similarly, confocal imaging of FeCO neurons

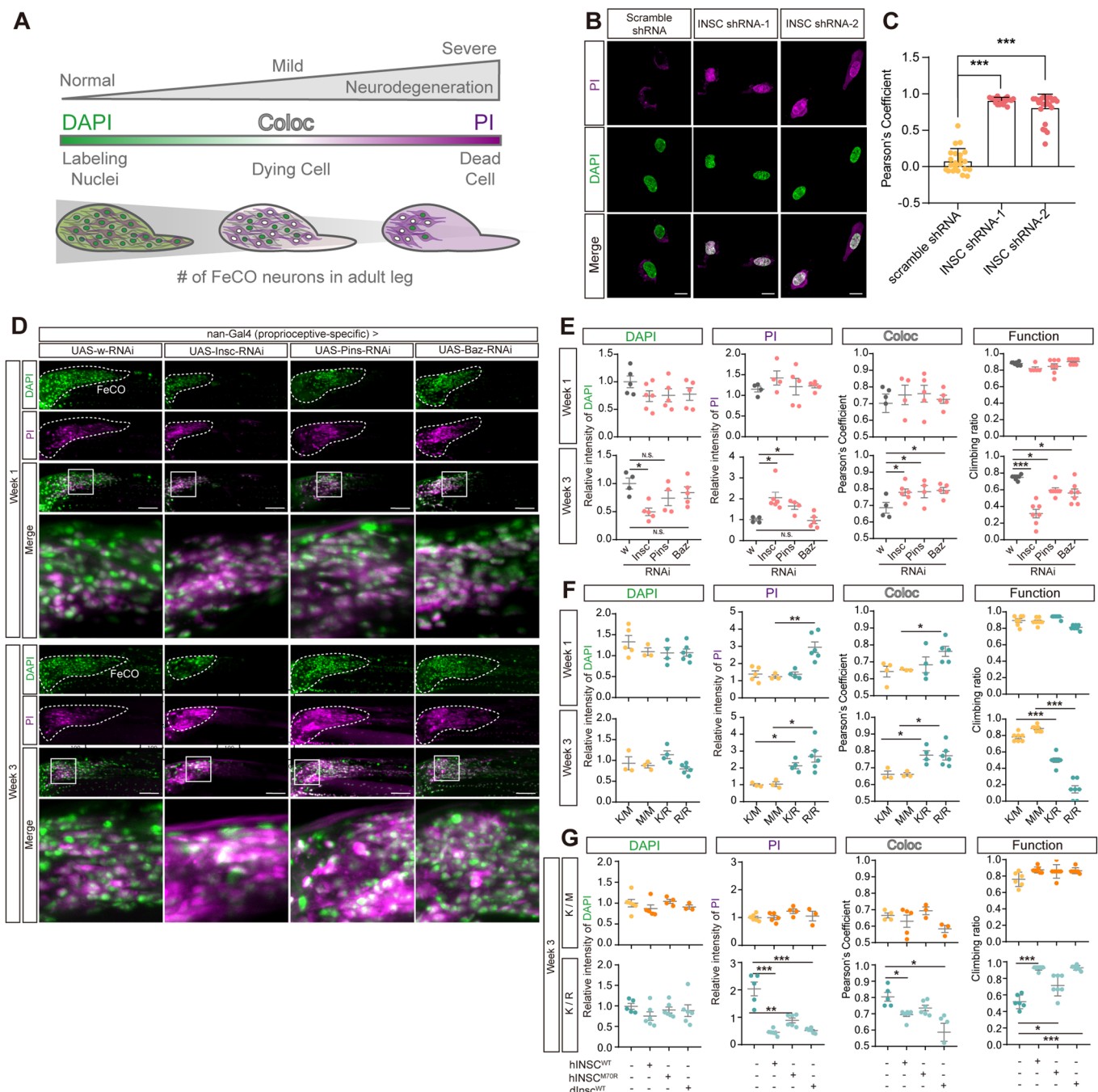

in 3-week-old flies revealed an increase in colocalized puncta of *hINSC^{M70R}-EGFP* and *Bazooka-mCherry* compared with *hINSC^{WT}* control (Fig. 4K,L). These findings highlight the mutation decreased mRNA abundance, protein level, and LGN binding.

## Dysfunction of PIL complex caused tubulin aggregation in adult proprioceptive organs

Our results likely revealed an association between the reduced association of hINSC^{M70R} with LGN and microtubule instability in the adult PNS. As hINSC competes with NuMA for LGN binding (Zhu et al, 2011) or forms stable tetramers with LGN (Culurgioni

et al, 2018) to regulate microtubule arrangement during spindle assembly in neuroblasts, we tested if reduced hINSC^{M70R} binding with LGN impaired microtubule function in adult PNS. Upon examining the scolopidium, a microtubule-rich region in adult FeCO neurons, we identified large tubulin puncta accumulated extracellularly near *dInsc*-RNAi FeCO neurons between the muscle fibers (Figs. 5A–C and EV1B–D). In contrast, we did not observe tubulin aggregates in the brain (Fig. EV1E). The tubulin aggregation was not due to overexpression of *tubulin-mCherry*, as we also observed aggregates with tubulin tracker to label the endogenous tubulin (Fig. EV1F). The hINSC^{M70R} also caused similar aggregates that were absent in the hINSC^{WT} animals. Likewise, *pins*-RNAi,

**Figure 3. Counter-correlation between necrosis and locomotor activity in PIL loss of function flies.**

(A) Schematic figure analyzing the relative intensity and colocalization of DAPI and PI staining indicated the progression of necrotic cell death in vivo. The intensity and colocalization of DAPI and PI may represent the stage of necrosis. For instance, in the healthy neurons, the signal of DAPI is largely stronger than PI; in the situation of progressive necrosis, DAPI highly colocalizes with PI; in the stage of severe neurodegeneration when most of the neuronal cell died, we can detect a weak signal of DAPI, but a relative stronger signal of PI. (B) Representative PI staining in scramble shRNA and *hINSC*-shRNA SH-SY5Y cell. Magenta is PI staining. Green is DAPI staining. Scale bar: 10 μm. (C) Pearson's coefficient of colocalization between PI and DAPI in *hINSC*-shRNA-1 and *hINSC*-shRNA-2, compared with scramble shRNA in (B); $n = 60$ cells/condition from 3 independent technical replicates. (D) Representative confocal images of *dInsc*-RNAi, *Pins*-RNAi and *Baz*-RNAi co-staining with PI (magenta) and DAPI (green) in FeCO neurons of 1- and 3-week-old flies, compared with *w*-RNAi control. The FeCO neurons are encircled by the dashed lines. Scale bar: 5 μm. (E) Quantification of the relative intensity and colocalization (coloc) of DAPI and PI in (D), and comparing the results with the functional assay. The relative intensity is normalized with *w*-RNAi; $n = 18$–28 flies/genotype from three independent fly crosses. (F) Quantification of the relative intensity and colocalization of DAPI and PI in (Fig. EV2A), and comparing the results with the functional assay. The relative intensity is normalized with $W^{m8}$ (K/K) flies; $n = 16$–28 flies/genotype from three independent fly crosses. (G) Quantification of the relative intensity and colocalization of DAPI and PI in (Fig. EV2B), and comparing the results with the functional assay. The relative intensity is normalized with *K/M* and *K/R* CRISPR-KI flies, respectively; $n = 15$–27 flies/genotype from three independent fly crosses. Data information: Error bars indicate mean ± SEM. Statistical analysis was performed using two-tailed Student's *t* test. *$P < 0.05$, **$P < 0.01$, ***$P < 0.001$. Source data are available online for this figure.

*bazooka*-RNAi, and the disease-relevant heterozygous *K/R* flies all exhibited tubulin aggregation in the proprioceptive structure, in which the severity of aggregation progressed over time (Figs. 5D,E and EV4A,B).

## Defects of PIL complex dysfunction were alleviated by microtubule-stabilizing agents

As a proof-of-concept experiment, we explored the therapeutic potential of the readily available microtubule-stabilizing agents Paclitaxel (Taxol) and Cevipabulin, both of which do not cross the blood-brain barrier (BBB) and can specifically target the PNS without affecting the brain. Because the drugs were provided by mixed into the food, we conducted a feeding assay and found that the overall food intake remained the same between different Taxol doses and fly genotypes (Appendix Fig. S8A–C), supporting that the effects were not due to differences in the amount of drug ingested. Although Taxol is known to cause peripheral neuropathy in 60% of patients receiving paclitaxel chemotherapy, we found that treatment of Taxol at low concentration in the PIL complex knockdown flies reduced the number and size of tubulin accumulations in the proprioceptive structure (Fig. 5F,G). Conversely, Colchicine, a microtubule-destabilizing drug, exerted the opposite effect and further exacerbated the aggregation of tubulin (Fig. 5F,G).

Consistently, Taxol alleviated tubulin aggregation in the disease-mimicking *dInsc^{K/R}* flies compared with *dInsc^{K/R}* flies treated with DMSO or the heterozygous *dInsc^{K/M}* flies (Fig. 6A–C). We then test whether microtubule destabilization resulting from PIL complex dysfunction is the underlying cause of PNS degeneration. We noticed that 50 μM Taxol induced tubulin aggregation in the *dInsc^{K/M}* flies (Fig. 6A–C). We assessed the dosage effect of Taxol on heterozygous *dInsc^{K/R}* flies and found that low concentrations of Taxol and Cevipabulin were neuroprotective in alleviating the necrotic status (Fig. EV5A) in FeCO neuron and restoring the locomotor activity, while high Taxol concentrations exerted toxic effects (Figs. 6D and EV4C). As both microtubule-targeting drugs alleviates defects in *dInsc^{K/R}* flies, it suggests that microtubule abnormalities contribute to the phenotypes observed in these flies. In addition, Taxol treatment also ameliorated the shortened neurite length in SH-SY5Y cells with *hINSC*-shRNA (Fig. 6E,F). We then tested the effects of Taxol treatment and genetic manipulation on microtubule stabilization. While *hINSC*-shRNA reduced acetylated

α-tubulin in SH-SY5Y cells (Fig. 6G,H), Taxol eliminated the effect of *hINSC*-shRNA on tubulin acetylation (Fig. 6I,J). We further generated the light-inducible microtubule disassembly system (MTDS) transgenic fly to genetically test tubulin acetylation. The MTDS system consists of a microtubule-severing enzyme spastin, which destabilizes microtubules upon exposure to blue light (Fig. EV5B, Liu et al, 2022). The MTDS system was expressed in the adult nervous system with *elav-GS*-Gal4, and was activated by light exposure. The blue light-exposed flies exhibited locomotor deficit compared to the red light-exposed controls (Fig. EV5C). Interestingly, we found a partial rescue of the locomotor deficit upon coexpression of dInsc, suggesting that upregulating Insc can enhance the climbing function (Fig. EV5C). Altogether, these data suggest that enhancing microtubule stability can ameliorate the neuronal necrosis and locomotive defect caused by PIL complex dysfunction.

## Discussion

We report here the identification of a large family with a novel CMT mutation affecting *INSC*, a gene known for its role in asymmetric cell division during neurodevelopment. Our research findings provide evidence that the loss-of-function of PIL complex results in progressive locomotor impairments and abnormal gait in aging flies, which closely resemble the phenotypic characteristics observed in individuals with the p.Mer70Arg mutation. Our investigations further revealed that the PIL complex is required for the control of gait and climbing by stabilizing microtubules within the adult PNS. Molecularly, haploinsufficiency of PIL complex induces tubulin aggregation, which is likely a remnant of necrotic cells that had disrupted plasma membrane integrity and underwent microtubule destabilization (see "Synopsis Image"). In addition, we demonstrated that treatment with optimal doses of Taxol, a drug known for stabilizing microtubule, can effectively alleviate the morphological and functional abnormalities associated with the loss of PIL complex function in the adult PNS in *Drosophila* carrying the same human mutation.

In this study, a combination of genome-wide linkage analysis and whole genome sequencing identified a missense genetic variant in the *INSC* gene, p.Met70Arg, which is associated with dominant CMT2 in this large pedigree. Notably, the *INSC* p.Met70Arg mutation was absent in the gnomAD and BRAVO databases

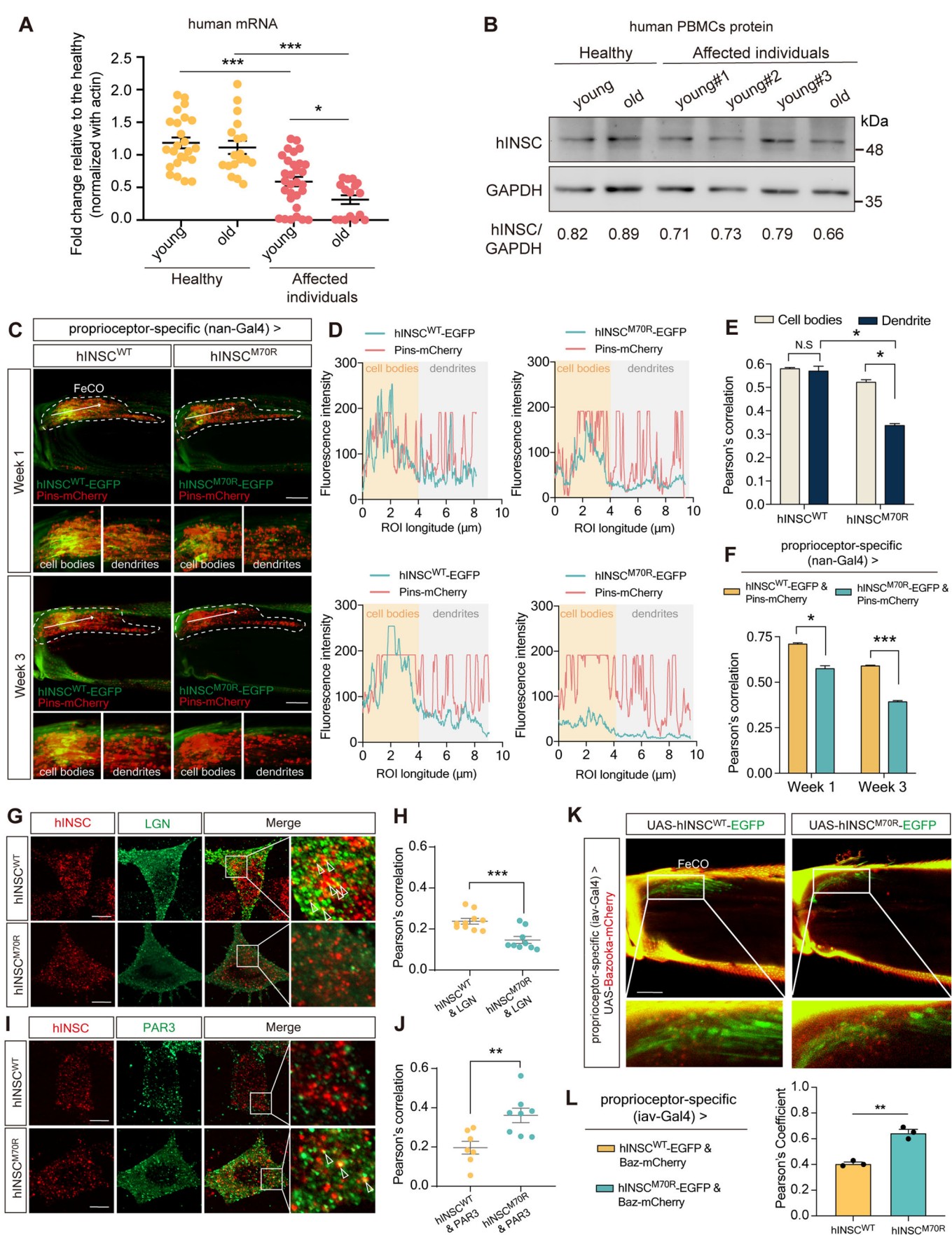

◄ **Figure 4. hINSC-M70R protein exhibited decreased levels and altered association with PIL complex.**

(A) Relative mRNA abundance of *hINSC* in the PBMCs of young (*n* = 2) and old (*n* = 1) affected individuals, compared with healthy young (*n* = 2) and old (*n* = 2) controls. For each trial, three replicates of three cDNA preparations per participant were performed. The data points collected from the same group of participants in two separate trials, each conducted 6 months apart, were pooled together. (B) Representative western blot of *hINSC* from PBMCs in young and old affected individuals, compared with healthy young and old controls. (C) Representative images of co-expressing *hINSC*^M70R^-EGFP (green) and *Pins-mCherry* (red) in 1- and 3-week-old flies under FeCO-expressing *nan*-Gal4, compared with the age-matched controls (*hINSC*^WT^-EGFP). The white arrow indicates the direction of the fluorescence intensity profile. Scale bars: 5 µm. Shown below are the magnified insets of cell bodies and dendrites in FeCO neurons, respectively. (D) Representative fluorescence intensity profiles were generated to visualize colocalization of *hINSC*^M70R^-EGFP and *Pins-mCherry* of cell bodies and dendrites in 1- and 3-week-old flies, compared with the age-matched controls (*hINSC*^WT^-EGFP). The linear region of interest (ROI) was drawn manually from left to right. (E) Pearson's coefficient of colocalization *hINSC*^WT^ and *hINSC*^M70R^ with *Pins* in (C) in cell bodies and dendrite in FeCO neurons of 1-week-old flies; *n* = 9 flies/genotype from three independent fly crosses. (F) Pearson's coefficient of colocalization of whole FeCO neurons of 1- and 3-week-old flies in (C); *n* = 8–9 flies/genotype from three independent fly crosses. (G) Immunostaining of SH-SY5Y cells transfected with FLAG-hINSC^M70R^ (red) to visualize the colocalization with MYC-LGN (green), comparing with FLAG-hINSC^WT^ (red) control. Scale bar: 10 µm. (H) Pearson's coefficient of colocalization of anti-FLAG (red) and anti-MYC (green) fluorescence in (G); *n* = 9–10 cells/condition from three independent transfection. (I) Immunostaining of SH-SY5Y cells transfected with FLAG-hINSC^M70R^ (red) to visualize the colocalization with HA-PAR3 (green), comparing with FLAG-hINSC^WT^ (red) control. Scale bar: 10 µm. (J) Pearson's coefficient of colocalization of anti-HA (green) and anti-FLAG (red) fluorescence in (I); *n* = 9–10 cells/condition from three independent transfection. (K) Representative images of co-expressing *hINSC*^M70R^-EGFP (green) and *Baz-mCherry* (red) in 3-week-old flies under *Iav*-Gal4, compared with the age-matched controls (*hINSC*^WT^-EGFP). Scale bars: 5 µm. (L) Pearson's coefficient of colocalization of EGFP (green) and mCherry (red) fluorescence in (K); *n* = 3 flies/genotype from three independent fly crosses. Data information: Error bars indicate mean ± SEM. Statistical analysis was performed using two-tailed Student's *t* test. *$P < 0.05$, **$P < 0.01$, ***$P < 0.001$. Source data are available online for this figure.

(approximately 800 thousand human exomes or genomes), as well as in 1517 ethnically matched control genomes. Interestingly, whereas CMT patients typically exhibit prominent motor symptoms, the affected individuals in this particular family showed sensory symptoms in the early disease stages, with significant motor symptoms appearing later in life, typically after the age of 50. This distinctive pattern of symptom manifestation raises further interest in the genetic basis of the condition. This intriguing parallel between the fly model and the family's condition adds weight to our conclusion that the *INSC* p.Met70Arg is indeed the genetic variant responsible for CMT in this specific family.

Modulating microtubule stability has been shown as a potential target for treating neurodegeneration (Dubey et al, 2015). Microtubule destabilization, the core phenotype observed in our study, have been implicated in the pathogenesis of neurodegenerative disorders such as Alzheimer's disease (AD), Parkinson's disease (PD), and Amyotrophic Lateral Sclerosis (ALS) (Dubey et al, 2015). AD is characterized by the accumulation of neurofibrillary tangles (NFTs) primarily composed of modified Tau protein. Tau protein acts as a stabilizer of microtubule, which is vital for axon outgrowth (Higuchi et al, 2002). Consequently, the loss of tau can lead to decreased microtubule stability, a condition that can potentially be addressed by treatment with a microtubule-stabilizing agent such as Epothilone D (Brunden et al, 2011). Epothilone is reported to enhance axonal growth in CNS injuries and cross the blood-brain barrier (Kugler et al, 2020). To explore the therapeutic potential for PNS neuropathy, we utilized Taxol, another FDA-approved microtubule-stabilizing drug which exhibits low capacity to access the brain (Fellner et al, 2002; Heimans et al, 1994). Taxol, when administrated at high dosage, has been observed to accumulate in the dorsal root ganglia (Cavaletti et al, 2000). Interestingly, we found that a relatively low concentration of Taxol increased neurite length and prevented necrotic cell death, whereas a relatively high concentration had the opposite effect. Consistent with our findings, previous studies have shown that high concentrations of Taxol may induce apoptosis-independent axonal degeneration and aberrant microtubule aggregates in both vertebrate and invertebrate organisms (Bhattacharya et al, 2012; Lipton et al, 1989). Conversely, low doses of Taxol promote axon regeneration by stimulating axonal growth (Derry et al, 1995).

Molecular investigations have demonstrated that high concentrations of Taxol induce the formation of microtubule bundles and increase polymer mass in cells, whereas lower concentrations merely suppress microtubule dynamics without affecting polymer mass (Jordan et al, 1993; Derry et al, 1995).

We have shown the expression of PIL complex in adult PNS mechanosensory neurons, particularly the FeCO neurons. At the molecular level, the regulation of tubulin acetylation involves α-tubulin acetyltransferase 1 (α-TAT1) and histone deacetylase 6 (HDAC6). α-TAT1 plays a critical role in the mechanosensation of the PNS in both *Drosophila* and mammals by modulating α-tubulin acetylation. Loss-of-function mutations in α-TAT1 have been observed to result in reduced mechanosensitivity in affected individuals (Yan et al, 2018; Morley et al, 2016). Furthermore, disrupted microtubule stability can lead to abnormal tubulin distribution. For example, the absence of Spastin, a regulator of microtubule stability, can lead to neurite swelling (Tarrade et al, 2006). Moreover, our investigation revealed a reduction in tubulin acetylation upon PIL loss of function due to haploinsufficiency of the *INSC* gene. Previous studies have established that tubulin acetylation serves as a marker of stable microtubule in neuronal cytoskeletons, and its decrease can contribute to the impairment of microtubule-mediated axonal transport, potentially leading to neurodegenerative disorders like CMT disease (D'Ydewalle et al, 2011). Importantly, this abnormality can be remedied by promoting the acetylation of α-tubulin (Reed et al, 2006).

In conclusion, our study presents compelling clinical and experimental evidence establishing a causal relationship between PIL haploinsufficiency and adult-onset hereditary neuropathy. Our findings underscore the significance of the PIL complex in regulating microtubule stability for adult proprioceptive function. Leveraging the robust genetic tools available in *Drosophila*, our model has allowed for multiple genetic manipulations and holds promise as a valuable resource for investigating therapeutic interventions. Through the integration of clinical observations and genetic models, our study elucidates the molecular mechanisms underlying necrotic neurodegeneration associated with PIL complex. Based on our findings, we propose enhancing microtubule acetylation as a potential therapeutic strategy for addressing CMT2 associated with mutations in PIL complex.

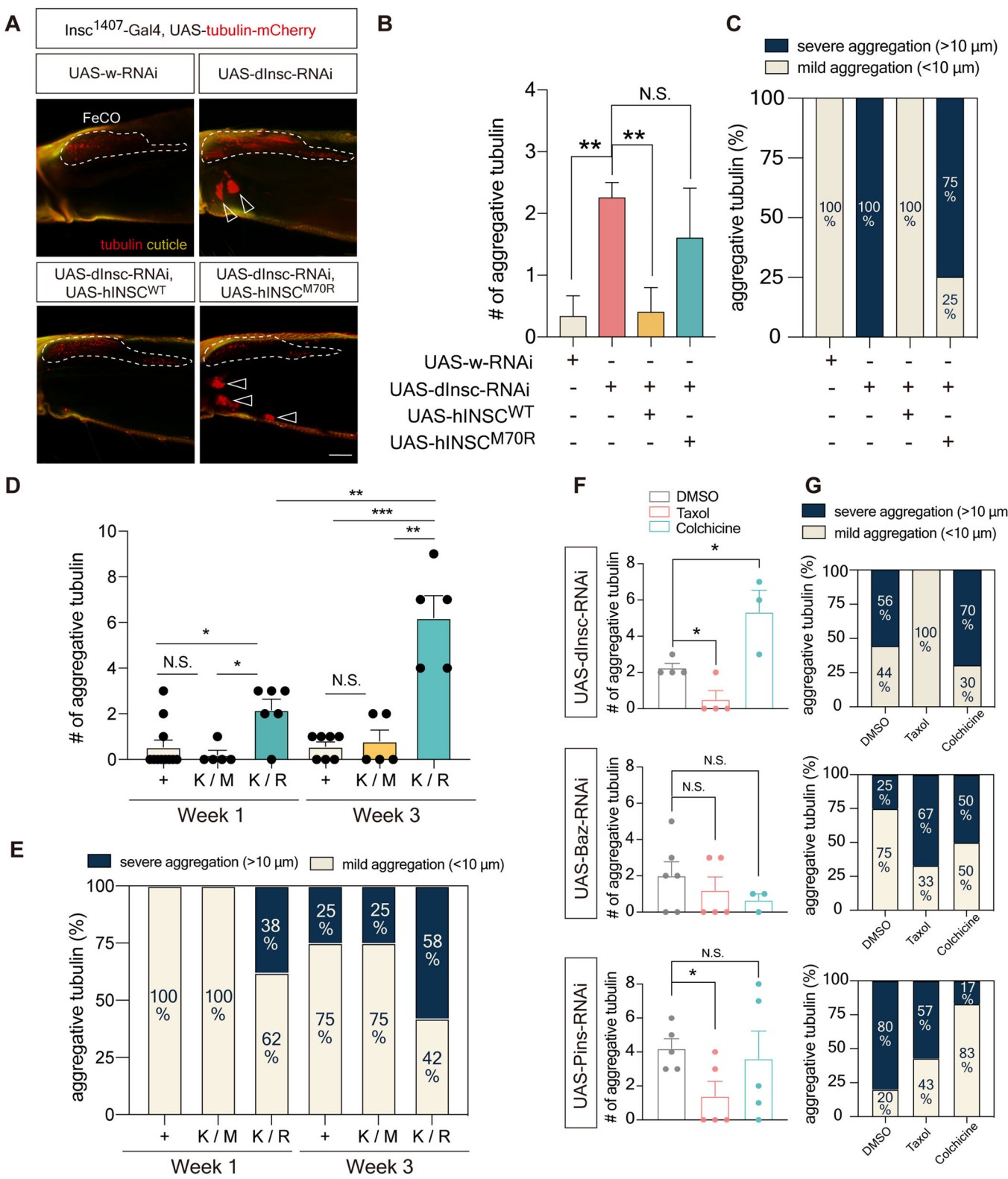

**Figure 5.  Aging flies of PIL loss-of-function exhibited tubulin aggregation in FeCO neurons.**

(A) Representative confocal images of tubulin aggregation in the femur in 3-week-old *dInsc*[1407]-Gal4>*dInsc*-RNAi flies compared with the control (*w*-RNAi) and the rescue (*hINSC*[WT] but not *hINSC*[M70R]) groups. Yellow indicates the auto-fluorescence of cuticles. The FeCO neurons are encircled by the dashed line. Arrowheads indicate the aggregative tubulin. Scale bars: 5 μm. (B) Quantification of the number of aggregative tubulins in (A); $n = 4–5$ flies/genotype from three independent fly crosses. (C) Relative abundance of aggregative tubulins of different sizes in (A); $n = 4–5$ flies/genotype from three independent fly crosses. (D) Quantification of the number of aggregative tubulins in (Fig. EV4A); $n = 5–11$ flies/genotype from three independent fly crosses. (E) Relative abundance of aggregative tubulins of different sizes in (Fig. EV4A); $n = 5–11$ flies/genotype from three independent fly crosses. (F) Quantification of the number of aggregative tubulins in (Fig. EV4B); $n = 3–6$ flies/condition. (G) Relative abundance of aggregative tubulins of different sizes in (Fig. EV4B); $n = 3–6$ flies/condition. Data information: Error bars indicate mean ± SEM. Statistical analysis was performed using two-tailed Student's *t* test. *$P < 0.05$, **$P < 0.01$, ***$P < 0.001$. Source data are available online for this figure.

# Methods

## Subjects

Twelve members of the Asian families with molecularly unassigned autosomal-dominantly CMT2 were enrolled into the study, including eight patients and four unaffected individuals (Fig. 1C). Mutations in 124 genes associated with inherited neuropathies were excluded in the proband (III-8) by targeted resequencing. Nerve conduction studies were performed by standard techniques utilizing a Medelec MS25 electromyograph (Mistro, Surrey, UK) with surface electrode stimulations and recordings. Written informed consents were obtained from all subjects and the study was approved by the Institutional Review Board of Taipei Veterans General Hospital, Taiwan (TPEVGH IRB No.:2020-02-016B).

## Genetics analyses

Whole genome sequencing was performed in the two patients (III-3 and III-8) from the CMT2 family. Illumina HiSeq2500 platform with a paired-end $2 \times 100$ bp protocol was used for sequencing. Because of the autosomal dominant inheritance, only heterozygous variants shared by the two patients were taken for further analyses. Variants present in the 1517 unrelated healthy Taiwanese genomes from the Taiwan biobank (https://taiwanview.twbiobank.org.tw/) and those that do not alter coding sequences were excluded. Sanger sequencing was then performed to determine whether any of the remaining variants completely co-segregated with CMT2 in the family (Appendix Tables S1 and S2). Mutation Taster (Schwarz et al, 2014) and Combined Annotation Dependent Depletion (CADD) (Rentzsch et al, 2019) were used to predict the pathogenicity of the identified variants. For linkage analysis, the twelve individuals in the family with CMT2 were genotyped with ~560 K markers by the Illumina Infinium assay using the Infinium CoreExome-24 v1.3 BeadChip (Illumina, San Diego, CA). Multi-point linkage was performed using the MERLIN program.

## Human whole blood collection and peripheral blood mononuclear cells (PBMCs) isolation

The human blood collection and PBMC isolation were performed (Marco-Casanova et al, 2021) with modifications. Whole blood was collected in an 8 ml BD Vacutainer® CPT™ Mononuclear Cell Preparation Tube (BD Vacutainer CPT #362761). PBMC was isolated using Ficoll-Paque density gradient centrifugation with $1500 \times g$ using a swing-out rotor for 20 min at room temperature. After centrifugation, erythrocytes and granulocytes were

sedimented to the bottom layer. In contrast, lower-density PBMCs, including lymphocytes, monocytes, and peripheral stem cells, are retained at the interface between the plasma and Ficoll-Paque. Next, a pipette was inserted directly through the plasma layer and the PBMCs layer was carefully harvested via gentle aspiration, and transferred to a fresh 15-ml tube. Ice-cold PBS was added to the harvested PBMCs and mixed well to achieve a final volume of 15 mL. The PBMCs mixture was then centrifugated at $600 \times g$ for 10 min at room temperature. After centrifugation, the supernatant was removed to leave a PBMC pellet in the base of the tube. The PBMCs cell pellet was cleaned up for the subsequent preparation of cell lysate and RNA after repeated washing and centrifugation for two times.

## Fly husbandry and stocks

*Drosophila* stocks and crosses were maintained at 25 °C on standard medium following standard fly husbandry. Information on individual fly strains and genotypes for experiments are listed in Appendix Tables S3 and S4 and can be found on FlyBase.

## Generation of point mutation CRISPR-KI flies

Fly *dInsc*[K305M] and *dInsc*[K305R] knock-in point mutation were generated using the CRISPR/Cas9 system. The gRNA for CRISPR/Cas9 target sites was designed using UCSC genome browser (https://genome.ucsc.edu/), and ligated into pBFv-U6.2 (Addgene #138400) plasmid.

To clone the donor DNA plasmid, both upstream and downstream homology arms of *dInsc* were amplified from fly genomic DNA by Phusion® High-Fidelity DNA Polymerase (New England Biolab #M0530S) under optimized conditions. The two homology arms were then cloned into pScarlessHD-DsRed (Addgene #64703) plasmid by SOEing PCR, followed by standard heat-shock transformation, colony PCR selection, and Sanger sequencing. Embryo microinjection service was provided by Wellgenetics, Taiwan, to inject the gRNA and donor DNA plasmids into fly embryos with Cas9 activity in the germ cell. Successful integration events were selected by following 3xP3-ScarlessDsRed marker in the progeny, and verified by junctional PCR. After selection, the 3xP3-ScarlessDsRed cassette in the genome of the CRISPR-modified flies was removed by PiggyBac transposase (BDSC #32174). Single-fly genomic PCR was performed to verify the CRISPR-KI alleles of *dInsc*. The CRISPR-KI flies were confirmed by Sanger sequencing and background mutations were cleaned by outcrossing to wild-type flies before conducting further morphological and functional tests.

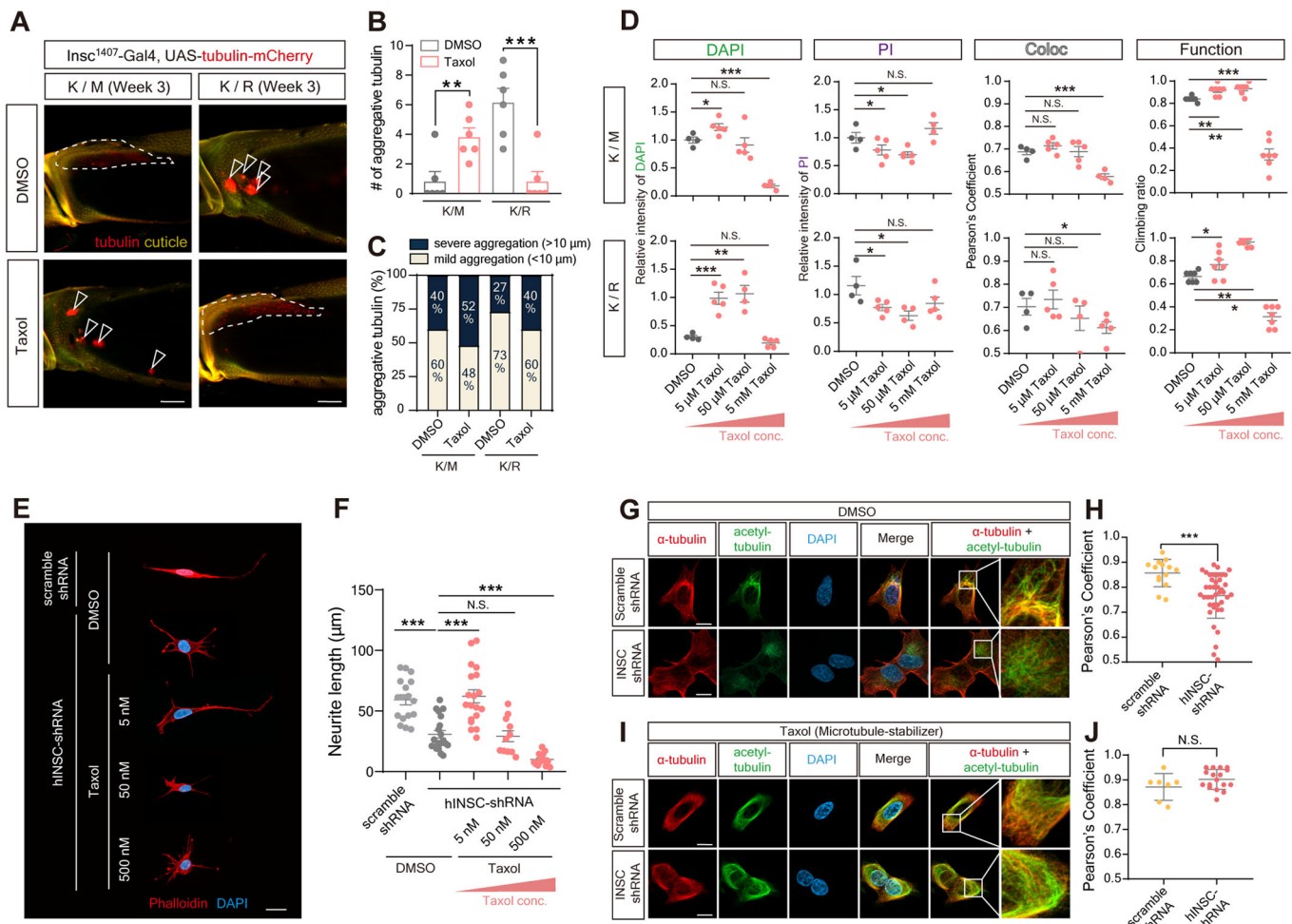

**Figure 6. Treatment of optimal concentration of Taxol rescued the morphological and functional defects in cell and aging fly.**

(A) Representative confocal images of tubulin aggregation in the femur of 3-week-old heterozygous CRISPR-KI flies, *K/M* and *K/R*, with *dInsc*[1407]-Gal4 driving *tubulin-mCherry* (Red). Flies were treated with Taxol for 7 days, compared with vehicle control (DMSO). Yellow indicates the auto-fluorescence of cuticles. The FeCO neurons are encircled by the dashed line. Arrowheads indicate the aggregative tubulin. Scale bars: 5 μm. (B) Quantification of the number of aggregative tubulins in (A); *n* = 6 flies/condition. (C) Relative abundance of aggregative tubulins of different sizes in (A); *n* = 6 flies/condition. (D) Quantification of the relative intensity and colocalization of DAPI and PI in (Fig. EV4C), and comparing the results with the functional assay. The relative intensity is normalized with the vehicle control (DMSO) group; *n* = 18–28 flies/condition. (E) Immunostaining of *hINSC*-shRNA transfected SH-SY5Y cells and treated with DMSO and Taxol to visualize the neurite length compared with scramble-shRNA control. Red is phalloidin. Blue is DAPI staining. Scale bar: 20 μm. (F) Quantification of the mean neurite length in (E); *n* = 11–20 cells/condition from three independent technical replicates. (G) Immunostaining of *hINSC*-shRNA transfected SH-SY5Y cells and treated with DMSO (a vehicle control of Taxol) to visualize the colocalization of α-tubulin (red) and acetylated tubulin (green) compared with scramble-shRNA control. Blue is DAPI staining. Scale bars: 10 μm. (H) Pearson's coefficient of colocalization of anti-α-tubulin (red) and anti-acetylated tubulin (green) fluorescence in (G); *n* = 12–41 cells/condition from three independent technical replicates. (I) Immunostaining of *hINSC*-shRNA transfected SH-SY5Y cells and treated with microtubule stabilizer Taxol to visualize the colocalization of α-tubulin (red) and acetylated tubulin (green) compared with scramble-shRNA control. Blue is DAPI staining. Scale bars: 10 μm. (J) Pearson's coefficient of colocalization of anti-α-tubulin (red) and anti-acetylated tubulin (green) fluorescence in (I); *n* = 7–17 cells/condition from three independent technical replicates. Data information: Error bars indicate mean ± SEM. Statistical analysis was performed using two-tailed Student's *t* test. *P < 0.05, **P < 0.01, ***P < 0.001. Source data are available online for this figure.

## Generation of transgenic flies

For the generation of UAS-*dInsc-EGFP*, UAS-*hINSC-WT-FLAG*, UAS-*hINSC-M70R-EGFP*, UAS-*dPins-mCherry*, UAS-*hINSC-WT-EGFP*, UAS-*hINSC-M70R-EGFP* and UAS-*MTDS*. The transgenesis, from molecular cloning, embryo injections, to screening and balancing of transformants was performed following standard protocol. For cloning, the plasmid, primers and restriction enzymes are listed in Appendix Tables S5 and S6.

## Dissection and fixation and mounting of fly tissue

The targeted tissues from larvae or adult flies were dissected according to protocols as previously described (Lien et al, 2020; Guan et al, 2018). The leg discs and wing discs were identified according to its shape and relative position as described in the 3rd instar larvae anatomy diagram (Blair, 2007). After dissection, the adult leg was fixed in 4% paraformaldehyde (PFA) for 1 day, while all other tissues were fixed for 20 min. The fixants were washed in

PBS with 0.4% Triton X-100 (0.4% PBST) three times for ten minutes. The samples were mounted in VECTASHIELD (Vector Laboratories #H-1000) for further analysis.

## Immunohistochemistry of fly tissues

For immunostaining, the fly tissues were blocked in 0.4% PBST with 1% bovine serum albumin (BSA, BioShop Canada Inc. #ALB001) at room temperature for 1 h, and then washed with 0.4% PBST for three times for 10 min. After blocking, all tissues were incubated in primary antibody at 4 °C for 1 day, washed in 0.4% PBST three times for 10 min, followed by secondary antibody incubation at 4 °C overnight. The catalog number and the working concentration of both primary and secondary antibodies were described in Appendix Table S7.

## Fluorescent microscopy and confocal imaging

All the fly tissues were scanned on a Zeiss LSM780 confocal microscope and the SH-SY5Y cell on a Zeiss ApoTome.2 microscope. The acquired images were then analyzed with Zen (Zeiss) software.

## Pharmacological induction and reduction in adult flies

For the induction of *GeneSwitch*-Gal4, 70 µl of 200 µM RU486 (mifepristone #84371-65-3) in 99% ethanol was added to the food. To bypass development, flies were incubated in normal food before eclosion. Freshly eclosed flies were transferred into the foods that contain RU486 for 7 days, and transferred to a new vial every 3 days.

For regulating microtubules, Taxol or Colchicine was added to the food, respectively. To stabilize microtubules in adult flies, the microtubule-stabilizing drug Taxol (Sigma Aldrich #T1912) in DMSO was added to the food; To destabilize microtubules, 5 µg/ml microtubule-depolymerizing agent Colchicine (Sigma Aldrich #C9754) was added to the food. Both drugs were added for 7 days, and the flies were transferred to a new vial with drug-containing food every 3 days.

## Microtubule-destabilization system (MTDS) in flies model

The original sequence was obtained from (Liu et al, 2022). The insert was codon optimized, synthesized, and cloned into XhoI/Xba sites of the multicloning site of pJFRC150-20XUAS-IVS-Flip1::PEST vector. Transgenic MTDS flies were generated via PhiC-mediated site-specific recombination and the transgene was inserted at the attP2 site of the fly genome (WellGenetics), following standard fly transgenesis. The flies were grown and maintained in a light-avoiding incubator prior to light stimulation. For light exposure, the flies were transferred into a new standard cornmeal medium in the blue (450–480 nm) and red (590–650 nm) lightbox systems, which were set up separately inside the 25 °C, 60% humidity incubator. The vials were arranged horizontally for complete exposure to the light. After 48 h of stimulation, the flies were examined for locomotor activity.

## Feeding assay in adult flies

The feeding assay was modified from protocols as previously described (Lien et al, 2020). The flies were transferred into a new

vial containing standard cornmeal food, the mixture of drug, and 37.5 mM of the food coloring agent Erioglaucine Disodium salt (Sigma Aldrich #861146). Following 6 h of feeding, the fed flies (2 female and 2 males) were homogenized with 400 µl PBS for each group. The homogenates were then centrifugated at $17,000 \times g$ for 5 min at room temperature. In total, 150 µl of supernatant was collected and put into a 96-well ELISA plate. The amount of food intake was determined by absorbance at wavelength 620 nm.

## Functional analysis of adult flies

### Climbing assay for locomotor defect
The climbing (negative geotaxis) assay examined the percentage of the flies that tapped down to the bottom of the vial to climb up and reach the 8-cm mark on the vial in 10 s (Hung et al, 2021). At the time of the experiment, groups of flies with specified genotypes were placed in a set of plastic vials. The whole experiment, performed with all the groups in ten repeats, was recorded by video for further analysis. The study was performed as an unblinded study.

### Gait analysis and quantification of walking parameters in adult flies
The gait analysis by leg tracking was performed following (Banerjee et al, 2020) with modifications. The setup consists of the 1.5-mm thick behavior chamber between two slides with the diffuser and LED light source on the top side and the high-speed camera (FLIR Blackflys® #BFS-U3-04S2M-CS) with 25-mm camera lens (Computar® #M2518-MPW2) recording on the bottom. Before the experiment, the flies were anesthetized by ice, moved to the 10 mm × 10 mm chamber individually, and kept at room temperature for more than 15 min for refreshing. After awakening, the walking flies were recording frame-by-frame with 500 fps for leg segmentation and tracking. The leg displacement plots and walking parameters were analyzed automatically by FLLIT software. All the groups were repeated at least three times.

### Propidium Iodide and DAPI dual-stain in adult FeCO neurons
The dual-staining protocol for PI and DAPI was adapted from the method described in (Venkatachalam et al, 2008; Klemm et al, 2021). To perform dual-staining in vivo, we dissected the adult fly leg and incubated it without fixation in a solution containing 5 µg/ml Propidium Iodide (PI) (Invitrogen #00-6990) and a relatively high concentration of DAPI (1:20), allowing for the specific labeling of dead cells and the nuclei of FeCO neurons, respectively. Subsequently, the staining dye was diluted in PBS and incubated at 4 °C for 1 day, followed by three 10-min washes in PBS. Finally, the samples were mounted in VECTASHIELD with DAPI (Vector Laboratories #H-1200) for further analysis.

### Tubulin tracker staining in adult FeCO neurons
A Tubulin tracker was used to visualize endogenous tubulin in vivo. The adult fly legs were dissected and incubated without fixation in PBS, then moved to staining solution containing 100X Tubulin Tracker™ Green (Oregon Green™ 488 Taxol, Bis-Acetate) (Invitrogen# T34075) at 4 °C for 1 day, followed by three 10-min washes in PBS. The samples were mounted in VECTASHIELD for further analysis.

### RNA extraction and real-time quantified PCR
Total RNA was extracted from both human PBMCs, SH-SY5Y cell, and fly samples using the Quick-RNA Miniprep Kit (ZYMO

RESEARCH #R1054), and RNA concentration was normalized prior to cDNA synthesis. Reverse transcription was performed using the MMLV high-performance cDNA synthesis kit (Epicentre #ERT12910K). qRT-PCR was conducted using the IQ2 SYBR Green Fast qPCR System Master Mix (Bio Genesis #DBU-006) using an ABI QuantStudio 5 (Applied Biosystems). Human GAPDH or beta-actin was used as the endogenous control for the human samples and RP49 was used as the endogenous control for the fly samples. All results represent the average of three independent experiments. The primers used are listed in Appendix Table S6.

### Protein lysis

For fly tissues, 5 flies were added into the Eppendorf containing 200 µl T-PER™ Tissue Protein Extraction Reagent (Thermo Scientific #78510) with protease inhibitors. The flies were homogenized on the ice. The cuticle debris were spun down, and the homogenates were transferred to a new vial for sonication and subsequent centrifugation at $17,000 \times g$ for 15 min at 4 °C. The supernatants were collected into the new Eppendorf for further analysis.

For human SH-SY5Y cells, the adherent cells were cultured to ~80% confluence on 6-well tissue culture plates. Media were aspirated and plates were kept on ice for all steps. The cell monolayer was washed gently one time with 10 ml ice-cold PBS. 200 of RIPA Lysis Buffer (Bio Basic #RB4475) with protease inhibitors was added to each plate. The cells were kept on ice for 5 min, and then scraped down and transferred to an Eppendorf. Next, the lysate was sonicated and centrifuged at $17,000 \times g$ for 15 min at 4 °C. The supernatant was collected into a new Eppendorf for further analysis.

For patients' blood PBMCs, the PBMCs cell pellet was put in an Eppendorf add of 250 µl of RIPA Lysis Buffer (Bio Basic #RB4475) with protease inhibitors. The lysate was incubated on ice for 15 min with pipetting every 5 min, then sonicated and centrifuged at $13,000 \times g$ for 5 min at 4 °C. The supernatant was collected into a new Eppendorf for further analysis.

The concentration of protein lysate was determined with Pierce™ BCA Protein Assay kit (Thermo Scientific #23225).

## Cell culture in SH-SY5Y cell

### Generation of INSC stable knockdown cell line

The pLKO.1 INSC shRNA plasmid was designed and provided by the RNA Technology Platform and Gene Manipulation Center of Academia Sinica in Taiwan and was used to generate a stable knockdown cell line of hINSC. The shRNA plasmid was purified using an Endotoxin-Free Midi-prep kit (Geneaid #PIFE25). Then, the shRNA and pGFP plasmids were co-transfected into SH-SY5Y cells (ATCC #CRL-2266) for 24 h. Prior to sorting, the cells were suspended in a sorting buffer with 1 µg/ml PI staining dye on ice. GFP-positive and PI-negative cells were sorted using a FACSAria IIIu flow cytometry sorter (BD bioscience), and single cells were sorted into 96-well plates in DMEM/F12 medium with 20% FBS. The sorted cells were incubated at 37 °C for 2 weeks. A subset of the cells was harvested to validate the knockdown efficiency using real-time quantitative PCR, while the remaining cells were cultured in the 96-well plate for subsequent experiments.

### Propidium iodide staining in SH-SY5Y cell

For PI staining in SH-SY5Y cells, the cells were seeded onto cover glasses in a 12-well plate at a density of $1 \times 10^5$ cells per well. After

seeding for 24 h, the cells were stained with 1 µg/ml Propidium Iodide (PI)(Invitrogen #00-6990) diluted in PBS for 20 min at room temperature without fixation or light exposure. The stained cells were mounted in VECTASHIELD (Vector Laboratories #H-1200) with DAPI for further imaging.

### Transfection (plasmid)

Human neuroblastoma SH-SY5Y cells were maintained at a density of $1 \times 10^5$ cells per 12-well plate in DMEM/F12 medium (Gibco #11320-033) containing 10% fetal bovine serum (FBS, Gibco #A3160502). Plasmids were transfected into cells by TransIT-X2® Dynamic Delivery System (Mirus Bio #MR-MIR6000) reagent for 24 h in the following transfection experiments.

### Immunostaining

For acetyl-tubulin immunostaining, cells were seeded on cover glasses at a density of $1 \times 10^5$ cells per 12-well plate. The transfected cells were treated with DMSO, 0.8 µg/ml Taxol (Sigma Aldrich #T1912) or 1 µg/ml Colchicine (Sigma Aldrich #C9754) for 6 h. After treatment, the cells were kept in DMEM/F12 for 24 h in 37 °C and then fixed with 4% paraformaldehyde (Electron Microscopy Sciences #15710) in PBS for 10 min at room temperature. Fixed cells were incubated in 0.1% Triton-X (0.1% PBST) 10 min for cellular permeabilization and blocked with 1% BSA in PBS for 1 h in room temperature. After blocking, the cells were stained with anti-alpha-tubulin (1:500) (Abcam #ab6160), anti-acetyl-tubulin (1:500) (Sigma Aldrich #T6793) and anti-FLAG tag (1:500) (Novus Biologicals #NBP1-06712) diluted in the blocking buffer overnight in 4 °C. On the next day, the primary antibodies were removed and cells washed for 5 min three times with 0.4% Tween-20 (0.4% PBST) and incubated in secondary antibody for 1 h at room temperature. The stained cells were mounted in the VECTA-SHIELD (Vector Laboratories #H-1200) with DAPI for further imaging. The catalog number and the working concentration of both primary and secondary antibodies are described in Appendix Table S7.

### Co-immunoprecipitation (Co-IP)

The Co-IP assays were performed following the protocol of DYKDDDDK Fab-Trap kit (ChromoTek #ffak) with modifications. The SH-SY5Y cells were transfected with the respective plasmids using the TransIT-X2® Dynamic Delivery System (Mirus Bio #MR-MIR6000). After 24 h, cells were harvested into cell pellets and resuspended in 200 µl ice-cold lysis buffer with protease inhibitor (Thermo Scientific #A32963) and 1 mM PMSF (Thermo Scientific #36978) by pipetting up and down. The lysate was plated on ice for 30 min and then centrifuged at $17,000 \times g$ for 10 min at 4 °C. The supernatant was transferred to a new Eppendorf and 300 µl ice-cold dilution buffer was added. Overall, 10% of the diluted-lysate was later used as input.

For bead equilibration, 25 µl of bead slurry was transferred into 500 µl ice-cold dilution buffer, and the beads was sedimented at $2500 \times g$ for 5 min at 4 °C. The equilibrate beads were added into the remaining diluted-lysate. The whole reaction was then rotated end-over-end for 1 h at 4 °C. Next, for the washing step, the beads were sedimented at $2500 \times g$ for 5 min at 4 °C and the supernatant was discarded. The beads were resuspended in washing buffer and sedimented at $2500 \times g$ for 5 min at 4 °C. The washing step was repeated for three times and the beads were then transferred to a

## The paper explained

### Problem

Charcot–Marie–Tooth (CMT) neuropathy, also known as hereditary motor and sensory neuropathy, is a group of inherited disorders that impact the peripheral nervous system (PNS). This condition is marked by a slow degeneration of PNS, resulting in diverse symptoms such as foot deformities and challenges with balance and coordination. However, current treatments primarily concentrate on symptom management and supportive care, with no cure identified thus far.

### Results

This study employed whole-exome sequencing to map a new locus for axonal CMT and discovered a missense mutation in the INSC gene. Through the establishment of disease-relevant CRISPR knock-in flies and the human neuronal cell line, we elucidated that the degeneration of peripheral neurons resulted from dysfunction of the PAR3/INSC/LGN complex, causing microtubule destabilization in the adult PNS. Importantly, the application of pharmacological agents targeting microtubule stabilization effectively mitigated both proprioception defects and necrosis in peripheral neurons.

### Impact

This study provided new insights into the microtubule-stabilizing function of PAR3/INSC/LGN in the adult PNS. These findings provide a foundation for the development of potential new therapeutic approaches for axonal CMT.

new tube. For elution step, the beads were resuspended in 50 μl 4× Laemmli sample buffer (Bio-Rad #1610747) and then boiled for 5 min at 95 °C. The beads were then sedimented at $2500 \times g$ for 5 min at 4 °C for further western blotting.

### Western blots

Protein extracts from flies, SH-SY5Y cell, and human PBMCs were separated by SDS-PAGE and transferred to polyvinylidene difluoride membranes (PVDF, Millipore #IPVH00010, pore size: 0.45 mm) as per the manufacturer's instructions (Bio-Rad). PVDF membranes were incubated with 5% bovine serum albumin (BSA, BioShop Canada Inc. #ALB001) in 0.1% TBST for 1 h at room temperature, washed three times with 0.1% TBST for 10 min, and incubated with primary antibodies in 5% BSA at 4 °C overnight. Membranes were washed three times with 0.1% TBST for 10 min before incubation with secondary antibodies in 0.1% TBST for 1 h at room temperature. Blots were then washed with 0.1% TBST three times for 10 min, developed with ECL reagents (Thermo #34580; Millipore #WBKLS0500), and captured by the BioSpectrumTM 600 Imaging System (UVP Ltd). The quantification of western blotting results was processed in ImageJ software (NIH, USA).

### Neurite length quantification

For the measurement of neurite length, the cells were seeded on cover glasses at a density of $1 \times 10^5$ cells per 12-well plate. Subsequently, transfection and immunostaining of the cell were performed following standard procedures. The cell edge was determined with phalloidin (Invitrogen #A12380) staining.

### Colocalization analysis

For colocalization analysis, cells and tissues were automatically processed in 3D, as the deconvolved two-channel 3D image was exported to the ImarisColoc plugin of the microscopy image analysis software Imaris (Oxford Instruments) for quantitative analysis. The levels of colocalization were output as Pearson's coefficients for further statistical analysis.

### Quantification and statistical analysis

The aggregative tubulin in the femur was quantified and analyzed by Imaris software in three-dimensional images.

This study was performed as an unblinded study. Quantitative data were analyzed using a two-tailed unpaired Student's $t$ test and the graphs were generated using GraphPad Prism 8. All experiments were performed with at least three independent biological replicates and expressed as the means ±SEM. $P$ values of less than 0.05 were considered statistically significant. *$P < 0.05$, **$P < 0.01$, and ***$P < 0.001$. All of the statistical details of experiments can be found in the figure legends. $n \geq 3$ for each experiment with $\geq 3$ independent experiments.

### Study approval

The study was conducted according to the Declaration of Helsinki Principles and the Department of Health and Human Services Belmont Report. Written informed consents were obtained from all subjects and the study was approved by the Institutional Review Board of Taipei Veterans General Hospital, Taiwan (TPEVGH IRB No.:2020-02-016B).

## For more information

See Taiwan Biobank database: https://taiwanview.twbiobank.org.tw/; BRAVO database: https://legacy.bravo.sph.umich.edu/freeze8/hg38/.

## Data availability

This study includes no data deposited in external repositories. Due to consent agreement restrictions, patient whole-exome sequencing data could not be made freely available and was therefore not deposited in a public database.

## Peer review information

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

## Acknowledgements

The authors are grateful for the patients who consented to being a part of this study. The authors thank all members of the Chan and Lee labs for critical comments on this manuscript. The authors thank the Bloomington Drosophila Stock Center, Vienna Drosophila RNAi Center, and Developmental Studies Hybridoma Bank for reagents, WellGenetics Inc. for the generation of transgenic flies, and the Biomedical Resource Core and the imaging core at the First Core Labs in National Taiwan University College of Medicine for technical support in molecular cloning, image acquisition, and analysis. This work was supported by grants from the National Science and Technology Council of Taiwan (111-2320-B-002-049-MY3 and 112-2314-B-002 -016 to C-CC; 109-2314-B-075-044-MY3 and 112-2314-B-075 -034 -MY3 to Y-CLee; 112-2636-B-007-008 and 112-2628-B-007-004 to Y-CLin), National Health Research Institutes (EX112-11228NI and EX113-11228NI to C-CC), National Taiwan University (112L895403 and 113L893103 to C-CC), and Brain Research Center, National Yang Ming Chiao Tung University from The Featured Areas Research Center Program within the framework of the Higher Education Sprout Project by the Ministry of Education (MOE) in Taiwan.

## Author contributions

**Jui-Yu Yeh**: Conceptualization; Data curation; Formal analysis; Investigation; Methodology; Writing—original draft; Writing—review and editing. **Hua-Chuan Chao**: Data curation; Investigation; Methodology. **Cheng-Li Hong**: Data curation; Investigation; Methodology. **Yu-Chien Hung**: Data curation; Investigation; Methodology. **Fei-Yang Tzou**: Data curation; Investigation; Methodology. **Cheng-Tsung Hsiao**: Data curation; Investigation; Methodology. **Jeng-Lin Li**: Data curation; Investigation; Methodology. **Wen-Jie Chen**: Data curation; Investigation; Methodology. **Cheng-Ta Chou**: Data curation; Investigation; Methodology. **Yu-Shuen Tsai**: Data curation; Investigation; Methodology. **Yi-Chu Liao**: Resources; Investigation; Methodology. **Yu-Chun Lin**: Resources; Investigation; Methodology. **Suewei Lin**: Resources; Investigation; Methodology. **Shu-Yi Huang**: Investigation; Writing—original draft; Writing—review and editing. **Marina Kennerson**: Data curation; Investigation; Writing—original draft. **Yi-Chung Lee**: Conceptualization; Resources; Supervision; Funding acquisition; Validation; Investigation; Methodology; Writing—original draft; Writing—review and editing. **Chih-Chiang Chan**: Conceptualization; Resources; Supervision; Funding acquisition; Validation; Investigation; Methodology; Writing—original draft; Writing—review and editing.

## Disclosure and competing interests statement

The authors declare no competing interests.

# Expanded View Figures

**Figure EV1.  Expression of *dInsc* in FeCO neuron and its role in inducing tubulin aggregation.**

(A) The patterns of UAS-*mCD8-GFP* driven by *dInsc*[1407]-Gal4 (left, same image in Fig. 2E) and *Iav*-Gal4 (right) in the leg. Green indicates *mCD8-GFP* and magenta indicates auto-fluorescence of the cuticle. Scale bars: 50 μm. (B) Representative confocal images of tubulin aggregation in the femur of 3-week-old *dInsc*[1407]-Gal4>UAS-*dInsc*-RNAi flies compared with UAS-*LacZ* control. Red indicates *tubulin-mCherry*. Scale bars: 50 μm. (C, D) Three-dimensional imaging of the femur of (D) *dInsc*[1407]-Gal4>*dInsc*-RNAi compared with (C) control flies. The intercellular mCherry fluorescence shows tubulin aggregation between muscle fibers. Red labeled tubulin, cyan labeled phalloidin (muscle fibers), and green indicate the auto-fluorescence of the cuticle. Arrowheads indicate the aggregative tubulin. Scale bars: 5 μm. (E) Representative confocal images of the adult brain of 3-week-old *dInsc*[1407]-Gal4>UAS-*dInsc*-RNAi compared with the UAS-*LacZ* control flies. Green indicates mCherry fluorescence. Magenta indicates anti-DLG. Scale bars: 50 μm. (F) Representative confocal images of tubulin aggregation in the FeCO neuron and the femur of 3-week-old *dInsc*[1407]-Gal4>*dInsc*-RNAi flies, compared with the control (UAS-*lacZ*) and the rescue (*hINSC*[WT] and *hINSC*[M70R]) groups. Red indicates tubulin tracker signals. Green indicates the auto-fluorescence of cuticles. The FeCO neurons are encircled by the dashed line. Scale bars: 5 μm.

                                                     

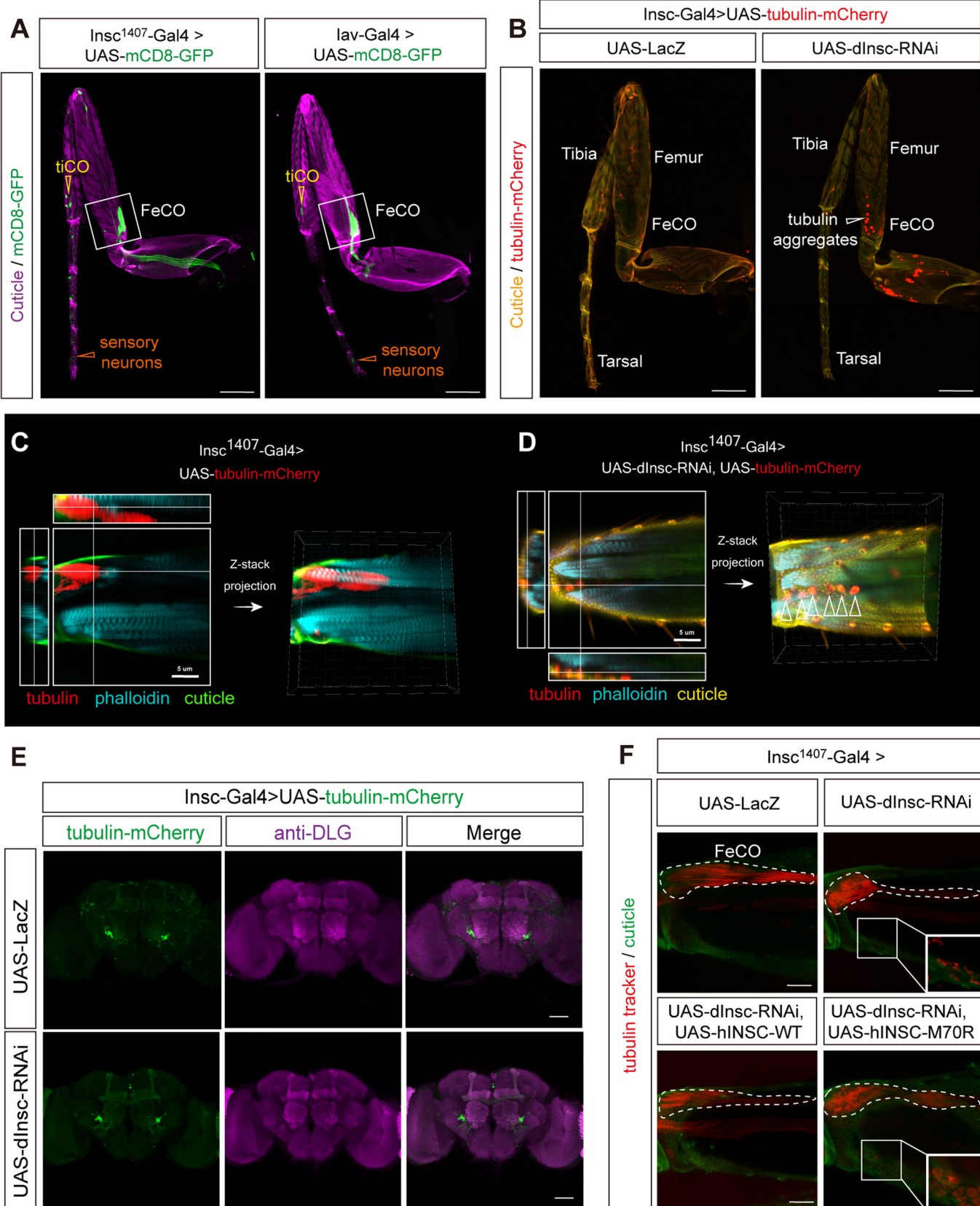

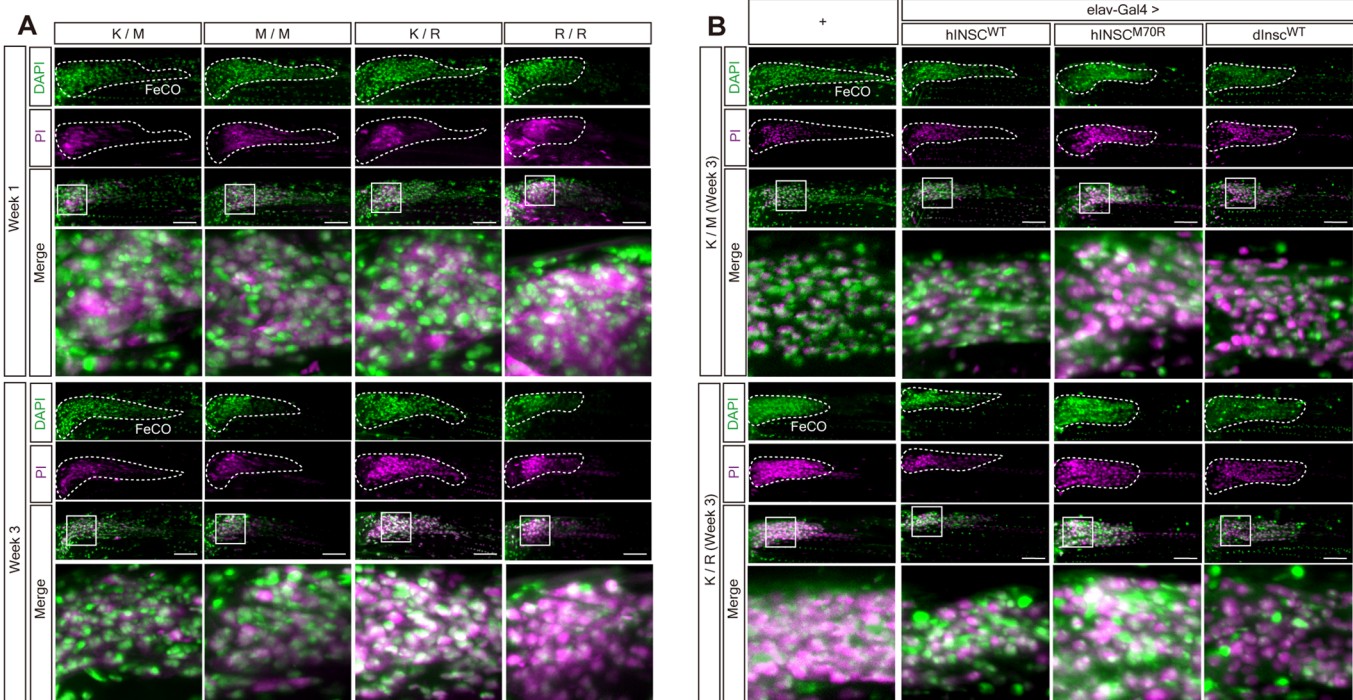

**Figure EV2. Representative confocal images of DAPI and PI co-staining in K305M and K305R CRISPR-KI flies.**

(A) Representative confocal images of FeCO neurons of 1- and 3-week-old $dInsc^{+/K305R}$ (K/R) and $dInsc^{K305R/K305R}$ (R/R) flies, co-stained with PI (magenta) and DAPI (green), compared with $dInsc^{+/K305M}$ (K/M) and $dInsc^{K305M/K305M}$ (M/M) CRISPR-KI flies. The FeCO neurons are encircled by the dashed line. Scale bar: 5 μm. (B) Representative confocal images of FeCO neurons of K/M and K/R flies overexpressing $hINSC^{WT}$, $hINSC^{M70R}$ and $dInsc^{WT}$ under the control of pan-neuronal elav-Gal4, co-stained with PI (magenta) and DAPI (green), compared with K/M and K/R CRISPR-KI flies, respectively. All flies are aged to 3 weeks. The FeCO neurons are encircled by the dashed line. Scale bar: 5 μm.

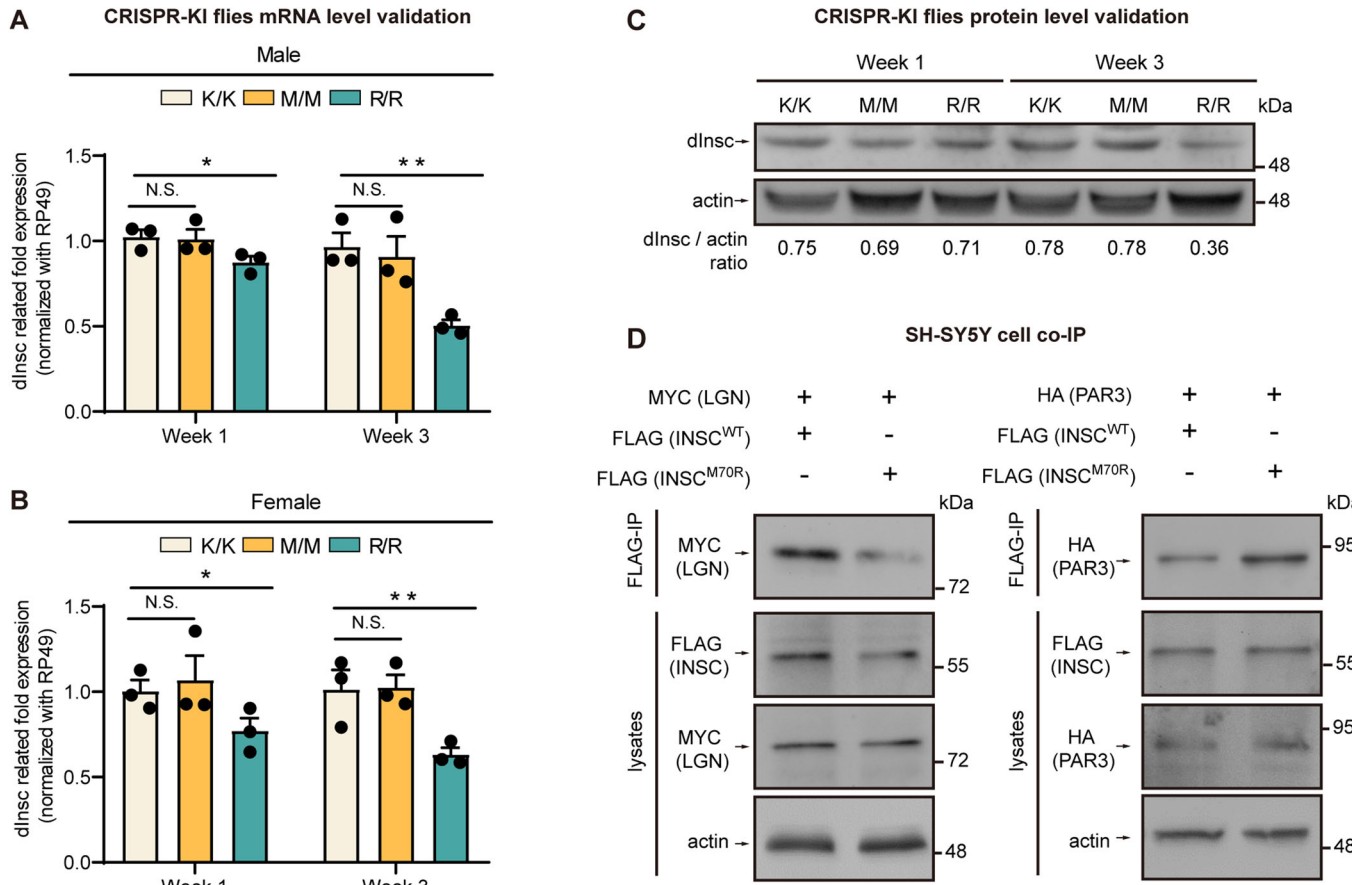

**Figure EV3.   Both mRNA and protein level of R/R decrease in aging flies induced by PIL complex dysregulation.**

(A) Relative mRNA abundance in whole fly extracts of *M/M* and *R/R* male flies with its corresponding *K/K* control in week 1 and week 3; *n* = 12 flies/genotype from 3 independent technical replicates. (B) Relative mRNA abundance in whole fly extracts of *M/M* and *R/R* female flies with its corresponding *K/K* control in week 1 and week 3; *n* = 12 flies/genotype from three independent technical replicates. (C) Representative western blotting of whole fly extracts from *M/M* and *R/R* flies with its corresponding *K/K* control in week 1 and week 3; *n* = 5 flies/genotype. (D) Co-immunoprecipitation to examine the association of FLAG-hINSC (WT and M70R) with MYC-LGN (left) or HA-PAR3 (right) in SH-SY5Y cells.

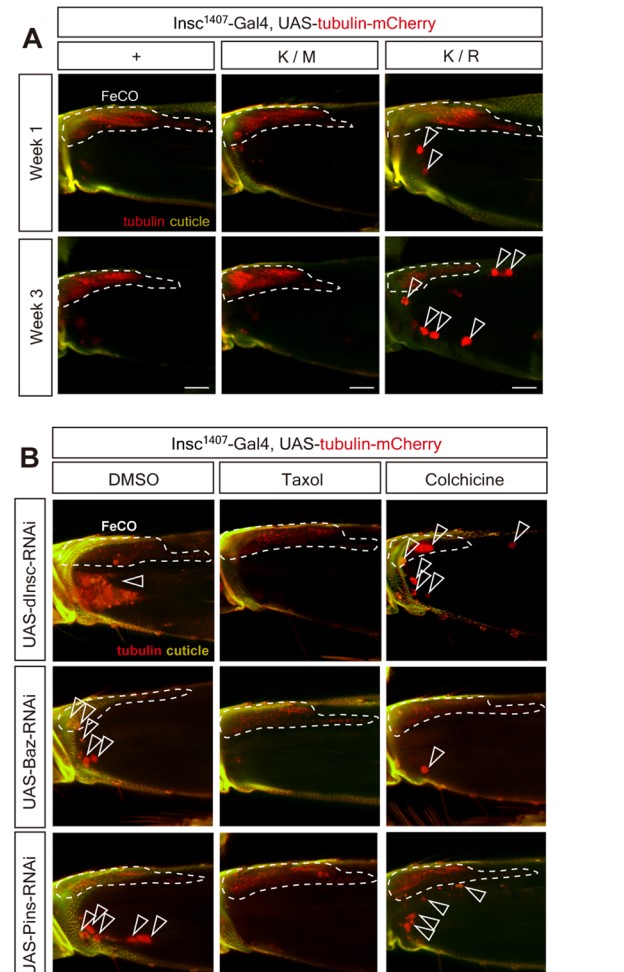

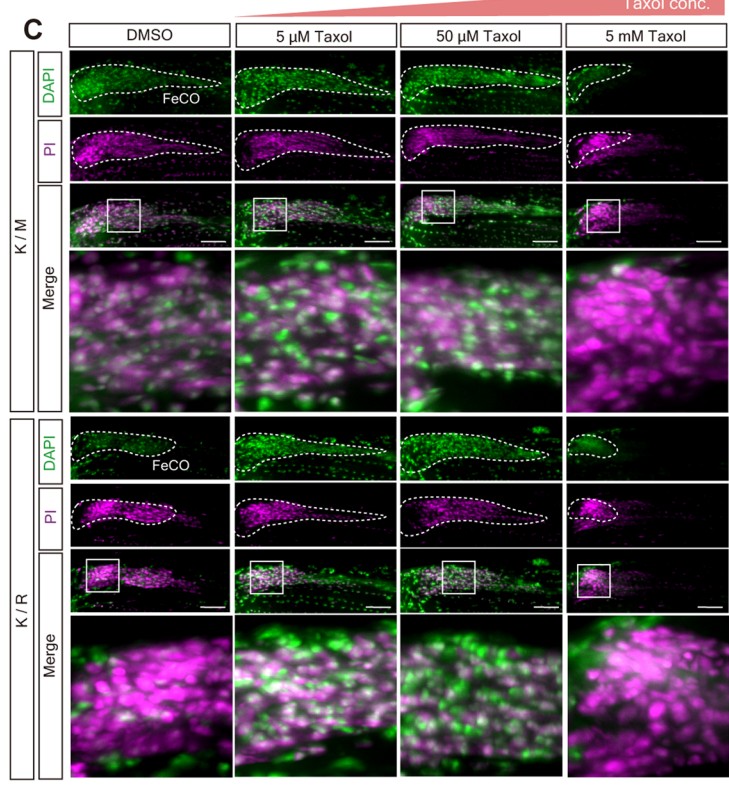

**Figure EV4. Taxol and Colchicine exert opposite effects on tubulin accumulation.**

(A) Representative confocal images of tubulin aggregation in *K/M* and *K/R* of 1- and 3-week-old CRISPR-KI flies with *dInsc[1407]*-Gal4>UAS-*tubulin-mCherry*. Arrowheads indicate the aggregative tubulin. Scale bars: 5 μm. (B) Representative confocal images of tubulin aggregation (Red indicates *tubulin-mCherry*) in the femur of 3-week-old *dInsc*-RNAi, *Baz*-RNAi and *Pins*-RNAi flies under *dInsc[1407]*-Gal4. Flies were treated with DMSO (vehicle control), Taxol (microtubule-stabilizing agent) and Colchicine (microtubule-destabilizing agent). Yellow indicates the auto-fluorescence of cuticles. The FeCO neurons are encircled by the dashed line. Arrowheads indicate the aggregative tubulin. Scale bars: 5 μm. (C) Representative confocal images of FeCO neurons of *K/M* and *K/R* CRISPR-KI flies treated with three different concentrations (5 μM, 50 μM, 5 mM) of Taxol for 7 days, co-stained with PI (magenta) and DAPI (green), and compared with vehicle control (DMSO). The FeCO neurons are encircled by the dashed line. Scale bars: 5 μm.

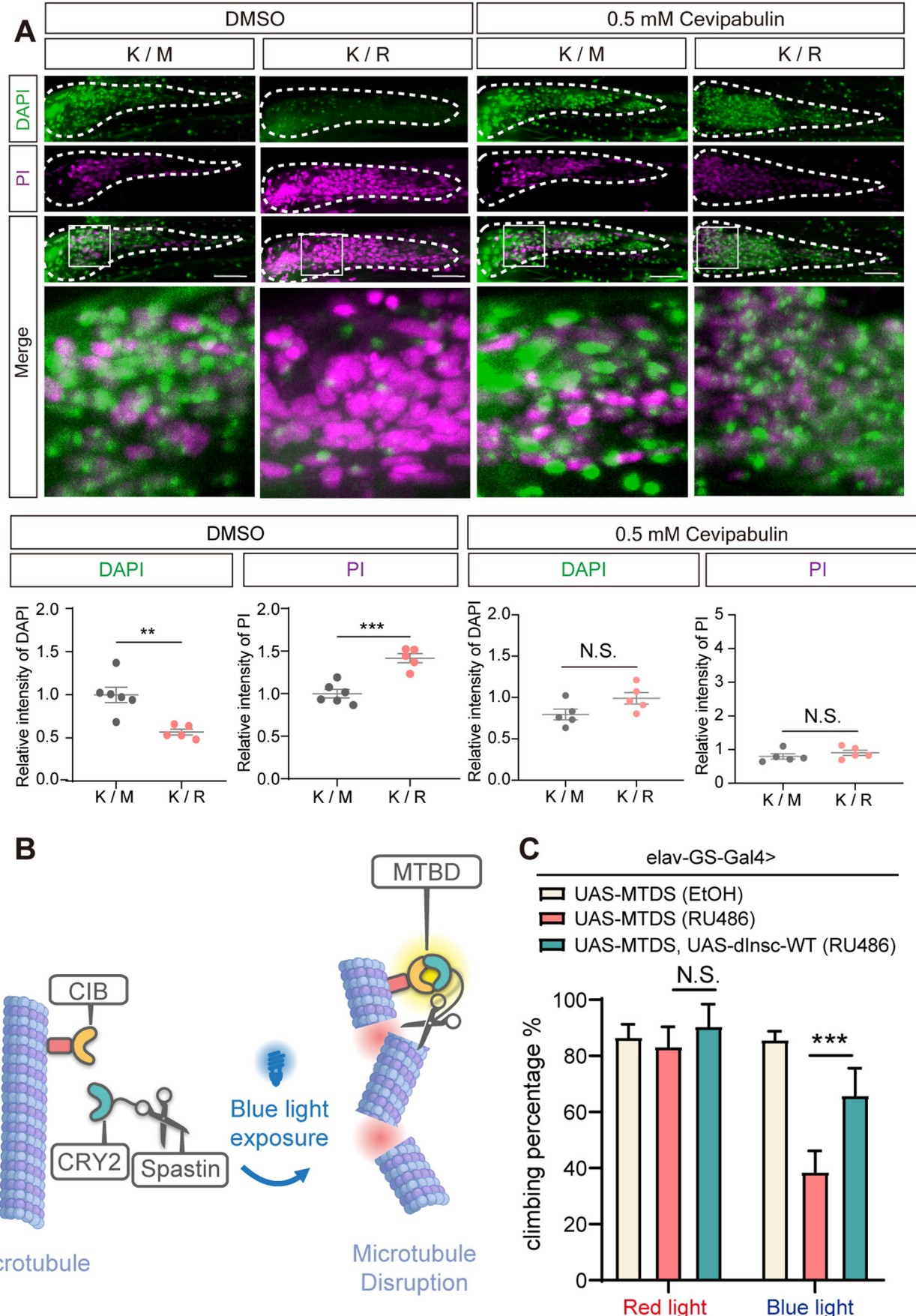

◀

**Figure EV5.  The destabilization of microtubules caused by M70R mutation can be rescued with both microtubule-stabilizing agents and genetic manipulation.**

(A) Representative confocal images of FeCO neurons of *K/M* and *K/R* 3-week flies treated with 0.5 mM of microtubule stabilizer Cevipabulin for 7 days, co-stained with PI (magenta) and DAPI (green), compared with vehicle control (DMSO). The FeCO neurons are encircled by the dashed line. Scale bars: 5 μm. Quantifications are shown in the lower panels. (B) The schematic of light-inducible microtubule disassembly system (MTDS) in a fly model. The dimerization of CRY2 and CIB can be induced by blue light. The CRY2 is fused with a microtubule-severing enzyme Spastin, and CIB is fused with microtubule-binding domain (MTBD). Dimerization upon blue light stimuli induces accumulation of Spastins on microtubules, which in turn induces disassembly of microtubules in MTDS-expressing cells. (C) Quantification of the climbing activity of Week 1 flies of UAS-*MTDS* (EtOH), UAS-*MTDS* (RU486), and UAS-*MTDS*, UAS-*dInsc-WT* (RU486) under the control of RU486-inducible pan-neuronal driver (*elav-GS-*Gal4) upon 48 h of blue light exposure, comparing with the conditional control (red light); $n = 30$ flies/genotypes from 3 independent fly crosses.

