## [Peer Review File · EMBO Molecular Medicine]

A missense mutation in human INSC causes peripheral neuropathy

Jui-Yu Yeh, Hua-Chuan Chao, Cheng-Li Hong, Yu-Chien Hung, Fei-Yang Tzou, Cheng-Tsung Hsiao, Jeng-Lin Li, Wen-Jie Chen, Cheng-Ta Chou, Yu-Shuen Tsai, Yi-Chu Liao, Yu-Chun Lin, Suewei Lin, Shu-Yi Huang, Marina Kennerson, Yi-Chung Lee, and Chih-Chiang Chan

Corresponding authors: Chih-Chiang Chan (chancc1@ntu.edu.tw) , Yi-Chung Lee (ycli@vghtpe.gov.tw)

Review Timeline:

Submission Date:	3rd Oct 23
Editorial Decision:	17th Nov 23
Revision Received:	6th Feb 24
Editorial Decision:	4th Mar 24
Revision Received:	8th Mar 24
Accepted:	15th Mar 24

Editor: Zeljko Durdevic

Transaction Report:

17th Nov 2023

Dear Dr. Chan,

Thank you for the submission of your manuscript to EMBO Molecular Medicine, and please accept my apologies for the delay in getting back to you, which is due to the fact that one referee needed more time to complete his/her review. We have now received feedback from the three reviewers who agreed to evaluate your manuscript. All three referees recognize potential interest of the study but also raise serious and partially overlapping concerns that should be addressed in a major revision. Providing experimental evidence to support proposed mechanism would be essential for further consideration of your manuscript. Further, please consider changing the title of the manuscript to better reflect the main findings but also the scope of our journal, e.g. "A missense mutation in human INSC causes peripheral neuropathy" or "A missense mutation in human INSC causes axonal Charcot-Marie-Tooth disease". If you would like to discuss further the points raised by the referees, I am available to do so via email or video. Let me know if you are interested in this option.

We would welcome the submission of a revised version within three months for further consideration. Please let us know if you require longer to complete the revision.

I look forward to receiving your revised manuscript.

Yours sincerely,

Zeljko Durdevic

We require:

2) Individual production quality figure files as .eps, .tif, .jpg (one file per figure). For guidance, download the 'Figure Guide PDF': (<https://www.embopress.org/page/journal/17574684/authorguide#figureformat>).

3) A .docx formatted letter INCLUDING the reviewers' reports and your detailed point-by-point responses to their comments. As part of the EMBO Press transparent editorial process, the point-by-point response is part of the Review Process File (RPF), which will be published alongside your paper.

4) A complete author checklist, which you can download from our author guidelines (<https://www.embopress.org/page/journal/17574684/authorguide#submissionofrevisions>). Please insert information in the checklist that is also reflected in the manuscript. The completed author checklist will also be part of the RPF.

6) It is mandatory to include a 'Data Availability' section after the Materials and Methods. Before submitting your revision, primary datasets produced in this study need to be deposited in an appropriate public database, and the accession numbers and database listed under 'Data Availability'. Please remember to provide a reviewer password if the datasets are not yet public (see <https://www.embopress.org/page/journal/17574684/authorguide#dataavailability>).

13) Author contributions: You will be asked to provide CRediT (Contributor Role Taxonomy) terms in the submission system.

These replace a narrative author contribution section in the manuscript.

14) A Conflict of Interest statement should be provided in the main text.

Please also suggest a striking image or visual abstract to illustrate your article as a PNG file 550 px wide x 300-800 px high.

***** Reviewer's comments *****

Referee #1 (Remarks for Author):

The manuscript by Yeh describes the important role of hINSC in the adult PNS function. The authors first identified a missense mutation Met70Arg in the hINSC protein from CMT2 patients. To understand the relevance of this mutation, they leveraged the fly model to understand whether and how hINSC and its fly homolog Insc regulate PNS functions in fly. Interestingly, knockdown of fly Insc caused locomotion/gait defects and adult-onset necrosis in aging fly, which can be rescued by overexpression hINSC but not hINSC-Met70Arg mutant form. Like Insc, knockdown of Insc-binding partners, such as Pins and Baz, resembled insc depletion. hINSC-M70R was expressed at a lower level and less stable in flies. Mechanistically, dysfunction of fly Insc caused microtubulin aggregation near adult proprioceptive organs, which is caused by destabilized microtubules. Treatment with low-dose Taxol, a microtubule destabilizer, can alleviate the microtubule aggregation and locomotive defects. Overall, this is an important study which establishes a causal relationship between hInsc mutation and CMT2 neuropathy. Moreover, it also reveals a mechanism whereby microtubule instability contributes to the disease pathogenesis. Thus, I recommend its publication after some minor revisions.

1) Figure EV1A-B. Are those Insc-positive cells in leg discs neurons?

2) Figure 2I. did *lav-Gal4* drive the expression in the same neurons in the adult legs as *Insc1407-Gal4* (Figure 2E)?

3) Figure 4D-E. The authors show the defects of the protein colocalization between hInsc-M70R and LGN/PAR3. Did the authors also test the defects in protein-protein interactions in co-IP assays? If the authors have the co-IP results, they can enhance the impact. But the results are not required for publication.

4) Figure 5A. the microtubule aggregation phenotype is interesting. Can the author discuss why the microtubules aggregate extracellularly near neurons upon *dInsc* or *Pins* knockdown? what happened to the endogenous microtubules when anti- α -tubulin antibody is used to detect them? Can they observe any aggregates for endogenous microtubule too?

5) Figure 5A. were the microtubule aggregates observed in other leg sites of *dInsc* RNAi flies? For example, near tibiotarsal chordotonal organ. Since *dInsc* is also expressed in CNS, any microtubule aggregated in CNS neurons?

Referee #2 (Comments on Novelty/Model System for Author):

The manuscript by Yeh et al reported the identification of a novel mutation in INSC gene in one family with CMT2 related neuropathy. The authors used *Drosophila* models to discern the causal relationship between the variant pMet70Arg (in heterozygous) and the neuropathy phenotypes. The authors showed that the loss of function of *dInsc* in flies causes locomotor and gait phenotypes in flies. They further examined the cellular phenotypes and colocalization among LGN, Par3 and Insc in fly leg tissue and observed increased necrosis, microtubule instability, and disruption of the PIL complex. The discovery of Insc variant is intriguing, and some of the experimental designs are well done. However, the main conclusion on a series of causal relationships, --INSC_M70R mutation leads to PIL complex dysfunction, then leads to microtubule instability, then leads to necrosis and neuropathy, were not well supported by the data. In particular, several concerns regarding the genetic and cellular analyses should be addressed to fully support the conclusion.

Referee #2 (Remarks for Author):

The manuscript by Yeh et al reported the identification of a novel mutation in INSC gene in one family with CMT2 related neuropathy. The authors used Drosophila models to discern the causal relationship between the variant pMet70Arg (in heterozygous) and the neuropathy phenotypes. The authors showed that the loss of function of dInsc in flies causes locomotor and gait phenotypes in flies. They further examined the cellular phenotypes and colocalization among LGN, Par3 and Insc in fly leg tissue and observed increased necrosis, microtubule instability, and disruption of the PIL complex. The discovery of Insc variant is intriguing, and some of the experimental designs are well done. However, the main conclusion on a series of causal relationships, --INSC_M70R mutation leads to PIL complex dysfunction, then leads to microtubule instability, then leads to necrosis and neuropathy, were not well supported by the data. In particular, several concerns regarding the genetic and cellular analyses should be addressed to fully support the conclusion.

1. Genetics, haploinsufficiency: the affected individuals in the family carry the variant pMet70Arg in heterozygous. It is critical to assess how a point mutation in heterozygous causes disease. There are two possibilities: loss of function as haploinsufficiency, or gain of function as dominant effects. The evidence supporting a loss of function conclusion is weak. The possibility of gain of function was not ruled out with the presented data.

a. Is the pMet70Arg allele expressed in the same level (mRNA and protein) as the wildtype allele in patients?

b. In Figure 2B, the author claims that the expression of human wild type INSC (hINSCWT) not the mutant (hINSCM70R) can rescue the impaired locomotion phenotype caused by heterozygous loss of function of dInsc. However, the author did not show the protein expression levels of each UAS-hINSC transgenic line.

c. The dilution effects of the UAS/GAL4 could affect the behavior results in UAS-dInsc-RNAi (red patterned bar) vs UAS-dInsc-RNAi, UAS-hINSCM70R (teal patterned bar), as well as the UAS-hINSCWT (orange bar) vs UAS-dInsc-RNAi, UAS- UAS-hINSCWT (orange patterned bar).

d. In Figure 2A, was the significance only compared with the UAS-w-RNAi (RU486) control? In Figure legend 2B it says that it is showing climbing activity for 1- and 3-week-old flies, but figure only shows 3-week-old data.

e. Is the phenotype of Baz KD shown in Fig. 2A significant compared to the first or second control group? Is the KD efficiency examined? The role of Baz in this condition might be more complicated than that proposed by the authors if authors believe the KD efficiency is sufficient.

2. Homology between fly and human: homology analysis between the human and drosophila INSC is lacking. Specifically, it is unclear how the conservation between residue K305 in drosophila and human M70R is established. The large difference in residue location is also rather unusual.

3. Complex formation and colocalization analysis: it was concluded that the LGN, Par3 and Insc form a complex in Drosophila neurons. The conclusion is mainly based on colocalization analysis in cultured cells and in fly leg prep. Biochemical evidence would be required for any statement of complex formation or protein-protein interaction. Furthermore, there are several issues with the presented colocalization analysis.

a. In Fig. 4D-4G, the cell morphology of hINSC_M70R is significantly different from hINSC-WT (Fig. 4F). Such difference precludes a reliable analysis of colocalization.

b. In the colocalization study, it was shown that hINSC_M70R resulted in a decrease in LGN and increase in PAR3 colocalization. The significance and relevance of this observation are unclear.

c. In Fig. 4H, the Baz-mCherry signal in the INSC M70R group is much weaker than that in the INSC WT group. Is this image representative? On the other hand, the INSC M70R-EGFP signal is much stronger than the WT-EGFP signal, which is opposite to the observation in Fig. 4B. Authors should explain the inconsistency.

d. It is unclear how Fig 4A is generated. Were healthy old individuals to test for panel A? Was only one healthy young control analyzed?

4. Necrosis and microtubule aggregation, cause and effect: the authors concluded a series of causal relationships, INSC M70R mutation leads to PIL complex dysfunction, then leads to microtubule instability, then leads to necrosis and neuropathy. The connection with microtubule stability was shown only with Taxol feeding/rescue experiment. It seems that the effect of Taxol is not specific for INSC mutants, as Fig 6D shows largely similar effects of Taxol at all doses on K/M and K/R. This suggests the general effects of Taxol on necrosis (DAPI/PI staining) and climbing rate, independent of K/M or K/R allele.

5. Taxol phenotype and rescue:

a. For Taxol feeding experiments, is there a difference in fly feeding among different Taxol doses or different genotypes?

b. Are the scale bars in Fig. 6G and 6I the same? The cells with Taxol treatment shown in Fig. 6I look significantly bigger than the cells in Fig 6G.

c. In Figure 5D-E, the authors suggest that there is a progressive increase in aggregation over time, was this statistically compared?

d. In the results, the authors did not discuss the results showing no effect after taxol or colchicine treatment for the UAS-Baz-RNAi group (Figure 5F-G).

Minor points:

1. Some labels in the figures are misspelled.

2. Appendix Table S2 is showing whole genome sequencing data from patients III-3 and III-8 instead of III-8 and III-10, as mentioned in the methods section.

3. Experimental sample sizes and number of flies used for climbing performance were not indicated.

4. Figure 2 E legend says scale bar is 50µm, but in the figure it shows nothing.

Referee #3 (Comments on Novelty/Model System for Author):

See for specific comments on the technical quality in my remarks. The novelty of the study is based on the reporting of a novel causative gene, albeit with a single mutation, in CMT patients. The medical impact in the current version is medium, as the link to microtubule stability and INSC-induced CMT is not clear, for specific comments see my remarks. Both the *Drosophila* and the cell lines the authors use are adequate and offer many possibilities to address their questions.

Referee #3 (Remarks for Author):

The manuscript by Yeh et al. reports the discovery of a single variant in the INSC gene in patients with autosomal dominant Charcot-Marie-Tooth (CMT) neuropathy from a three-generation family. The authors used a combination of *Drosophila* and SH-SY5Y cellular models to study the pathogenicity of the CMT-associated INSCMet70Arg and to compare its effect to conditions of down-regulated INSC.

INSC is a part of a three-component complex which also includes the proteins PAR3 and LGN, with a role predominantly studied in asymmetric cell division. PAR3 and LGN were previously associated with neurodevelopmental disorders, while this manuscript is a first report of mutant INSC being involved in neurodegeneration and CMT. The authors based most of the work presented on the premise that the CMT-associated INSC mutation is: 1) inducing INSC haploinsufficiency, 2) might affect microtubule organization as a result of impaired LGN/INSC interaction (based on the concept that the dynein-adaptor NuMA competes to bind to LGN, and that the LGN-bound NuMA complex can recruit the microtubule motor dynein), and 3) is sharing microtubule instability as a pathomechanism with several other CMT subtypes.

The manuscript depicts several functional defects resulting from the loss/decrease/expression of the INSC CMT mutation, and the other components of the PAR3/INSC/LGN or PIL complex in the femoral chordotonal organ (FeCO) of adult *Drosophila* and SH-SY5Y cells. The message of the manuscript will be of interest to the field of CMT studies and neurodegeneration in general. However, there are several concerns that will need to be addressed to solidify the findings described and to make this manuscript accessible to readers.

Major experimental concerns:

- There is a general lack of information on what the individual data points represent. The authors should clarify what the N in each graph presented (single leg, cell, et c) throughout the manuscript.
- The authors should show the data set related to the claim "in a locomotor assay, *dInsc*-RNAi flies exhibited normal behavior at day 3 post eclosion" in the main or supplementary material. Also in figure 6 they mention a colchicine treatment that could not be found in the figures.
- The authors should include evidence for validation of the RNAi lines used in their experiments - How was the putative down-regulation of *dInsc*, *bazooka* or *pins* validated (in particular upon the GS induction)? Similarly, were the *hInsc*-shRNAs used in SH-SY5Y validated? To what extent was the protein down-regulated?
- The recurrent use of the FeCO throughout the figures should be supported with a diagram of the organ, which would improve the accessibility of the results related to it for readers. This is particularly important already in Fig 2E-H where the reader is introduced to a specific *Insc* expression. Regarding this specific figure, the legend should add the specific name of the cuticle marker.
- In Fig. 4A, there is an obvious dichotomy of the data and almost half of the young affected individuals have higher mRNA levels than the other half. Could the authors explain if this could be a result of a gender or any other difference with the rest of the young affected individuals from the experiment? How do the authors explain such an effect on mRNA level induced by a missense mutation, when the other wild type allele should still provide wild type copy of INSC. Is it possible to perform analysis of the INSC protein levels in the same conditions that the mRNA was tested?
- In Fig. 4B and C, the *hInsc* protein levels in the FeCO should be quantified within the specific regions (cell bodies and dendrites). Currently, the line plots do not support the claim of decrease of the INSC levels, as they represent single line, that provides a single region in just one animal.
- Again in Fig. 4D, the authors should clarify if the co-localization was performed in 3D, on single slices, or on Z-stack projections. The second will not provide very accurate estimate of the co-localization of the different proteins. Similar comment for the H panel in the same figure.
- There is a complete lack of information on the image analysis details in the Method section. Detailed description should be added, which might resolve part of the questions on the co-localization analyses.
- In general, the description of the results in Fig. 6 is insufficient thus the reader is left to interpret them on their own. In panel A, there is a contradictory effect of Taxol on tubulin accumulations in K/M or K/R flies, briefly justified by dosage effect. On the other hand, the Result section finalizes with a conclusion that "low dose of Taxol can enhance microtubule stability" which contradicts both with the data in panel A and G-J. Moreover, in panel G, there is a co-localization analysis of α -tubulin and acetylated tubulin, with a biased choice of inset in the lower panel of the DMSO treatment in the INSC shRNA. Looking at the representative cells in the INSC shRNA condition, the acetylated tubulin signal does not overlap with the α -tubulin one because the authors chose to zoom-in to an area that does not contain acetylated tubulin. The authors should zoom-in to areas that contain both signals in order to demonstrate the decrease in co-localization. Again here, it is unclear how the co-loc analysis was performed. Since there is already a data-set for all conditions, the authors should analyze the MT organization in these cells to support the claims that perhaps that there is MT-stability defect.

Major concern regarding the overall text:

- The genetic and functional studies of CMT are a broad field that cannot be covered in a single introduction section. However, in its current shape the introduction does not even mention the complexity of the underlying pathomechanisms in CMT with different etiologies and the fact that in parallel to destabilized microtubules there are numerous other defects reported that might also contribute to the CMT pathology for the listed "several CMT2 subtypes, including CMT2D (GARS1), CMT2E (NEFL), and CMT2F (HSPB1)".
- The manuscript in its current shape does not coherently connect the different findings in the Results section. The text will benefit from extra information and improved description of the logic behind their results. For example, they start the result section (5) with the introductory sentence "Our results suggest reduced association of hINSCM70R with LGN may lead to MT instability in the adult PNS, as hINSC competes with NuMA for LGN-binding to regulate MT arrangement during spindle assembly in neuroblasts (Zhu et al, 2011)." While the authors describe possible relocation of the PILS components in Fig. 4, this does not suggest MT instability. This jumping to unsupported conclusions is typical throughout the manuscript and the authors should revisit this writing style and provide contextual description of their results and their possible impact in more logic and toned-down manner.
- A brief literature browse on INSC demonstrates that the authors implement more up-dated references, and include also studies that might point to alternative conclusions regarding the INSC/LGN interaction. For example, the study of Culurgioni et al., 2018 which suggests a possibility that fraction of the "Insc-bound pool of LGN acting independently of microtubule motors to promote asymmetric fate specification".
- The authors state that the Met70Arg substitution is located in the LGN-binding motif, which they claim that it is functionally conserved. As modelling the mutation in *Drosophila* is at the basis of their study the authors should show the protein sequence of this motif in evolutionary distant species to demonstrate the conserved motif. In this line, the absence (or presence) of highly conserved human/fly homolog should be clearly stated. This information, preferably accompanied by sequence alignment and/or diagrams of the human and fly homologs should be added. This will help the reader to understand readily the CRISPR CMT variant modeling and make sense of the variants generated.
- While the authors do observe rescue of *dInsc* hypomorphs with the transgenic human INSC, the statement "providing the evolutionary conservation of INSC in neurodevelopment" should be accompanied by additional information on whether they specifically refer to their own result or there are also other references (that should be included).
- The authors should provide more details on why was the scolopidium examined in fig.5 (besides for having MT-rich region). What is the function of these MT-rich regions in control flies? How and why would the tubulin puncta accumulate extracellularly? How is this relevant to modeling CMT?

Minor concerns:

- Spell out some of the acronyms to improve readability, for example ACD for asymmetric cell division.
- There are typos throughout the text and the figures.
- Clarify if the used transgenic *d/hInsc* are RNAi-resistant.
- The authors should clarify how they identified the different tissues in Fig. EV1A-B.
- Are the metrics presented in fig. 2L-N established or the authors used them for a first time. The analogy that they try to make with human patient is understandable, however the different measurements should be put in a context.

Response to Reviewers

Referee #1 (Remarks for Author)

The manuscript by Yeh describes the important role of hINSC in the adult PNS function. The authors first identified a missense mutation Met70Arg in the hINSC protein from CMT2 patients. To understand the relevance of this mutation, they leveraged the fly model to understand whether and how hINSC and its fly homolog Insc regulate PNS functions in fly. Interestingly, knockdown of fly Insc caused locomotion/gait defects and adult-onset necrosis in aging fly, which can be rescued by overexpression hINSC but not hINSC-Met70Arg mutant form. Like Insc, knockdown of Insc-binding partners, such as Pins and Baz, resembled Insc depletion. hINSC-M70R was expressed at a lower level and less stable in flies. Mechanistically, dysfunction of fly Insc caused microtubulin aggregation near adult proprioceptive organs, which is caused by destabilized microtubules. Treatment with low-dose Taxol, a microtubule destabilizer, can alleviate the microtubule aggregation and locomotive defects. Overall, this is an important study which establishes a causal relationship between hInsc mutation and CMT2 neuropathy. Moreover, it also reveals a mechanism whereby microtubule instability contributes to the disease pathogenesis. Thus, I recommend its publication after some minor revisions.

Response 1:

We thank Referee #1 (R1) for the positive and comprehensive assessment of this study. We have responded point-to-point to the issues raised by the reviewer. Please find below the evidence we provided to further support or refine the argument of this study.

- 1) Figure EV1A-B. Are those Insc-positive cells in leg discs neurons?

Response 1-1:

We stained with anti-elav antibody to label neurons in the leg disc, and found colocalization between *Insc*^{L407}-Gal4>UAS-*mCD8-GFP* and anti-elav as shown in the dashed circle below, indicating that Insc-positive cells are neurons in leg discs. In the revised manuscript, we move this set of figures from Fig EV1 to the NEW Appendix Fig S6.

Appendix Fig S6. The expression pattern and protein localization of *Drosophila Inscuteable* in larval tissues.

C-E. Representative confocal images of leg disc labeled with *mCD8-GFP* (green) under the control of *Insc¹⁴⁰⁷-Gal4* in third instar larvae. (Blue) DAPI staining; (magenta) anti-elav immunofluorescence. Scale bars: 50 μ m.

2) Figure 2I. did *Iav-Gal4* drive the expression in the same neurons in the adult legs as *Insc1407-Gal4* (Figure 2E)?

Response 1-2:

Yes, we compared the *mCD8-GFP* patterns driven by *Insc¹⁴⁰⁷-* and *Iav-Gal4* drivers, and found similar expression patterns in FeCO, tiCO, and tarsi sensory cell bodies, indicating the two Gal4 drivers express in the same neurons. The comparison is shown below. In the revision, we include these figures as the NEW Fig EV1A.

Figure EV1. Expression of *dInsc* in FeCO neuron and its role in inducing tubulin aggregation.

A. The patterns of UAS-*mCD8-GFP* driven by *dInsc*¹⁴⁰⁷-Gal4 (left) and *Iav*-Gal4 (right) in the leg. Green indicates *mCD8-GFP* and magenta indicates auto-fluorescence of the cuticle. Scale bars: 50 μ m.

3) Figure 4D-E. The authors show the defects of the protein colocalization between hInsc-M70R and LGN/PAR3. Did the authors also test the defects in protein-protein interactions in co-IP assays? If the authors have the co-IP results, they can enhance the impact. But the results are not required for publication.

Response 1-3:

We thank R1 for this constructive comment, which is also pointed out by the other 2 referees. We performed coIP experiments in SH-SY5Y cells to examine whether the M70R mutation affects the protein interactions between INSC and LGN or PAR3. We found that M70R associates more with PAR3 and less with LGN, as shown below. This finding is consistent with our colocalization experiments (now moved to Figure 4G-J), and further supports the impact of M70R on interacting with PIL components. In the revision, we add the coIP data as the NEW Fig EV3D.

Figure EV3. Both mRNA and protein level of *R/R* decrease in aging flies induced by PIL complex dysregulation

D. Co-immunoprecipitation to examine the association of FLAG-hINSC (WT and M70R) with MYC-LGN (left) or HA-PAR3 (right) in SH-SY5Y cells.

- 4) Figure 5A. the microtubule aggregation phenotype is interesting. Can the author discuss why the microtubules aggregate extracellularly near neurons upon *dInsc* or *Pins* knockdown? what happened to the endogenous microtubules when anti-alpha-tubulin antibody is used to detect them? Can they observe any aggregates for endogenous microtubule too?

Response 1-4:

Anti-alpha-tubulin does not work, as the leg cuticle is too hard for antibodies to penetrate. That being said, we managed to observe the endogenous microtubules with tubulin tracker, which is a tubulin dye that can penetrate through the leg cuticle. As shown in the revised Fig EV1F (also shown below), the pattern of tubulin tracker appears disorganized in the *dInsc*-RNAi group (upper right panel), indicating the aggregation of endogenous tubulin.

Regarding the extracellular tubulin aggregates, we re-examined its pattern in detail, and observed different tubulin patterns between *dInsc* knockdown flies and the controls with tubulin tracker. As shown below, the tubulin was well-organized in the control FeCO, representing microtubules. In contrast, *dInsc*-RNAi caused tubulin aggregates both intracellularly and extracellularly (Fig EV1D). As tubulin constitutes a significant portion of cellular components, our hypothesis suggests that the observed aggregates likely correspond to the remnants of deceased cells. We add these findings in the NEW Fig EV1F and modify the corresponding text in the revised manuscript.

Figure EV1. Expression of *dInsc* in FeCO neuron and its role in inducing tubulin aggregation.

F. Representative confocal images of tubulin aggregation in the FeCO neuron and the femur of 3-week-old *dInsc*¹⁴⁰⁷-Gal4>*dInsc*-RNAi flies, compared with the control (UAS-*lacZ*) and the rescue (*hINSC*^{WT} and *hINSC*^{M70R}) groups. Red indicates tubulin tracker signals. Green indicates the auto-fluorescence of cuticles. The FeCO neurons are encircled by the dashed line. Scale bars: 5 μ m.

- 5) Figure 5A. were the microtubule aggregates observed in other leg sites of *dInsc* RNAi flies? For example, near tibiotarsal chordotonal organ. Since *dInsc* is also expressed in CNS, any microtubule aggregated in CNS neurons?

Response 1-5:

We thank R1 for raising this potential issue. We did not observe obvious tubulin aggregation near tibiotarsal chordotonal organ nor in the adult brain when *dInsc* was knocked down, as shown below. Thus, the tubulin aggregation is likely specific to the FeCO neurons in the PNS. Given that FeCO is highly abundant in microtubules within the leg tissue, it is conceivable that the observed aggregation represents remnants of the deceased FeCO neurons. In the revised manuscript, we provide these figures as the NEW Fig EV1B and EV1E to support the hypothesis that “dysfunction of PIL complex caused tubulin aggregation in adult proprioceptive organs” in Results (5).

Figure EV1. Expression of *dInsc* in FeCO neuron and its role in inducing tubulin aggregation.

- A. Representative confocal images of tubulin aggregation in the femur of 3-week-old *dInsc*¹⁴⁰⁷-Gal4>UAS-*dInsc*-RNAi flies compared with UAS-*LacZ* control. Red indicates *tubulin-mCherry*. Scale bars: 50 μ m.
- D. Representative confocal images of the adult brain of 3-week-old *dInsc*¹⁴⁰⁷-Gal4>UAS-*dInsc*-RNAi compared with the UAS-*LacZ* control flies. Green indicates *tubulin-mCherry* fluorescence. Magenta indicates anti-DLG. Scale bars: 50 μ m.

Referee #2

(Comments on Novelty/Model System for Author)

The manuscript by Yeh et al reported the identification of a novel mutation in INSC gene in one family with CMT2 related neuropathy. The authors used *Drosophila* models to discern the causal relationship between the variant pMet70Arg (in heterozygous) and the neuropathy phenotypes. The authors showed that the loss of function of *dInsc* in flies causes locomotor and gait phenotypes in flies. They further examined the cellular phenotypes and colocalization among LGN, Par3 and *Insc* in fly leg tissue and observed increased necrosis, microtubule instability, and disruption of the PIL complex. The discovery of *Insc* variant is intriguing, and some of the experimental designs are well done. However, the main conclusion on a series of causal relationships, --INSC_M70R mutation leads to PIL complex dysfunction, then leads to microtubule instability, then leads to necrosis and neuropathy, were not well supported by the data. In particular, several concerns regarding the genetic and cellular analyses should be addressed to fully support the conclusion.

(Remarks for Author)

The manuscript by Yeh et al reported the identification of a novel mutation in INSC gene in one family with CMT2 related neuropathy. The authors used *Drosophila* models to discern the causal relationship between the variant pMet70Arg (in heterozygous) and the neuropathy phenotypes. The authors showed that the loss of function of *dInsc* in flies causes locomotor and gait phenotypes in flies. They further examined the cellular phenotypes and colocalization among LGN, Par3 and *Insc* in fly leg tissue and observed increased necrosis, microtubule instability, and disruption of the PIL complex. The discovery of *Insc* variant is intriguing, and some of the experimental designs are well done. However, the main conclusion on a series of causal relationships, --INSC_M70R mutation leads to PIL complex dysfunction, then leads to microtubule instability, then leads to necrosis and neuropathy, were not well supported by the data. In particular, several concerns regarding the genetic and cellular analyses should be addressed to fully support the conclusion.

Response 2:

We thank Referee #2 (R2) for providing both positive feedback and constructive criticism. Below we address the remaining issues raised.

1. Genetics, haploinsufficiency: the affected individuals in the family carry the variant pMet70Arg in heterozygous. It is critical to assess how a point mutation in heterozygous causes disease. There are two possibilities: loss of function as haploinsufficiency, or gain of function as dominant effects. The evidence supporting a loss of function conclusion is weak. The possibility of gain of function was not ruled out with the presented data.
 - a. Is the pMet70Arg allele expressed in the same level (mRNA and protein) as the wildtype allele in patients?

Response 2-1 (a):

This is not an easy task, as we did not have enough tissue samples donated from the patients. However, we managed to recall the patients, took blood samples again, and performed Western blotting. Consistent with our RT-PCR findings, the western blotting of blood samples showed a decreased level of INSC proteins in the patients as shown below. In the revised manuscript, we include these findings as the NEW Fig 4A and B.

We also performed qPCR and western blotting to the knock-in flies. The R/R flies exhibited reduced *dInsc*, both at mRNA and protein levels as shown below. This is consistent with the findings from the patient's samples, again suggesting that point mutation affects the protein level. In the revision,

we add these findings in the NEW Fig EV3A-C.

These findings indicate that the pMet70Arg allele expressed at a lower level than the wildtype allele in patients, suggesting partial loss of function.

The dominant negative effect is ruled out in Figure 2B, wherein *hINSC*^{M70R} transgene was expressed in *insc*^{I407} heterozygous mutant background. We found that expressing *hINSC*^{M70R} did not cause further toxicity than the *Lacz* control.

Figure 4. *hINSC*-M70R protein exhibited decreased levels and altered association with PIL complex.

- Relative mRNA abundance of *hINSC* in the PBMCs of young (n = 2) and old (n = 1) affected individuals, compared with healthy young (n = 2) and old (n = 2) controls. For each trial, 3 replicates of 3 cDNA preparations per participant were performed. The data points collected from the same group of participants in 2 separated trials, each conducted 6 months apart, were pooled together.
- Representative western blot of *hINSC* from PBMCs in young and old affected individuals, compared with healthy young and old controls.

Figure EV3. Both mRNA and protein level of *R/R* decrease in aging flies induced by PIL complex dysregulation

- Relative mRNA abundance in whole fly extracts of *M/M* and *R/R* male flies with its corresponding *K/K* control in week 1 and week 3. $n = 12$ flies/genotype from 3 independent technical replicates.
- Relative mRNA abundance in whole fly extracts of *M/M* and *R/R* female flies with its corresponding *K/K* control in week 1 and week 3. $n = 12$ flies/genotype from 3 independent technical replicates.
- Representative western blotting of whole fly extracts from *M/M* and *R/R* flies with its corresponding *K/K* control in week 1 and week 3. $n = 5$ flies/genotype.

Figure 2. Adult-onset depletion of PII complex causes locomotor and proprioceptive defects.

B. Quantification of the climbing activity of 1- and 3-week-old flies with overexpressing *dInsc*-WT, *LacZ*, *hINSC^{WT}*, *hINSC^{M70R}*, *dInsc*-RNAi, UAS-*GFP* and the groups co-overexpressing *dInsc*-RNAi with *hINSC^{WT}* or *hINSC^{M70R}* under the control of *dInsc^{InSITE}*-Gal4 driver; n = 57 - 110 flies/genotype from 5 independent fly crosses.

- b. In Figure 2B, the author claims that the expression of human wild type INSC (hINSCWT) not the mutant (hINSCM70R) can rescue the impaired locomotion phenotype caused by heterozygous loss of function of *dInsc*. However, the author did not show the protein expression levels of each UAS-hINSC transgenic line.

Response 2-1 (b):

We thank R2 for pointing out the issue. We performed Western blotting to address the concern. We observed in the transgenic flies decreased levels of M70R protein as shown below. This finding is consistent with our confocal results shown in Fig 4C and D. This piece of evidence is now included as the NEW Appendix Fig S4C. Please note that, in our hands, anti-hINSC antibody does not work well in fly lysates. We enhanced the contrast, but then saw a very weak band in the *LacZ* group, possibly representing a background signal. That being said, there is significant difference between WT and M70R groups upon averaging to the loading control.

Appendix Fig S4. *hINSC* transgenic fly mRNA and protein level validations

C. Representative western blotting of whole fly extracts from UAS-*hINSC*^{WT} and UAS-*hINSC*^{M70R}, and UAS-*LacZ* control, driven by *elav-GS-Gal4*; n = 5 flies/genotypes.

c. The dilution effects of the UAS/GAL4 could affect the behavior results in UAS-dInsc-RNAi (red patterned bar) vs UAS-dInsc-RNAi, UAS-hINSCM70R (teal patterned bar), as well as the UAS-hINSCWT (orange bar) vs UAS-dInsc-RNAi, UAS- UAS-hINSCWT (orange patterned bar).

Response 2-1 (c):

We thank R2 for pointing out the potential issue. We performed the experiment again with control groups to examine whether the behavioral rescue is due to dilution effect. As shown below, the knockdown and rescue effects are not due to dilution effect, but specific to *dInsc*-RNAi and *hINSC*-WT, respectively. In the revised manuscript, we replaced the original Fig2B with this figure as the NEW Fig 2B.

Figure 2. Adult-onset depletion of PIL complex causes locomotor and proprioceptive defects.

B. Quantification of the climbing activity of 1- and 3-week-old flies with overexpressing *dInsc*-WT, *LacZ*, *hINSC^{WT}*, *hINSC^{M70R}*, *dInsc*-RNAi, UAS-*GFP* and the groups co-overexpressing *dInsc*-RNAi with *hINSC^{WT}* or *hINSC^{M70R}* under the control of *dInsc^{InSITE}*-Gal4 driver; n = 57 - 110 flies/genotype from 5 independent fly crosses.

d. In Figure 2A, was the significance only compared with the UAS-w-RNAi (RU486) control? In Figure legend 2B it says that it is showing climbing activity for 1- and 3-week-old flies, but figure only shows 3-week-old data.

Response 2-1 (d):

We are sorry for the confusion. The climbing defect achieved statistical significance when comparing the knockdown groups of PIL components with each of the control groups, including UAS-*GFP* (RU486) and UAS-*w*-RNAi (RU486). The climbing defects are age-dependent, as we found more severe defects in week 3 than week 1. For simplicity, we have rearranged the groups to show all the statistical significance in the revised Fig 2A as below.

Also, we apologize for omitting the 1-week-old flies in the old Fig 2B. In the revised manuscript we include both 1- and 3-week-old flies in the NEW Fig 2B.

Figure 2. Adult-onset depletion of PIL complex causes locomotor and proprioceptive defects.

- A. Quantification of the climbing activity of 3-day-old to 3-week-old adult flies with *dInsc*-RNAi, *Baz*-RNAi and *Pins*-RNAi under the control of inducible pan-neuronal driver (*elav-GS-Gal4*) upon feeding with RU486, comparing with the age-matched controls (*mCD8-GFP*) and (*w*-RNAi); n = 100 - 249 flies/genotype from 5 independent fly crosses.

Figure 2. Adult-onset depletion of PIL complex causes locomotor and proprioceptive defects.

B. Quantification of the climbing activity of 1- and 3-week-old flies with overexpressing *dInsc*-WT, *LacZ*, *hINSC^{WT}*, *hINSC^{M70R}*, *dInsc*-RNAi, UAS-*GFP* and the groups co-overexpressing *dInsc*-RNAi with *hINSC^{WT}* or *hINSC^{M70R}* under the control of *dInsc^{InSITE}-Gal4* driver; n = 57 - 110 flies/genotype from 5 independent fly crosses.

- e. Is the phenotype of *Baz* KD shown in Fig. 2A significant compared to the first or second control group? Is the KD efficiency examined? The role of *Baz* in this condition might be more complicated than that proposed by the authors if authors believe the KD efficiency is sufficient.

Response 2-1 (e):

To examine whether *Baz*-RNAi exhibited significant locomotor impairment compared to the first or second control group, we increased the sample size and performed the climbing experiment again. As shown below, we found *Baz*-RNAi to cause little defects in week 1, but the defect reached

statistical significance in weeks 2 and 3 when compared to both controls. This finding is added to the revised Fig 2A.

We also performed qPCR to examine the KD efficiency. As shown below, while *dInsc*-RNAi and *Pins*-RNAi achieved 50% KD efficiency, *Baz*-RNAi only removed 24% of the endogenous *Baz* mRNA. This could explain why the climbing defects in the *Baz*-RNAi group were less severe. In the revised manuscript, we added this finding in the Appendix Fig S3D.

Figure 2. Adult-onset depletion of PIL complex causes locomotor and proprioceptive defects.

A. Quantification of the climbing activity of 3-day-old to 3-week-old adult flies with *dInsc*-RNAi, *Baz*-RNAi and *Pins*-RNAi under the control of inducible pan-neuronal driver (*elav-GS-Gal4*) upon feeding with RU486, comparing with the age-matched controls (*mCD8-GFP*) and (*w*-RNAi); *n* = 100 - 249 flies/genotype from 5 independent fly crosses.

Baz/*dInsc*/*Pins* Knockdown flies validation

Appendix Fig S3. *hINSC* shRNA knockdown SH-SY5Y cell and *Baz/dInsc/Pins* knockdown fly validation

D. Relative mRNA abundance in whole fly extracts of *dInsc*, *Pins* and *Baz* RNAi flies driven by *elav-GS-Gal4*; n = 24 flies/genotype from 3 independent fly crosses.

2. Homology between fly and human: homology analysis between the human and drosophila INSC is lacking. Specifically, it is unclear how the conservation between residue K305 in drosophila and human M70R is established. The large difference in residue location is also rather unusual.

Response 2-2:

We thank R2 for raising this issue, which is also pointed out by R3. The homology between human INSC and fly Insc at the primary sequence level is not high, with sequence identity of 21 % and similarity of 32%. However, the M70R is located within a stretch of 38 amino acid peptides (human a.a. 70-118 or *Drosophila* a.a. 303-340) that are known to mediate polar and hydrophobic interactions with the N-terminal TPR domain of LGN, stabilizing the binding at nanomolar affinity (Culurgioni *et al.*, 2011. PMID: 22171003). The observed discrepancy in sequence length between fly Insc and human INSC preceding the conserved region could be attributed to evolutionary divergence and subsequent non-conservation.

In the revised manuscript, we show sequence alignment of the LGN binding domain between hINSC and dInsc as the NEW Appendix Fig S5B. The sequence alignment of this region is also demonstrated by previous literatures (Fig. 1C in Yuzawa *et al.*, 2011, PMID: 22074847; Fig. 3G in Culurgioni *et al.*, 2018, PMID: 29523789), showing that M70 resides in the N' of LGN-binding domain, and that K305 is the corresponding residue in fly.

Appendix Figure S5. Schematic of a seamless editing knock-in strategy and sequence alignment of LGN binding domain.

B. Sequence alignment of human and flies LGN-binding domain of INSC.

3. Complex formation and colocalization analysis: it was concluded that the LGN, Par3 and Insc form a complex in Drosophila neurons. The conclusion is mainly based on colocalization analysis in cultured cells and in fly leg prep. Biochemical evidence would be required for any statement of complex formation or protein-protein interaction. Furthermore, there are several issues with the presented colocalization analysis.

Response 2-3:

We thank R2 for this constructive comment, which is also pointed out by the other 2 referees. We performed the coIP experiments in SH-SY5Y cells to examine whether the M70R mutation affects the protein interactions between INSC and LGN or PAR3. We found that M70R associates more with PAR3 and less with LGN as shown below. This finding is consistent with our colocalization experiments (Figure 4G-J), and further supports the impact of M70R on interacting with PIL components. We add the coIP data as the NEW Fig EV3D in the revised manuscript.

Figure EV2. Both mRNA and protein level of *R/R* decrease in aging flies induced by PIL complex dysregulation

D. Co-immunoprecipitation to examine the association of FLAG-hINSC (WT and M70R) with MYC-LGN (left) or HA-PAR3 (right) in SH-SY5Y cells.

a. In Fig. 4D-4G, the cell morphology of hINSC_M70R is significantly different from hINSC-WT (Fig. 4F). Such difference precludes a reliable analysis of colocalization.

Response 2-3 (a):

We are sorry about presenting cells of different morphologies. Under microscopy, we observed both spindle-shaped and flat cells. In the revised manuscript, we add representative pictures of cells of similar morphology for better comparison. Please see the NEW Fig 4G-J as shown below.

For a reliable analysis of colocalization, we took a random, unbiased approach by averaging all of the cells in the field (approximately 9-10 cells in each condition; a total of 3 independent transfections), and presented the statistics in Fig 4H and 4J.

Figure 4. hINSC-M70R protein exhibited decreased levels and altered association with PIL complex

- G. Immunostaining of SH-SY5Y cells transfected with FLAG-hINSC^{M70R} (red) to visualize the colocalization with MYC-LGN (green), comparing with FLAG-hINSC^{WT} (red) control. Scale bar: 10 μ m.
- H. Pearson's coefficient of colocalization of anti-FLAG (red) and anti-MYC (green) fluorescence in (G); n = 9 - 10 cells/condition from 3 independent transfection.
- I. Immunostaining of SH-SY5Y cells transfected with FLAG-hINSC^{M70R} (red) to visualize the colocalization with HA-PAR3 (green), comparing with FLAG-hINSC^{WT} (red) control. Scale bar: 10 μ m.
- J. Pearson's coefficient of colocalization of anti-HA (green) and anti-FLAG (red) fluorescence in (I); n = 9 - 10 cells/condition from 3 independent transfection.

- b. In the colocalization study, it was shown that hINSC_M70R resulted in a decrease in LGN and increase in PAR3 colocalization. The significance and relevance of this observation are unclear.

Response 2-3 (b):

To emphasize the significance of the interaction between hINSC and LGN or Par3, we add a paragraph in the introduction as shown below. In that paragraph, we state that microtubule instability causes CMT, which is supported by previous literatures. Since hINSC-LGN interaction is crucial for maintaining microtubule stability, decreased colocalization of hINSC and LGN could affect CMT.

“Several common gene mutations have been linked to CMT2, including *MFN2* (CMT2A), *RAB7* (CMT2B), *GARS* (CMT2D), and *NEFL* (CMT2E). Previous literature shows that these mutations affect mitochondrial trafficking (CMT2A), microtubule trafficking of lysosomes (CMT2B), acetylated tubulin (CMT2D), and neurofilaments (CMT2E) (Markworth *et al*, 2021). Notably, these CMT2 subtypes share a common defect of decreased microtubule-stabilization caused by disrupting α -tubulin acetylation (Ackerley *et al*, 2006; Mo *et al*, 2018; Brownlees *et al*, 2002), which subsequently leads to microtubule breakdown and cause axonal defects. Since microtubules regulate axonal transport by forming a dynamic network that enables efficient intraneuronal transport, these findings suggest that microtubule destabilization within axons can be a common feature across genetically diverse forms of CMT2. Understanding the underlying genetic and molecular mechanisms can aid in improving the diagnosis, treatment, and management of CMT and provide crucial insights into the pathogenic mechanisms of this neurodegenerative disorder.”

- c. In Fig. 4H, the Baz-mCherry signal in the INSC M70R group is much weaker than that in the INSC WT group. Is this image representative? On the other hand, the INSC M70R-EGFP signal is much stronger than the WT-EGFP signal, which is opposite to the observation in Fig. 4B.

Authors should explain the inconsistency.

Response 2-3 (c):

Baz-mCherry is much weaker than Pin-mCherry. In full Z-stack projection, Baz-mCherry fluorescence would be masked by the relatively abundant hINSC-EGFP, hence we showed a single representative layer in the old Fig 4H. Indeed, in that layer, the hINSC-M70R-EGFP signal may not be representative, as it appeared much stronger than the WT-EGFP signal. For the revision, we show the sum slice projection of the stacked images as shown below in K. For the quantification, we examined the colocalization of all the layers, and then performed 3D colocalization analysis. The results are shown below in L. We include these figures as the NEW Fig 4KL.

Figure 4. hINSC-M70R protein exhibited decreased levels and altered association with PIL complex.

K. Representative images of co-expressing *hINSC^{M70R}-EGFP* (green) and *Baz-mCherry* (red) in 3-week-old flies under *Iav-Gal4*, compared with the age-matched controls (*hINSC^{WT}-EGFP*). Scale bars: 5 μ m.

L. Pearson's coefficient of colocalization of EGFP (green) and mCherry (red) fluorescence in (K); n = 3 flies/genotype from 3 independent fly crosses.

d. It is unclear how Fig 4A is generated. Were healthy old individuals to test for panel A? Was only one healthy young control analyzed?

Response 2-3 (d):

For Fig 4A, we prepared cDNA and protein lysate from the peripheral blood mononuclear cells (PBMCs) of both affected and unaffected family members, then performed qPCR to quantify the mRNA abundance of *hINSC*. As it is voluntary to participate in the test, we were only able to recruit 2 healthy young, 2 affected young, 2 healthy old, and 1 affected old individuals from the family. We include all the available data points in the revised manuscript.

Figure 4. *hINSC*-M70R protein exhibited decreased levels and altered association with PIL complex.

A. Relative mRNA abundance of *hINSC* in the PBMCs of young (n = 2) and old (n = 1) affected individuals, compared with healthy young (n = 2) and old (n = 2) controls. For each trial, 3 replicates of 3 cDNA preparations per participant were performed. The data points collected from the same group of participants in 2 separated trials, each conducted 6 months apart, were pooled together.

4. Necrosis and microtubule aggregation, cause and effect: the authors concluded a series of causal relationships, *INSC* M70R mutation leads to PIL complex dysfunction, then leads to microtubule instability, then leads to necrosis and neuropathy. The connection with microtubule stability was shown only with Taxol feeding/rescue experiment. It seems that the effect of Taxol is not specific for *INSC* mutants, as Fig 6D shows largely similar effects of Taxol at all doses on K/M and K/R. This suggests the general effects of Taxol on necrosis (DAPI/PI staining) and climbing rate, independent of K/M or K/R allele.

Response 2-4:

We appreciate the reviewer for bringing up this subtle yet important point. We believe that Taxol treatment showed different effects on the K/M and K/R animals for the following reasons. First, the K/M and K/R flies were heterozygous animals. From the result of Fig. 2D, it is obvious that the climbing defect is much milder in the K/R flies comparing to the homozygous R/R flies. Because the differences in phenotype severity was not great, we did not expect the effect of Taxol treatment to be very striking. Second, although the overall patterns of the K/M vs. K/R panels in Fig. 6D were similar, the results of treatment with the DMSO control and the two lower doses of Taxol were slightly different. To be specific, the K/R flies had a much lower level of DAPI signal in the DMSO

group, which increased drastically in flies treated with 5 and 50 μ M Taxol. In the K/M flies, although 5 μ M Taxol resulted in a significant increase in DAPI signal, the level of increase was much milder comparing to that of the K/R, and the DAPI signal level decreased to a level similar to that of DMSO under 50 μ M Taxol. The same trend can be found for the climbing assay, with the K/R group showing a worse phenotype than the K/M flies and a striking rescue effect upon 5 and 50 μ M Taxol treatment. Together, we believe that the effect of Taxol to the K/M and K/R show slight but important differences.

To strengthen the connection with microtubule stability, we treated the flies with another microtubule-stabilizing drug Cevipabulin. While Cevipabulin has a multifunctional role on microtubule, it stabilizes microtubule via a distinct mechanism than Taxol (Beyer et al., 2008. PMID: 18381436; Xiao *et al.*, 2006. PMID: 16801540). We found that treating the flies with 0.5 mM Cevipabulin can also rescue the morphological and functional defects of the K/R flies as shown below. As both drugs ameliorate the defects of the K/R flies, we conclude that microtubule defects underlie the phenotypes seen in the K/R flies.

We also took a genetic approach to establish the link between *Insc* and microtubule stability. We used an optogenetics tool wherein a microtubule-severing enzyme spastin (MTDS) can be activated to destabilize microtubule (Liu GY *et al.*, 2022. PMID: 35686621). As shown below, flies with MTDS expressed in the adult nervous system exhibited a climbing defect upon blue light exposure, which is partially reversed upon coexpression of *Insc*. This piece of evidence further supports that enhancing *Insc* activity can enhance the climbing function.

In the revision, we edited the Results (6) to include these findings, which is shown as the NEW Fig EV5.

Figure EV5. The destabilization of microtubules caused by M70R mutation can be rescued with both microtubule-stabilizing agents and genetic manipulation.

- A. Representative confocal images of FeCO neurons of *K/M* and *K/R* 3-week flies treated with 0.5 mM of microtubule stabilizer Cevipabulin for 7 days, co-stained with PI (magenta) and DAPI (green), compared with vehicle control (DMSO). The FeCO neurons are encircled by the dashed line. Scale bars: 5 μ m. Quantifications are shown in the lower panels.
- B. The schematic of light-inducible microtubule disassembly system (MTDS) in a fly model. The dimerization of CRY2 and CIB can be induced by blue light. The CRY2 is fused with a microtubule-severing enzyme Spastin, and CIB is fused with microtubule-binding domain (MTBD). Dimerization upon blue light stimuli induces accumulation of Spastins on microtubules, which in turn induces disassembly of microtubules in MTDS-expressing cells.

C. Quantification of the climbing activity of Week 1 flies of UAS-*MTDS* (EtOH), UAS-*MTDS* (RU486), and UAS-*MTDS*, UAS-*dInsc-WT* (RU486) under the control of RU486-inducible pan-neuronal driver (*elav-GS-Gal4*) upon 48 hrs of blue-light exposure, comparing with the conditional control (red light); n = 30 flies/genotypes from 3 independent fly crosses.

5. Taxol phenotype and rescue:

a. For Taxol feeding experiments, is there a difference in fly feeding among different Taxol doses or different genotypes?

Response 2-5 (a):

We performed feeding assay (Lien *et al.*, PMID: 32627932) to examine whether there is a difference in fly feeding among different Taxol doses or different genotypes. As shown below, neither Taxol doses nor genotype differences affected feeding. We added these findings in Appendix Fig S8 in the revised manuscript.

Appendix Fig S8. Feeding assay of Taxol feeding in different doses and genetic background

A. Feeding index of *W1118* (*K/K*) flies fed with different doses (5 μM, 50 μM and 5 mM) of Taxol; 24 flies/genotypes from 6 independent technical replicates.

- B. Feeding index of *K/K*, *K/M*, *K/R* flies fed with 50 μ M Taxol; 24 flies/genotypes from 6 independent technical replicates.
- C. Feeding index of UAS-*LacZ*, UAS-*dInsc*-RNAi, UAS-*Pins*-RNAi and UAS-*Baz*-RNAi flies driven by *elav-GS-Gal4*. The flies were fed with 50 μ M Taxol; 12 flies/genotypes from 3 independent technical replicates.

b. Are the scale bars in Fig. 6G and 6I the same? The cells with Taxol treatment shown in Fig. 6I look significantly bigger than the cells in Fig 6G.

Response 2-5 (b):

We apologize for the confusion. We have replaced Fig 6G with cells of representative size in the revision.

Figure 6. Treatment of optimal concentration of Taxol rescued the morphological and functional defects in cell and aging fly.

- G. Immunostaining of *hINSC*-shRNA transfected SH-SY5Y cells and treated with DMSO (a vehicle control of Taxol) to visualize the colocalization of α -tubulin (red) and acetylated-tubulin (green) compared with scramble-shRNA control. Blue is DAPI staining. Scale bars: 10 μ m.
- I. Immunostaining of *hINSC*-shRNA transfected SH-SY5Y cells and treated with microtubule-stabilizer Taxol to visualize the colocalization of α -tubulin (red) and acetylated-tubulin (green) compared with scramble shRNA control. Blue is DAPI staining. Scale bars: 10 μ m.

c. In Figure 5D-E, the authors suggest that there is a progressive increase in aggregation over time, was this statistically compared?

Response 2-5 (c):

Yes, there is a statistically significant increase of tubulin aggregation in the K/R group over time. In the revised Fig 5D and E (as shown below), we combined the experimental groups in one figure to better present the difference between Weeks 1 and 3.

Figure 5. Aging flies of PIL loss-of-function exhibited tubulin aggregation in FeCO neurons.

- D. Quantification of the number of aggregative tubulins in (Fig EV4A); n = 5 - 11 flies/genotype from 3 independent fly crosses.
- E. Relative abundance of aggregative tubulins of different sizes in (Fig EV4A); n = 5 - 11 flies/genotype from 3 independent fly crosses.

d. In the results, the authors did not discuss the results showing no effect after taxol or colchicine treatment for the UAS-Baz-RNAi group (Figure 5F-G).

Response 2-5 (d):

Indeed, in Fig 5F we did not observe obvious tubulin aggregation in *Baz*-RNAi. This could be attributed to the relatively low knockdown efficiency of *Baz*-RNAi. As shown below, we performed qPCR to examine the KD efficiency. While *dInsc*-RNAi and *Pins*-RNAi achieved ~50% KD efficiency, *Baz*-RNAi only removed 24% of the endogenous *Baz* mRNA.

Given the minimal impact of *Baz*-RNAi on tubulin aggregation, there is not much left to be modulated by Taxol or colchicine. This could also explain why we observed less climbing defects in the *Baz*-RNAi group (Fig 2A).

Baz/dInsc/Pins Knockdown flies validation

Appendix Fig S3. *hINSC* shRNA knockdown SH-SY5Y cell and *Baz/dInsc/Pins* knockdown fly validation

D. Relative mRNA abundance in whole fly extracts of *dInsc*, *Pins* and *Baz* RNAi flies driven by *elav-GS-Gal4*; n = 24 flies/genotype from 3 independent fly crosses.

Minor points:

1. Some labels in the figures are misspelled.

Response 2-6:

Thanks so much for pointing this out. We went through the figures and identified many typos, which are corrected in the revised manuscript.

2. Appendix Table S2 is showing whole genome sequencing data from patients III-3 and III-8 instead of III-8 and III-10, as mentioned in the methods section.

Response 2-7:

We are sorry about the mislabeling. We have corrected it in the Methods section.

3. Experimental sample sizes and number of flies used for climbing performance were not indicated.

Response 2-8:

We thank R2 for these constructive comments, which are also pointed out by R3. In the revised manuscript, we have clarified the experimental sample sizes and number of flies used in the legend of each graph presented.

4. Figure 2 E legend says scale bar is 50µm, but in the figure it shows nothing.

Response 2-9:

Thank you! We have added the scale bar to Fig 2E.

Referee #3

(Comments on Novelty/Model System for Author)

See for specific comments on the technical quality in my remarks. The novelty of the study is based on the reporting of a novel causative gene, albeit with a single mutation, in CMT patients. The medical impact in the current version is medium, as the link to microtubule stability and INSC-induced CMT is not clear, for specific comments see my remarks. Both the *Drosophila* and the cell lines the authors use are adequate and offer many possibilities to address their questions.

(Remarks for Author)

The manuscript by Yeh et al. reports the discovery of a single variant in the INSC gene in patients with autosomal dominant Charcot-Marie-Tooth (CMT) neuropathy from a three-generation family. The authors used a combination of *Drosophila* and SH-SY5Y cellular models to study the pathogenicity of the CMT-associated INSCMet70Arg and to compare its effect to conditions of down-regulated INSC.

INSC is a part of a three-component complex which also includes the proteins PAR3 and LGN, with a role predominantly studied in asymmetric cell division. PAR3 and LGN were previously associated with neurodevelopmental disorders, while this manuscript is a first report of mutant INSC being involved in neurodegeneration and CMT. The authors based most of the work presented on the premise that the CMT-associated INSC mutation is: 1) inducing INSC haploinsufficiency, 2) might affect microtubule organization as a result of impaired LGN/INSC interaction (based on the concept that the dynein-adaptor NuMA competes to bind to LGN, and that the LGN-bound NuMA complex can recruit the microtubule motor dynein), and 3) is sharing microtubule instability as a pathomechanism with several other CMT subtypes.

The manuscript depicts several functional defects resulting from the loss/decrease/expression of the INSC CMT mutation, and the other components of the PAR3/INSC/LGN or PIL complex in the femoral chordotonal organ (FeCO) of adult *Drosophila* and SH-SY5Y cells. The message of the manuscript will be of interest to the field of CMT studies and neurodegeneration in general.

However, there are several concerns that will need to be addressed to solidify the findings described and to make this manuscript accessible to readers.

Response 3:

We appreciate Referee #3 (R3) for recognizing the impact of the messages we hope to deliver with this study. Below we provide experimental evidence to address the concerns.

Major experimental concerns:

- There is a general lack of information on what the individual data points represent. The authors should clarify what the N in each graph presented (single leg, cell, etc) throughout the manuscript.

Response 3-1:

We thank R3 for this constructive comment, which is also pointed out by R2. In the revised manuscript, we have clarified the N in each graph presented.

- The authors should show the data set related to the claim "in a locomotor assay, *dInsc*-RNAi flies exhibited normal behavior at day 3 post eclosion" in the main or supplementary material. Also in figure 6 they mention a colchicine treatment that could not be found in the figures.

Response 3-2:

We apologize for the omission. We added the results to the NEW Fig 2A in the revised manuscript to support the claim "in the locomotor assay, *dInsc*-RNAi flies exhibited normal behavior at day 3 post eclosion."

For the colchicine treatment, we are sorry about the confusion. The results of colchicine treatment were originally shown in the old Figure 5FG, but the corresponding sentence "Conversely, Colchicine, a MT-destabilizing drug, exerted the opposite effect and further exacerbated the aggregation of tubulin" was not referred to the Figure. To better connect the text to its corresponding figures, we edited the text in the revision as shown below in bold:

"...we found that treatment of Taxol at low concentration in the PIL complex knockdown flies reduced the number and size of tubulin accumulations in the proprioceptive structure (Fig 5F and G). Conversely, Colchicine, a microtubule-destabilizing drug, exerted the opposite effect and further exacerbated the aggregation of tubulin (**Fig 5F and G**)."

Figure 2. Adult-onset depletion of PIL complex causes locomotor and proprioceptive defects.

B. Quantification of the climbing activity of 3-day-old to 3-week-old adult flies with *dInsc*-RNA, *Baz*-RNAi and *Pins*-RNAi under the control of inducible pan-neuronal driver (*elav-GS-Gal4*) upon feeding with RU486, comparing with the age-matched controls (*mCD8-GFP*) and (*w*-RNAi); n = 100 - 249 flies/genotype from 5 independent fly crosses.

- The authors should include evidence for validation of the RNAi lines used in their experiments - How was the putative down-regulation of *dInsc*, *bazooka* or *pins* validated (in particular upon the GS induction)? Similarly, were the *hINSC*-shRNAs used in SH-SY5Y validated? To what extent was the protein down-regulated?

Response 3-3:

We performed qPCR to examine the KD efficiency. As shown below, while *dInsc*-RNAi and *Pins*-RNAi exerted 50% KD efficiency, *Baz*-RNAi only removed 24% of the endogenous *Baz* mRNA. This could explain why we observed lesser degree of climbing defects in the *Baz*-RNAi group in Fig 2A. In the revised manuscript, we added this finding in the NEW Appendix S3D.

We also performed qPCR and western blotting to examine the mRNA abundance and protein levels of *hINSC*, respectively. As shown below, both the mRNA and protein are reduced in *hINSC* shRNA-treated SH-SY5Y cells. In the revised manuscript, we included these results in the NEW Appendix

Fig S3A-C.

hINSC shRNA SH-SY5Y cell knockdown validation

Baz/dInsc/Pins Knockdown flies validation

Appendix Fig S3. *hINSC* shRNA knockdown SH-SY5Y cell and *Baz/dInsc/Pins* knockdown fly validation

- A. Representative blot from 3 independent western blottings of SH-SY5Y cell extracts of *hINSC*-shRNA and scramble shRNA control.
- B. Quantification of (A).
- C. Relative *hINSC* mRNA abundance in SH-SY5Y cell extracts of *hINSC*-shRNA and scramble shRNA control.
- D. Relative mRNA abundance in whole fly extracts of *dInsc*, *Pins* and *Baz* RNAi flies driven by *elav-GS-Gal4*; n = 24 flies/genotype from 3 independent fly crosses.

• The recurrent use of the FeCO throughout the figures should be supported with a diagram of the organ, which would improve the accessibility of the results related to it for readers. This is particularly important already in Fig 2E-H where the reader is introduced to a specific *Insc*

expression. Regarding this specific figure, the legend should add the specific name of the cuticle marker.

Response 3-4:

Thank you for the suggestion. We have added a diagram of the fly leg with an emphasis on FeCO in the NEW Fig 2E in the revised manuscript. The magenta color is the autofluorescence of the cuticle. In the revised manuscript, we state it in the corresponding legend of Fig 2E.

Figure 2. Adult-onset depletion of PIL complex causes locomotor and proprioceptive defects.

E. Thoracic segments 1 (T1) leg of an adult fly expressing *mCD8-GFP* (green) under the control of *dInsc¹⁴⁰⁷-Gal4*. (Right) a schematic of the FeCO neuron in adult leg. Scale bars: 50 μ m.

Magenta is the auto-fluorescence of the cuticle.

- In Fig. 4A, there is an obvious dichotomy of the data and almost half of the young affected individuals have higher mRNA levels than the other half. Could the authors explain if this could be a result of a gender or any other difference with the rest of the young affected individuals from the experiment? How do the authors explain such an effect on mRNA level induced by a missense mutation, when the other wild type allele should still provide wild type copy of INSC. Is it possible to perform analysis of the INSC protein levels in the same conditions that the mRNA was tested?

Response 3-5:

As R2 also expressed concern about this experiment, we recalled the patients, took blood samples, performed RT-PCR again, and pooled together all the data points. This time, we also examined INSC protein levels by western blotting. Consistent with our RT-PCR findings, the western blotting of blood samples showed a decreased level of INSC proteins in the patients. The Western blot also indicates approximately 20-30% reduction of hINSC protein in the affected patients compared to the controls. We include these findings as the NEW Fig 4A and B and also show below.

We examined whether the gender affects INSC expression in the KI flies. As shown below, R/R flies showed a reduction in *dInsc* mRNA abundance and protein level independent of sex. In the revised manuscript, we include these findings as the NEW Fig EV2.

Figure 4. hINSC-M70R protein exhibited decreased levels and altered association with PIL complex.

- A.** Relative mRNA abundance of *hINSC* in the PBMCs of young (n = 2) and old (n = 1) affected individuals, compared with healthy young (n = 2) and old (n = 2) controls. For each trial, 3 replicates of 3 cDNA preparations per participant were performed. The data points collected from the same group of participants in 2 separated trials, each conducted 6 months apart, were pooled together.
- B.** Representative Western blot of *hINSC* from PBMCs in young and old affected individuals, compared with healthy young and old controls.

Figure EV2. Both mRNA and protein level of *R/R* decrease in aging flies induced by PIL complex dysregulation

- A. Relative mRNA abundance in whole fly extracts of *M/M* and *R/R* male flies with its corresponding *K/K* control in week 1 and week 3. n = 12 flies/genotype from 3 independent technical replicates.
- B. Relative mRNA abundance in whole fly extracts of *M/M* and *R/R* female flies with its corresponding *K/K* control in week 1 and week 3. n = 12 flies/genotype from 3 independent technical replicates.
- C. Representative western blotting of whole fly extracts from *M/M* and *R/R* flies with its corresponding *K/K* control in week 1 and week 3. n = 5 flies/genotype.

• In Fig. 4B and C, the hINSC protein levels in the FeCO should be quantified within the specific regions (cell bodies and dendrites). Currently, the line plots do not support the claim of decrease of the INSC levels, as they represent single line, that provides a single region in just one animal.

Response 3-6:

The hINSC protein levels in the FeCO were quantified within the specific regions (cell bodies and dendrites). The cell body and dendrite regions are defined according to Fig 1A and B in Mamiya *et*

al., 2023 (PMID: 37562405) as shown below. The line plots in the NEW Fig 4CD (old Fig 4B and C) are the representative plots of the NEW Fig 4E and F (old Fig 4I and J), which are the quantification results from N=3 of 9 flies per genotype.

Fig 1AB in Mamiya *et al.*, 2023 (PMID: 37562405)

Figure 4. hINSC-M70R protein exhibited decreased levels and altered association with PIL complex.

- D. Representative fluorescence intensity profiles were generated to visualize co-localization of *hINSC^{M70R}-EGFP* and *Pins-mCherry* of cell bodies and dendrites in 1- and 3-week-old flies, compared with the age-matched controls (*hINSC^{WT}-EGFP*). The linear region of interest (ROI) was drawn manually from left to right.
- E. Pearson's coefficient of colocalization *hINSC^{WT}* and *hINSC^{M70R}* with *Pins* in (C) in cell bodies and dendrite in FeCO neurons of 1-week-old flies; n = 9 flies/genotype from 3 independent fly crosses.
- F. Pearson's coefficient of colocalization of whole FeCO neurons of 1- and 3-week-old flies in (C); n = 8 - 9 flies/genotype from 3 independent fly crosses.

• Again in Fig. 4D, the authors should clarify if the colocalization was performed in 3D, on single slices, or on Z-stack projections. The second will not provide very accurate estimate of the co-localization of the different proteins. Similar comment for the H panel in the same figure.

Response 3-7:

The colocalization was determined in 3D using the Imaris software (Oxford Instruments). Briefly, cells and tissues were automatically processed in 3D using the ImarisColoc plugin. We have added this paragraph in the revised manuscript (Materials and Methods/ Colocalization analysis) as shown below.

“For colocalization analysis, cells and tissues were automatically processed in 3D, as the deconvolved two-channel 3D image was exported to the ImarisColoc plugin of the microscopy image analysis software Imaris (Oxford Instruments) for quantitative analysis. The levels of colocalization were output as Pearson's coefficients for further statistical analysis.”

• There is a complete lack of information on the image analysis details in the Method section. Detailed description should be added, which might resolve part of the questions on the colocalization analyses.

Response 3-8:

We apologize for the omission. In the revised manuscript, we have added a paragraph in (Materials and Methods/ Colocalization analysis) as shown below:

“For colocalization analysis, cells and tissues were automatically processed in 3D, as the

deconvolved two-channel 3D image was exported to the ImarisColoc plugin of the microscopy image analysis software Imaris (Oxford Instruments) for quantitative analysis. The levels of colocalization were output as Pearson's coefficients for further statistical analysis."

- In general, the description of the results in Fig. 6 is insufficient thus the reader is left to interpret them on their own. In panel A, there is a contradictory effect of Taxol on tubulin accumulations in K/M or K/R flies, briefly justified by dosage effect. On the other hand, the Result section finalizes with a conclusion that " low dose of Taxol can enhance microtubule stability" which contradicts both with the data in panel A and G-J. Moreover, in panel G, there is a co-localization analysis of α -tubulin and acetylated tubulin, with a biased choice of inset in the lower panel of the DMSO treatment in the INSC shRNA. Looking at the representative cells in the INSC shRNA condition, the acetylated tubulin signal does not overlap with the α -tubulin one because the authors chose to zoom-in to an area that does not contain acetylated tubulin. The authors should zoom-in to areas that contain both signals in order to demonstrate the decrease in co-localization. Again here, it is unclear how the co-loc analysis was performed. Since there is already a data-set for all conditions, the authors should analyze the MT organization in these cells to support the claims that perhaps that there is MT-stability defect.

Response 3-9

We apologize for the poor choices of representative figures. The quantification was done by analyzing deconvolved two-channel 3D images using the ImarisColoc plugin. We now replace the original Figure 6GI with different representative figures where we zoomed in to areas that contain both signals. In the revised manuscript, we have also added a paragraph in (Materials and Methods/ Colocalization analysis) as shown below:

"For colocalization analysis, cells and tissues were automatically processed in 3D, as the deconvolved two-channel 3D image was exported to the ImarisColoc plugin of the microscopy image analysis software Imaris (Oxford Instruments) for quantitative analysis. The levels of colocalization was output as Pearson's coefficients for further statistical analysis."

Figure 6. Treatment of optimal concentration of Taxol rescued the morphological and functional defects in cell and aging fly.

- G. Immunostaining of *hINSC*-shRNA transfected SH-SY5Y cells and treated with DMSO (a vehicle control of Taxol) to visualize the colocalization of α -tubulin (red) and acetylated-tubulin (green) compared with scramble-shRNA control. Blue is DAPI staining. Scale bars: 10 μ m.
- I. Immunostaining of *hINSC*-shRNA transfected SH-SY5Y cells and treated with microtubule-stabilizer Taxol to visualize the colocalization of α -tubulin (red) and acetylated-tubulin (green) compared with scramble shRNA control. Blue is DAPI staining. Scale bars: 10 μ m.

Major concern regarding the overall text:

- The genetic and functional studies of CMT are a broad field that cannot be covered in a single introduction section. However, in its current shape the introduction does not even mention the complexity of the underlying pathomechanisms in CMT with different etiologies and the fact that in parallel to destabilized microtubules there are numerous other defects reported that might also contribute to the CMT pathology for the listed "several CMT2 subtypes, including CMT2D

(GARS1), CMT2E (NEFL), and CMT2F (HSPB1)".

Response 3-10:

We agree that the underlying pathomechanisms in CMT are complex with different etiologies. In the revised manuscript, we have modified the second paragraph of the introduction as shown below.

“Several common gene mutations have been linked to CMT2, including *MFN2* (CMT2A), *RAB7* (CMT2B), *GARS* (CMT2D), and *NEFL* (CMT2E). Previous literature shows that these mutations affect mitochondrial trafficking (CMT2A), microtubule trafficking of lysosomes (CMT2B), acetylated tubulin (CMT2D), and neurofilaments (CMT2E) (Markworth *et al*, 2021). Notably, these CMT2 subtypes share a common defect of decreased microtubule-stabilization caused by disrupting α -tubulin acetylation (Ackerley *et al*, 2006; Mo *et al*, 2018; Brownlees *et al*, 2002), which subsequently leads to microtubule breakdown and axonal defects. Since microtubules regulate axonal transport by forming a dynamic network that enables efficient intraneuronal transport, these findings suggest that microtubule destabilization within axons can be a common feature across genetically diverse forms of CMT2. Understanding the underlying genetic and molecular mechanisms can aid in improving the diagnosis, treatment, and management of CMT and provide crucial insights into the pathogenic mechanisms of CMT2.”

- The manuscript in its current shape does not coherently connect the different findings in the Results section. The text will benefit from extra information and improved description of the logic behind their results. For example, they start the result section (5) with the introductory sentence "Our results suggest reduced association of hINSCM70R with LGN may lead to MT instability in the adult PNS, as hINSC competes with NuMA for LGN-binding to regulate MT arrangement during spindle assembly in neuroblasts (Zhu *et al.*, 2011)." While the authors describe possible relocation of the PILS components in Fig. 4, this does not suggest MT instability. This jumping to unsupported conclusions is typical throughout the manuscript and the authors should revisit this writing style and provide contextual description of their results and their possible impact in more logic and toned-down manner.

Response 3-11:

We appreciate R3 for these constructive comments. We have revised the manuscript accordingly. Please see below as an example, where we restructured Results session (5) in a more logic and toned-down manner.

“Our results likely revealed an association between reduced association of hINSC^{M70R} with LGN and microtubule instability in the adult PNS. As hINSC competes with NuMA for LGN-binding (Zhu *et al.*, 2011) or form stable tetramers with LGN (Culurgioni *et al.*, 2018) to regulate microtubule arrangement during spindle assembly in neuroblasts, we tested if reduced hINSC^{M70R} binding with LGN impaired microtubule function in adult PNS. Upon examining the scolopidium, a microtubule - rich region in adult FeCO neurons, we identified large tubulin puncta accumulated extracellularly near *dInsc*-RNAi FeCO neurons between the muscle fibers (Fig 5A-C, and Fig EV1, B-D). In contrast, we did not observe tubulin aggregates in the brain (Fig EV1E). The tubulin aggregation was not due to overexpression of *tubulin-mCherry*, as we also observed aggregates with tubulin tracker to label the endogenous tubulin (Fig EV1F). The hINSC^{M70R} also caused similar aggregates that were absent in the hINSC^{WT} animals. Likewise, *pins*-RNAi, *bazooka*-RNAi, and the disease-relevant heterozygous *K/R* flies all exhibited tubulin aggregation in the proprioceptive structure, in which the severity of aggregation progressed over time (Fig 5D and E, and Fig EV4, A and B).”

- A brief literature browse on INSC demonstrates that the authors implement more up-dated references, and include also studies that might point to alternative conclusions regarding the INSC/LGN interaction. For example, the study of Culurgioni *et al.*, 2018 which suggests a possibility that fraction of the "Insc-bound pool of LGN acting independently of microtubule motors to promote asymmetric fate specification".

Response 3-12:

We thank R3 for the suggestion. In the revision, we include the main conclusion of Culurgioni *et al.*, 2018 in the 4th paragraph of introduction as shown below.

“Unlike *PAR3* and *LGN*, *INSC* has never been linked to any genetic disorder. INSC was first identified in *Drosophila* larval neuroblasts (Kraut & Campos-Ortega, 1996). *Drosophila* LGN is required for INSC to asymmetrically localize during asymmetric cell division (Yu *et al.*, 2000). INSC and LGN participate in the cytoskeleton-membrane association in the apical side of neuroblasts and induce pulling forces on the astral microtubule for the asymmetric division (Yu *et al.*, 2006). After the association between INSC and LGN, different modes of regulation on asymmetric division have been proposed. In one scenario, the dynein-adaptor protein NuMA competes with INSC for LGN binding (Zhu *et al.*, 2011). The LGN-bound NuMA complex then recruits dynein, a microtubule motor protein, to induce pulling forces on the astral microtubule for asymmetric divisions (Fig 1A) (Wang & Chia, 2005). Alternatively, INSC and LGN can form stable tetramers to regulate

asymmetric cell division without involving dynein (Culurgioni *et al*, 2018). Both findings indicate the importance of INSC-LGN association in regulating microtubules. LGN encodes an evolutionarily conserved tetratricopeptide repeat (TPR) motif that interacts with the LGN-binding domain of INSC (Fig 1B) (Yu *et al*, 2000, 2003). Whether the PIL complex, especially INSC, may be involved in CMT2 pathology via its role in microtubule regulation is not known.”

- The authors state that the Met70Arg substitution is located in the LGN-binding motif, which they claim that it is functionally conserved. As modelling the mutation in Drosophila is at the basis of their study the authors should show the protein sequence of this motif in evolutionary distant species to demonstrate the conserved motif. In this line, the absence (or presence) of highly conserved human/fly homolog should be clearly stated. This information, preferably accompanied by sequence alignment and/or diagrams of the human and fly homologs should be added. This will help the reader to understand readily the CRISPR CMT variant modeling and make sense of the variants generated.

Response 3-13:

We thank R3 for raising this issue, which is also pointed out by R2. The homology between human and fly INSC at the primary sequence level is not high, with sequence identity of 21 % and similarity of 32%. However, the M70R is located within a stretch of 38 amino acid peptides (human a.a. 70-118 or Drosophila a.a. 303-340) that are known to mediate polar and hydrophobic interactions with the N-terminal TPR domain of LGN, stabilizing the binding at nanomolar affinity (Culurgioni *et al.*, 2011. PMID: 22171003).

In the revised manuscript, we show sequence alignment of the LGN binding domain between hINSC and dInsc as shown below. The sequence alignment of this region is also demonstrated by previous literatures (Fig. 1C in Yuzawa *et al.*, 2011, PMID: 22074847; Fig. 3G in Culurgioni *et al.*, 2018, PMID: 29523789) showing that M70 resides in the N' of LGN-binding domain, and that fly K305 is the corresponding residue. We include this figure as the NEW Appendix Fig S5B.

B

Appendix Figure S5. Schematic of a seamless editing knock-in strategy and sequence alignment of LGN binding domain.

B. Sequence alignment of human and flies LGN-binding domain of INSC.

- While the authors do observe rescue of dInsc hypomorphs with the transgenic human INSC, the statement "providing the evolutionary conservation of INSC in neurodevelopment" should be accompanied by additional information on whether they specifically refer to their own result or there are also other references (that should be included).

Response 3-14:

Thank you for the suggestion. Indeed, the evolutionarily conserved role of INSC has previously been demonstrated. Specifically, Culurgioni *et al.*, 2011 (PMID: 22171003) showed that INSC::LGN interface is conserved between fly and human. Also Postiglione *et al.*, 2011 (PMID: 22017987) demonstrated that mInsc can functionally replace the fly protein in neuroblasts.

In the revised manuscript, we add these references to support our statement.

- The authors should provide more details on why was the scolopidium examined in fig.5 (besides for having MT-rich region). What is the function of these MT-rich regions in control flies? How and why would the tubulin puncta accumulate extracellularly? How is this relevant to modeling CMT?

Response 3-15:

We thank R3 for the suggestion. In the revised manuscript, we provide rationale why the scolopidium was examined. Please see below, which are added as the 5th paragraph of the introduction.

“The *Drosophila* femoral chordotonal organ (FeCO) is considered functionally homologous to

human muscle spindles, the primary proprioceptive sensory organs. (Tuthill & Azim, 2018), playing a critical role in detecting mechanical stretches such as muscle tension and joint position (Chen *et al.*, 2021). The chordotonal organ is formed by scolopidia, the basic unit of mechanoreceptor organ comprising over a hundred of mechanosensory neurons (Lipovšek *et al.*, 1999). FeCO neurons are crucial for the precise control of leg movements during behaviors like walking and target reaching. Proprioceptive cell death is known to cause neurological disorders (Ilieva *et al.*, 2008). Patients of Charcot-Marie-Tooth (CMT) disease show proprioception defects, making FeCO a suitable organ for studying peripheral neuropathy because of its microtubule-rich structure and highly conserved function. Taken together, the chordotonal organs in general, and the FeCO neurons in specific, provide an excellent platform for studying proprioceptive biology and unraveling the underlying mechanisms of peripheral neuropathy.”

Regarding the extracellular tubulin puncta, we re-examined its pattern using tubulin tracker, a tubulin dye that can penetrate through the leg cuticle. We observed different tubulin patterns between *dInsc* knockdown flies and the controls, as shown in the NEW Fig EV1F. In the control legs, the tubulin was well-organized in FeCO, representing microtubules. In contrast, *dInsc*-RNAi caused tubulin aggregates both intracellularly and extracellularly, which was also carefully examined as shown in Figure EV1C and D. As tubulin constitutes a significant portion of cellular components, our hypothesis suggests that the observed aggregates likely correspond to the remnants of a deceased cell.

Figure EV1. Expression of *dInsc* in FeCO neuron and its role in inducing tubulin aggregation.

F. Representative confocal images of tubulin aggregation in the FeCO neuron and the femur of 3-week-old *dInsc*¹⁴⁰⁷-Gal4>*dInsc*-RNAi flies, compared with the control (UAS-*lacZ*) and the rescue (*hINSC*^{WT} and *hINSC*^{M70R}) groups. Red indicates tubulin tracker signals. Green indicates the auto-fluorescence of cuticles. The FeCO neurons are encircled by the dashed line. Scale bars: 5 μ m.

Figure EV1. Expression of *dInsc* in FeCO neuron and its role in inducing tubulin aggregation.

C, D. Three-dimensional imaging of the femur of (D) *dInsc*¹⁴⁰⁷-Gal4>*dInsc*-RNAi compared with (C) control flies. The intercellular mCherry fluorescence shows tubulin aggregation between muscle fibers. Red labeled tubulin, cyan labeled phalloidin (muscle fibers), and green indicates the auto-fluorescence of the cuticle. Arrowheads indicate the aggregative tubulin. Scale bars: 5 μ m

Minor concerns:

- Spell out some of the acronyms to improve readability, for example ACD for asymmetric cell division.

Response 3-16:

We have spelled out the acronyms including ACD, LBD, and MT, to improve readability. We decide to keep the acronym PIL for the PAR3(PARD3)/INSC/LGN(GPSM2) complex, as it appears 33 times in the text.

- There are typos throughout the text and the figures.

Response 3-17:

We apologize for the typos. We have double-checked for typos in the text and the figures, and have corrected them in the revised manuscript.

- Clarify if the used transgenic d/hInsc are RNAi-resistant.

Response 3-18:

We performed qPCR to examine if the used transgenic *d/hInsc* are RNAi-resistant. As shown below, *dInsc*-RNAi targets *dInsc* but not *hINSC*. In the revised manuscript, we included these findings in Appendix Fig S4AB and in the corresponding Results.

Appendix Fig S4. *hINSC* transgenic fly mRNA and protein level validations

- A. Relative *dInsc* mRNA abundance in whole fly extracts of the indicated UAS transgenes driven by *elav-GS-Gal4*; n = 12 flies/genotype from 3 independent fly crosses.
- B. Relative *hINSC* mRNA abundance in whole fly extracts of the above indicated groups driven by *elav-GS-Gal4*; n = 12 flies/genotype from 3 independent fly crosses.

- The authors should clarify how they identified the different tissues in Fig. EV1A-B.

Response 3-19:

The fly CNS, leg disc, wing disc, and midgut have distinct structural morphologies that allow the recognition during dissection, as shown in the figure below. We have added a paragraph, as shown below, in Materials and Methods to clarify how the different tissues are identified.

“For dissection, all tissues from larvae or adult fly were dissected according to protocols as previously described (Lien *et al.*, 2020. PMID: 32627932; Guan *et al.*, 2018. PMID: 30451217). The leg discs and wing discs were identified according to its shape and relative position as described in Fig1: the 3rd instar larvae anatomy diagram as shown in Blair, 2007. PMID: 21356988.

Fig 1 in Blair, 2007 (PMID: 21356988)

- Are the metrics presented in fig. 2L-N established or the authors used them for a first time. The analogy that they try to make with human patient is understandable, however the different measurements should be put in a context.

Response 3-20:

The system to examine gait in flies was set up according to Wu *et al.*, 2019, PMID: 31246996. The metrics were also established in that paper, which was cited in Results and also in Methods. In the revised manuscript, we edit the Results (second paragraph) as shown below to explain how the gait parameters correspond to gait features of neurodegenerative diseases.

“In addition to recapitulating the molecular mechanisms, *Drosophila* models of neurodegenerative diseases have also been shown to share sufficient molecular machineries, allowing the measurement of locomotive characteristics such as gait and tremor (Gonçalves *et al*, 2022. PMID: 35158168; Sreedharan *et al*, 2015. PMID: 26234214). An automated leg tracking system was utilized to analyze and quantify leg trajectory in aged flies (Wu *et al*, 2019). We characterized gait features including footprint regularity, stride length, ratio of hind/mid (T3/T2) legs, and leg intersection domain. By comparing the gait signatures between the control and *dInsc* knockdown flies, significant changes were observed in gait patterns. Specifically, the *dInsc*-RNAi flies exhibited poor footprint regularity (Fig 2J and K), increased stride length in the hind (T3) leg (Fig 2L) and the ratio of hind/mid (T3/T2) legs (Fig 2M), as well as uncoordinated leg displacement with an enlarged leg intersection domain (Fig 2N). These alterations in gait resembled the walking difficulties and movement dysfunction observed in patients with CMT disease (Appendix Fig S7; Appendix Movie S2 and S3).”

References

- Ackerley S, James PA, Kalli A, French S, Davies KE & Talbot K (2006) A mutation in the small heat-shock protein HSPB1 leading to distal hereditary motor neuronopathy disrupts neurofilament assembly and the axonal transport of specific cellular cargoes. *Hum Mol Genet* 15: 347–354
- Beyer CF, Zhang N, Hernandez R, Vitale D, Lucas J, Nguyen T, Discafani C, Ayril-Kaloustian S & Gibbons JJ (2008) TTI-237: a novel microtubule-active compound with in vivo antitumor activity. *Cancer Res* 68: 2292–2300
- Blair SS (2007) Dissection of imaginal discs in *Drosophila*. *CSH Protoc* 2007: pdb.prot4794
- Brownlees J, Ackerley S, Grierson AJ, Jacobsen NJO, Shea K, Anderton BH, Leigh PN, Shaw CE & Miller CCJ (2002) Charcot-Marie-Tooth disease neurofilament mutations disrupt neurofilament assembly and axonal transport. *Hum Mol Genet* 11: 2837–2844
- Chen C, Agrawal S, Mark B, Mamiya A, Sustar A, Phelps JS, Lee WCA, Dickson BJ, Card GM & Tuthill JC (2021) Functional architecture of neural circuits for leg proprioception in *Drosophila*. *Curr Biol* 31: 5163-5175.e7
- Culurgioni S, Alfieri A, Pendolino V, Laddomada F & Mapelli M (2011) Inscuteable and NuMA proteins bind competitively to Leu-Gly-Asn repeat-enriched protein (LGN) during asymmetric cell divisions. *Proc Natl Acad Sci U S A* 108: 20998–21003
- Culurgioni S, Mari S, Bonetti P, Gallini S, Bonetto G, Brennich M, Round A, Nicassio F & Mapelli M (2018) Insc:LGN tetramers promote asymmetric divisions of mammary stem cells. *Nat Commun* 9
- Gonçalves AI, Zavatone-Veth JA, Carey MR & Clark DA (2022) Parallel locomotor control strategies in mice and flies. *Curr Opin Neurobiol* 73: 102516
- Guan W, Venkatasubramanian L, Baek M, Mann RS & Enriquez J (2018) Visualize *Drosophila* Leg Motor Neuron Axons Through the Adult Cuticle. *J Vis Exp* 2018
- Ilieva HS, Yamanaka K, Malkmus S, Kakinohana O, Yaksh T, Marsala M & Cleveland DW (2008) Mutant dynein (Loa) triggers proprioceptive axon loss that extends survival only in the SOD1 ALS model with highest motor neuron death. *Proc Natl Acad Sci U S A* 105: 12599–12604
- Lien WY, Chen YT, Li YJ, Wu JK, Huang KL, Lin JR, Lin SC, Hou CC, Wang HD, Wu CL, *et al* (2020) Lifespan regulation in α/β posterior neurons of the fly mushroom bodies by Rab27. *Aging Cell* 19
- Lipovšek S, Pabst MA & Devetak D (1999) Femoral chordotonal organ in the legs of an insect, *Chrysoperla carnea* (Neuroptera). *Tissue Cell* 31: 154–162
- Liu GY, Chen S, Lee G, Shaiv K, Chen P, Cheng H, Hong S, Yang W, Huang S, Chang Y, *et al* (2022) Precise control of microtubule disassembly in living cells. *EMBO J* 41

- Mamiya A, Sustar A, Siwanowicz I, Qi Y, Lu TC, Gurung P, Chen C, Phelps JS, Kuan AT, Pacureanu A, *et al* (2023) Biomechanical origins of proprioceptor feature selectivity and topographic maps in the *Drosophila leg*. *Neuron* 111: 3230-3243.e14
- Markworth R, Bähr M & Burk K (2021) Held Up in Traffic-Defects in the Trafficking Machinery in Charcot-Marie-Tooth Disease. *Front Mol Neurosci* 14
- Mo Z, Zhao X, Liu H, Hu Q, Chen XQ, Pham J, Wei N, Liu Z, Zhou J, Burgess RW, *et al* (2018) Aberrant GlyRS-HDAC6 interaction linked to axonal transport deficits in Charcot-Marie-Tooth neuropathy. *Nat Commun* 9
- Postiglione MP, Jüschke C, Xie Y, Haas GA, Charalambous C & Knoblich JA (2011) Mouse *inscuteable* induces apical-basal spindle orientation to facilitate intermediate progenitor generation in the developing neocortex. *Neuron* 72: 269–284
- Sreedharan J, Neukomm LJ, Brown RHJ & Freeman MR (2015) Age-Dependent TDP-43-Mediated Motor Neuron Degeneration Requires GSK3, *hat-trick*, and *xmas-2*. *Curr Biol* 25: 2130–2136
- Tuthill JC & Azim E (2018) Proprioception. *Curr Biol* 28: R194–R203
- Wu S, Tan KJ, Govindarajan LN, Stewart JC, Gu L, Ho JWH, Katarya M, Wong BH, Tan EK, Li D, *et al* (2019) Fully automated leg tracking of *Drosophila* neurodegeneration models reveals distinct conserved movement signatures. *PLoS Biol* 17
- Xiao H, Verdier-Pinard P, Fernandez-Fuentes N, Burd B, Angeletti R, Fiser A, Horwitz SB & Orr GA (2006) Insights into the mechanism of microtubule stabilization by Taxol. *Proc Natl Acad Sci U S A* 103: 10166–10173
- Yu F, Morin X, Cai Y, Yang X & Chia W (2000) Analysis of partner of *inscuteable*, a novel player of *Drosophila* asymmetric divisions, reveals two distinct steps in *inscuteable* apical localization. *Cell* 100: 399–409
- Yu F, Kuo CT & Jan YN (2006) *Drosophila* neuroblast asymmetric cell division: recent advances and implications for stem cell biology. *Neuron* 51: 13–20
- Yuzawa S, Kamakura S, Iwakiri Y, Hayase J & Sumimoto H (2011) Structural basis for interaction between the conserved cell polarity proteins *inscuteable* and Leu-Gly-Asn repeat-enriched protein (LGN). *Proc Natl Acad Sci U S A* 108: 19210–19215
- Zhu J, Wen W, Zheng Z, Shang Y, Wei Z, Xiao Z, Pan Z, Du Q, Wang W & Zhang M (2011) LGN/mInsc and LGN/NuMA complex structures suggest distinct functions in asymmetric cell division for the Par3/mInsc/LGN and Gai/LGN/NuMA pathways. *Mol Cell* 43: 418–431

4th Mar 2024

Dear Dr. Chan,

Thank you for the submission of your revised manuscript to EMBO Molecular Medicine. I am pleased to inform you that we will be able to accept your manuscript pending the following final amendments:

1) Figures: We note that some images/panels are reused. Figure 2E is reused in Figure EV1A. Please cite in the respective figure legend every reused image/panel.

2) In the main manuscript file, please do the following:

- Please address all comments suggested by our data editors listed below:

o Figure legends:

1. Please define the annotated p values *****/**/*** in the legend of figure EV 2a-b; EV 4a, c; as appropriate.

2. Please indicate the statistical test used for data analysis in the legends of figures EV 2a-b; EV 4a, c.

3. Please note that the error bars are not defined in the legends of figures EV 2a-b; Ev 4a, c.

4. Please note that the scale bar needs to be defined for figure EV 4a.

5. Please note that the white arrowheads are not defined in the legend of figure EV 4a. This needs to be rectified.

- All figures should be called out in a sequential order. Currently, Fig EV1B-F are called out after Fig EV2 and EV3. Please correct.

- In M&M, provide the statement the experiments involving human subjects conformed to the principles set out in WMA Declaration of Helsinki and the Department of Health and Human Services Belmont Report.

- Please confirm that a signed statement of informed consent to publish any identifiable format (video, recording, photograph, image) has been obtained from each person (parents or legal guardians for minors) who appears in a study. Please check "Author Guidelines" for more information: <https://www.embopress.org/page/journal/17574684/authorguide#humansubjects>

- In M&M, statistical paragraph should reflect all information that you have filled in the Authors Checklist, especially regarding randomization, blinding, replication.

- Please rename "Competing interests" to "Disclosure Statement & Competing Interests". We updated our journal's competing interests policy in January 2022 and request authors to consider both actual and perceived competing interests. Please review the policy <https://www.embopress.org/competing-interests> and update your competing interests if necessary.

- Author contributions: Please remove it from the manuscript and specify author contributions in our submission system. CRediT has replaced the traditional author contributions section because it offers a systematic machine-readable author contributions format that allows for more effective research assessment. You are encouraged to use the free text boxes beneath each contributing author's name to add specific details on the author's contribution. More information is available in our guide to authors:

<https://www.embopress.org/page/journal/17574684/authorguide#authorshipguidelines>

- Data availability: The journal encourages authors to provide access to genotype and clinical data with as few restrictions as possible while respecting ethical obligations to the patients and relevant medical and legal issues. A signed statement of informed consent to publish any human clinical, large-scale, and genomic datasets must be obtained from each person (parents or legal guardians for minors) who appears in a study. Please check "Author Guidelines" for more information <https://www.embopress.org/page/journal/17574684/authorguide#datadeposition>

3) Appendix: Please add page numbers to the table of content and remove movie legends.

4) Movies: Please rename them to Movie EV1 etc. (also in the main text) and zip each movie file with the corresponding movie legend.

1) Funding: Please merge it with the "Acknowledgements" and make sure that information about all sources of funding are complete in both our submission system and in the manuscript. Currently, project grants 12-2636-B-007-008 and 112-2628-B-007-004 and nd Brain Research Center, National Yang Ming Chiao Tung University from The Featured Areas Research Center Program within the framework of the Higher Education Sprout Project by the Ministry of Education (MOE) in Taiwan are missing in our submission system.

2) The Paper Explained: Please add it to the main manuscript file.

3) Synopsis: Every published paper now includes a "Synopsis" to further enhance discoverability. Synopses are displayed on the journal webpage and are freely accessible to all readers. They include separate synopsis image and synopsis text.

- Synopsis image: Please resize the image to 550 px-wide x (250-400)-px high and upload it as a high-resolution jpeg file .

4) For more information: This space should be used to list relevant web links for further consultation by our readers. Could you identify some relevant ones and provide such information as well? Some examples are patient associations, relevant databases, OMIM/proteins/genes links, author's websites, etc...

5) As part of the EMBO Publications transparent editorial process initiative (see our Editorial at

<http://embomolmed.embopress.org/content/2/9/329>), EMBO Molecular Medicine will publish online a Review Process File (RPF) to accompany accepted manuscripts. This file will be published in conjunction with your paper and will include the anonymous referee reports, your point-by-point response and all pertinent correspondence relating to the manuscript. Let us know whether you agree with the publication of the RPF and as here, if you want to remove or not any figures from it prior to publication.

6) Please provide a point-by-point letter INCLUDING my comments as well as the reviewer's reports and your detailed responses (as Word file).

I look forward to reading a new revised version of your manuscript as soon as possible.

Yours sincerely,

Zeljko Durdevic

*** Instructions to submit your revised manuscript ***

1) a .docx formatted version of the manuscript text (including Figure legends and tables)

2) Separate figure files*

3) supplemental information as Expanded View and/or Appendix. Please carefully check the authors guidelines for formatting Expanded view and Appendix figures and tables at <https://www.embopress.org/page/journal/17574684/authorguide#expandedview>

4) a letter INCLUDING the reviewer's reports and your detailed responses to their comments (as Word file).

5) The paper explained: EMBO Molecular Medicine articles are accompanied by a summary of the articles to emphasize the major findings in the paper and their medical implications for the non-specialist reader. Please provide a draft summary of your article highlighting

This may be edited to ensure that readers understand the significance and context of the research.

Please refer to any of our published articles for an example.

6) For more information: There is space at the end of each article to list relevant web links for further consultation by our readers. Could you identify some relevant ones and provide such information as well? Some examples are patient associations, relevant databases, OMIM/proteins/genes links, author's websites, etc...

7) Author contributions: the contribution of every author must be detailed in a separate section.

8) EMBO Molecular Medicine now requires a complete author checklist (<https://www.embopress.org/page/journal/17574684/authorguide>) to be submitted with all revised manuscripts. Please use the

checklist as guideline for the sort of information we need WITHIN the manuscript. The checklist should only be filled with page numbers where the information can be found. This is particularly important for animal reporting, antibody dilutions (missing) and exact values and n that should be indicated instead of a range.

9) Every published paper now includes a 'Synopsis' to further enhance discoverability. Synopses are displayed on the journal webpage and are freely accessible to all readers. They include a short stand first (maximum of 300 characters, including space) as well as 2-5 one sentence bullet points that summarise the paper. Please write the bullet points to summarise the key NEW findings. They should be designed to be complementary to the abstract - i.e. not repeat the same text. We encourage inclusion of key acronyms and quantitative information (maximum of 30 words / bullet point). Please use the passive voice. Please attach these in a separate file or send them by email, we will incorporate them accordingly.

You are also welcome to suggest a striking image or visual abstract to illustrate your article. If you do please provide a jpeg file 550 px-wide x 300-800px high.

10) A Conflict of Interest statement should be provided in the main text

11) Please note that we now mandate that all corresponding authors list an ORCID digital identifier. This takes <90 seconds to complete. We encourage all authors to supply an ORCID identifier, which will be linked to their name for unambiguous name identification.

Currently, our records indicate that the ORCID for your account is 0000-0003-2626-3805.

Please click the link below to modify this ORCID:
Link Not Available

Graphs 800-1,200 DPI
Photos 400-800 DPI
Colour (only CMYK) 300-400 DPI"

*Additional important information regarding figures and illustrations can be found at
<https://bit.ly/EMBOPressFigurePreparationGuideline>. See also figure legend preparation guidelines:
<https://www.embopress.org/page/journal/17574684/authorguide#figureformat>

***** Reviewer's comments *****

Referee #2 (Comments on Novelty/Model System for Author):

the authors have addressed my concerns in the revised manuscript.

Referee #3 (Comments on Novelty/Model System for Author):

The novelty of the study is based on the reporting of a novel causative gene, albeit with a single mutation, in CMT patients. The medical impact in the current version is medium but promising to be solidified in future studies. With addressing the major and minor concerns raised by the reviewers the technical quality of the manuscript improved significantly. Both the *Drosophila* and the cell lines the authors use are adequate and offer a parallel model systems that combine each ones powers to address disease-related research questions.

After addressing the major and minor comments of the reviewers, the manuscript significantly improved its quality. I do not have further concerns.

The authors addressed the minor editorial issues.

15th Mar 2024

Dear Dr. Chan,

We are pleased to inform you that your manuscript is accepted for publication and is now being sent to our publisher to be included in the next available issue of EMBO Molecular Medicine.
